# ACKR3 regulates platelet activation and ischemia-reperfusion tissue injury

Anne-Katrin Rohlfing [1], Kyra Kolb [1], Manuel Sigle[1], Melanie Ziegler[1], Alexander Bild[1], Patrick Münzer[1], Jessica Sudmann[1], Valerie Dicenta[1], Tobias Harm[1], Mailin-Christin Manke [1,2], Sascha Geue[1], Marcel Kremser [1], Madhumita Chatterjee [1], Chunguang Liang [3], Hendrik von Eysmondt [4], Thomas Dandekar [3], David Heinzmann[1], Manina Günter[5], Saskia von Ungern-Sternberg[1], Manuela Büttcher[1], Tatsiana Castor[1], Stine Mencl[6], Friederike Langhauser[6], Katharina Sies [7,8], Diyaa Ashour[9], Mustafa Caglar Beker[10], Michael Lämmerhofer [11], Stella E. Autenrieth[5], Tilman E. Schäffer [4], Stefan Laufer [12], Paulina Szklanna[13], Patricia Maguire[13], Matthias Heikenwalder [7], Karin Anne Lydia Müller[1], Dirk M. Hermann[14], Ertugrul Kilic[10], Ralf Stumm [15], Gustavo Ramos[9], Christoph Kleinschnitz[6], Oliver Borst[1,2], Harald F. Langer [16,17], Dominik Rath[1] & Meinrad Gawaz [1✉]

Platelet activation plays a critical role in thrombosis. Inhibition of platelet activation is a cornerstone in treatment of acute organ ischemia. Platelet ACKR3 surface expression is independently associated with all-cause mortality in CAD patients. In a novel genetic mouse strain, we show that megakaryocyte/platelet-specific deletion of ACKR3 results in enhanced platelet activation and thrombosis in vitro and in vivo. Further, we performed ischemia/ reperfusion experiments (transient LAD-ligation and tMCAO) in mice to assess the impact of genetic ACKR3 deficiency in platelets on tissue injury in ischemic myocardium and brain. Loss of platelet ACKR3 enhances tissue injury in ischemic myocardium and brain and aggravates tissue inflammation. Activation of platelet-ACKR3 via specific ACKR3 agonists inhibits platelet activation and thrombus formation and attenuates tissue injury in ischemic myocardium and brain. Here we demonstrate that ACKR3 is a critical regulator of platelet activation, thrombus formation and organ injury following ischemia/reperfusion.

[1] Department of Cardiology and Angiology, University Hospital Tübingen, Eberhard Karls University Tübingen, Tübingen, Germany. [2] DFG Heisenberg Group Thrombocardiology, University of Tübingen, Tübingen, Germany. [3] Department of Bioinformatics, Biocenter, University of Würzburg, Würzburg, Germany. [4] Institute of Applied Physics, University of Tübingen, Tübingen, Germany. [5] Department of Dendritic Cells in Infection and Cancer, German Cancer Research Centre, Heidelberg, Germany. [6] Department of Neurology, University Hospital Essen, Essen, Germany. [7] Division of Chronic Inflammation and Cancer, German Cancer Research Center (DKFZ), Heidelberg, Germany. [8] Department of Dermatology, University of Heidelberg, Heidelberg, Germany. [9] Immunocardiology Lab, University Hospital Würzburg, Comprehensive Heart Failure Center (CHFC), Würzburg, Germany. [10] Department of Physiology, School of Medicine, Istanbul Medipol University, Istanbul, Turkey. [11] University of Tübingen, Institute of Pharmaceutical Sciences, Pharmaceutical (Bio-) Analysis, Tübingen, Germany. [12] Department of Pharmaceutical and Medicinal Chemistry, Institute of Pharmaceutical Sciences, University of Tübingen, Tübingen, Germany. [13] Conway-SPHERE Research Group, Conway Institute, University College Dublin, Dublin, Ireland. [14] Vascular Neurology, Dementia and Ageing Research, Department of Neurology, Essen, Germany. [15] Institute of Pharmacology and Toxicology, Jena University Hospital, Jena, Germany. [16] University Hospital, Medical Clinic II, University Heart Center, Lübeck, Germany. [17] DZHK (German Research Centre for Cardiovascular Research), Partner Site Hamburg/Lübeck/Kiel, Lübeck, Germany. ✉email: meinrad.gawaz@med.uni-tuebingen.de

Platelets play a significant role in regulation of inflammation in various disease states, including acute myocardial[1] and cerebral ischemia[2], atherosclerosis[3,4], acute lung injury[5], or non-alcoholic steatohepatitis[6]. Platelets are a major source of inflammatory mediators that are stored in large quantities in granules of circulating platelets[7]. Upon activation, platelets rapidly release substantial amounts of cytokines and chemokines (e.g. macrophage migration inhibitor factor (MIF), C-X-C motif chemokines (e.g. CXCL4, 5, 12, 14) into the surrounding microenvironment[8]. Platelet-derived chemokines propagate infiltration of monocytes in areas of vascular and tissue injury and modulate thrombo-inflammation. Inhibition of platelet activation through antagonists such as aspirin or $P_2Y_{12}$-receptor inhibitors limits thrombus formation and inflammation[9]. Anti-platelet therapy has become a widespread pharmacological prevention strategy to improve clinical outcome in cardiovascular diseases. Most of the current anti-platelet drugs target mechanisms of platelet activation, and thereby effectively prevent thrombus-related organ ischemia; however, they increase the risk of severe bleeding complications[10]. Therefore, the development of strategies to control platelet activation, which preserves hemostasis and, thus, reduces bleeding tendency, is a major challenge for clinical medicine. Platelet activation is tightly regulated by inhibitory mechanisms that limit platelet accumulation at sites of vascular or tissue injury[11]. The most well-defined platelet inhibitory receptors are the $G_s$-coupled prostacyclin (prostaglandin $I_2$ ($PGI_2$)) receptor, the nitric oxide (NO) receptor soluble guanylate cyclase (sGC), and the immunoreceptor tyrosine-based inhibition motif (ITIM)-containing receptors platelet endothelial cell adhesion molecule 1 (PECAM-1) which are potential targets to control platelet-mediated thrombus formation with limited bleeding tendency[12]. Recently, we described that platelets express the chemokine receptor ACKR3 (*Atypical chemokine receptor 3*, formally known as CXCR7 (*C-X-C chemokine receptor type 7*) or RDC-1)[13,14]. The binding of ligands such as MIF to ACKR3 attenuates thrombus formation suggesting a role of platelet-ACKR3 in platelet inhibition[15]. Here, we addressed the role of platelet ACKR3 in platelet and organ function following ischemia/reperfusion and found that activation of platelet ACKR3 limits ischemic injury and thrombo-inflammation.

## Results

### Prognostic impact of surface expression of ACKR3 on circulating platelets in patients with coronary artery disease (CAD).

Previously, we showed that ACKR3 platelet surface expression is elevated in patients with acute coronary syndrome (ACS) when compared to those with chronic coronary syndrome (CCS)[16]. Furthermore, in patients with ST-elevation myocardial infarction (STEMI), elevated platelet ACKR3 is associated with improved recovery of myocardial function following reperfusion[17,18]. To assess the prognostic impact of ACKR3 platelet surface expression, we performed regression analysis to determine the impact of the expression levels of ACKR3 in a consecutive cohort of patients with symptomatic CAD ($n = 389$) (Supplementary Tables 4 and 5). Patients with high levels of platelet-ACKR3 surface expression (2nd/3rd tertile) had a more favorable outcome for all-cause mortality compared to patients with lower levels in a 3-years follow-up ($p = 0.012$) (Fig. 1a and Supplementary Table 4). Subgroup analysis revealed that the prognostic impact of high platelet-ACKR3 expression levels was more prominent in patients with ACS (Fig. 1b). Finally, CAD patients with high levels of platelet-ACKR3 showed reduced ex vivo platelet aggregation in response to thrombin-related activating peptide (TRAP) (platelet-ACKR3 high (2nd/3rd tertile) vs. low (1st tertile); mean AUC ± S.E.M.; $69.4 ± 2.0$ vs. $82.8 ± 2.6$, $p < 0.001$)

(Fig. 1c). In subgroup analyses, a decreased response to TRAP was observed in CCS ($64.1 ± 3.0$ vs. $81.3 ± 3.3$, $p < 0.001$) and ACS ($70.2 ± 2.6$ vs. $86.2 ± 3.8$, $p < 0.001$) patients (Fig. 1c). Multivariable cox-regression analysis with all-cause mortality as an independent variable and clinical factors as covariates revealed that platelet-ACKR3 is an independent predictor for all-cause mortality ($p < 0.01$) (Supplementary Table 5). These clinical results indicate that the chemokine receptor ACKR3 is of critical importance for the modulation of platelet reactivity and clinical outcome in patients with CAD.

### Generation and phenotyping of platelet-specific *Ackr3*-deficient mice.

Previously, we defined MIF as ligand for platelet ACKR3[15]. The binding of MIF to ACKR3 reduces platelet activation indicating a role of platelet-ACKR3 in thrombus formation[15]. To further define the relevance of platelet ACKR3 for platelet function and thrombo-inflammation, we generated a megakaryocyte-lineage-specific *ACKR3*-deficient knock-out mouse ($Ackr3^{-/-}$). Global genetic deletion of *Ackr3* results in embryonic lethality in mice[19].

We crossbred $Ackr3^{fl/fl}$ with transgenic mice expressing Cre-recombinase under the control of the Pf4-Cre promotor to obtain mouse strains deficient in platelet ACKR3 $Ackr3^{fl/fl}$ Pf4-Cre$^+$ ($Ackr3^{-/-}$) and $Ackr3^{fl/fl}$ Pf4-Cre$^-$ ($Ackr3^{fl/fl}$) littermates (Supplementary Fig. 1a, b). ACKR3 deficiency (molecular weight 42 kDa) in platelets and megakaryocytes was verified by immunoblotting and immunostaining using monoclonal antibody specific for ACKR3 (Fig. 2b, c and Supplementary Fig. 2c–e). Mice with $Ackr3^{-/-}$-deficient megakaryocyte/platelet lineage were viable, fertile, exhibited normal growth properties, gross morphology, immune cell composition, and an inconspicuous phenotype in contrast to an endothelial-specific ACKR3 deletion (Fig. 2d and Supplementary Figs. 2–4). Peripheral platelet count, surface receptor expression of GPV, GPVI, GPIX, integrin $α_5$ or $β_3$, or tail bleeding time, adhesion, spreading and morphology as verified by transmission electron microscopy (TEM), differential interference contrast (DIC), and scanning ion conductance microscopy (SICM) were not altered in $Ackr3^{-/-}$ compared to wild-type littermates (Fig. 2e–j and Supplementary Fig. 5).

### Loss of platelet ACKR3 promotes platelet activation and thrombus formation.

We found that release of α-granules of platelets derived from $Ackr3^{-/-}$ mice was significantly enhanced compared to controls as indicated by an increase in P-selectin (CD62P) surface expression upon CRP-stimulation ($p < 0.01$) but not thrombin stimulation (0.2 U/ml) (Fig. 3a and Supplementary Figs. 6 and 7). But the total amount and distribution of CD62P do not differ between $Ackr3^{-/-}$ and $Ackr3^{fl/fl}$ platelets (Supplementary Fig. 7b, c). Surface exposure of phosphatidylserine (Annexin V binding) was significantly increased in $Ackr3^{-/-}$ versus $Ackr3^{fl/fl}$ platelets upon CRP (1 μg/ml) and ABT737 (20 μM) stimulation (Supplementary Fig. 8). In contrast, release of ATP (dense-granules) was not altered in $Ackr3^{-/-}$ vs. $Ackr3^{fl/fl}$ platelets (Fig. 3b). Further, CRP (5 μg/ml) and thrombin (0.02 U/ml)-induced extrinsic calcium influx and intrinsic calcium release was significantly enhanced in $Ackr3^{-/-}$ compared to $Ackr3^{fl/fl}$ platelets (Fig. 3c). In addition, ex vivo platelet-dependent thrombus formation on immobilized collagen was significantly increased under flow (shear rate 1000 s$^{-1}$) in $Ackr3^{-/-}$ versus $Ackr3^{fl/fl}$ platelets (Fig. 3d). To analyze the impact of ACKR3 on thrombus formation in vivo, we performed intravital microscopy in $Ackr3^{-/-}$ and wildtype mice using the ligation of carotid artery[20–22]. As found in in vitro platelet function assays, ligation-induced carotid artery thrombosis was increased in $Ackr3^{-/-}$ compared to littermate controls (Fig. 3e). Thus, loss of platelet-

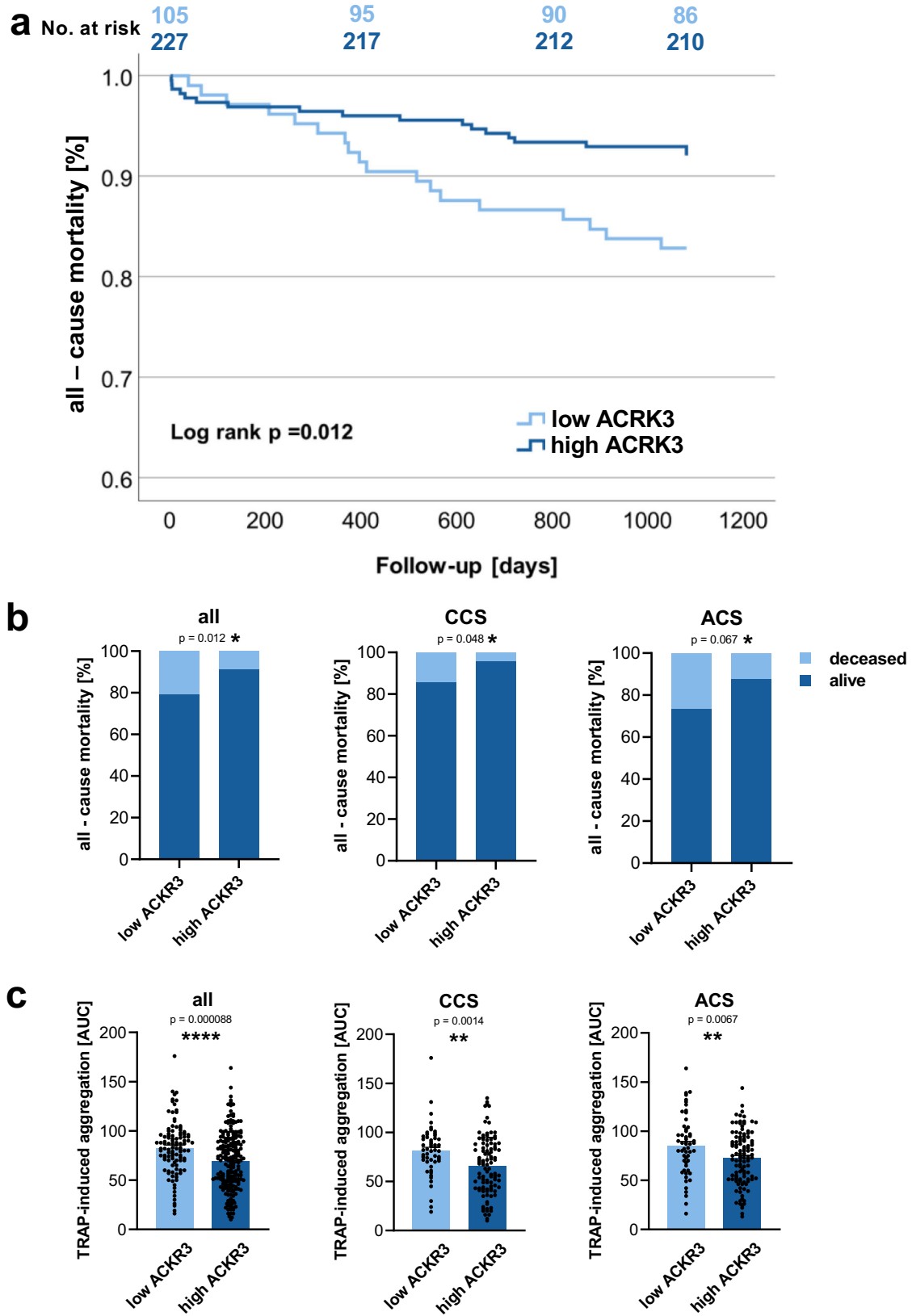

ACKR3 results in substantially enhanced platelet activation and thrombus formation.

**_Ackr3_<sup>−/−</sup> platelets promote thrombo-inflammation in vitro.** Upon activation, platelets release a variety of pro-inflammatory mediators that promote monocyte migration and formation of platelet/neutrophil co-aggregates[7]. Since $Ackr3^{-/-}$ platelets are hyperreactive upon activation, we were wondering whether loss of platelet-ACKR3 affects monocyte function. Thus, we tested the effect of supernatants derived from CRP-XL-activated (5 µg/ml) $Ackr3^{-/-}$ or $Ackr3^{fl/fl}$ platelets on human monocyte migration

**Fig. 1 Platelet surface expression of ACKR3 and clinical prognosis. a** Kaplan–Meier curves showing all-cause mortality stratified according to low (1st tertile) vs high (2nd and 3rd tertile) platelet ACKR3 surface exposure levels in patients with symptomatic CAD ($n = 320$). **b** Bar diagrams showing all-cause mortality stratified according to low (1st tertile) vs high (2nd and 3rd tertile) platelet ACKR3 surface exposure levels in the overall cohort as well as CCS ($n = 172$) and ACS ($n = 160$) patients. **c** Bar diagrams (mean ± S.E.M.) showing multiplate® multiple electrode aggregometry with thrombin receptor activating peptide (TRAP) stimulation stratified according to low (1st tertile) vs high (2nd and 3rd tertile) platelet ACKR3 surface exposure levels in the overall cohort (low ACKR3: $n = 107$; high ACKR3: $n = 213$) as well as CCS (low ACKR3: $n = 51$; high ACKR3: $n = 103$) and ACS (low ACKR3: $n = 55$; high ACKR3: $n = 111$) patients. Plotted: mean ± S.D.; statistics: two-tailed Student's $t$ test; 95% confidence interval.

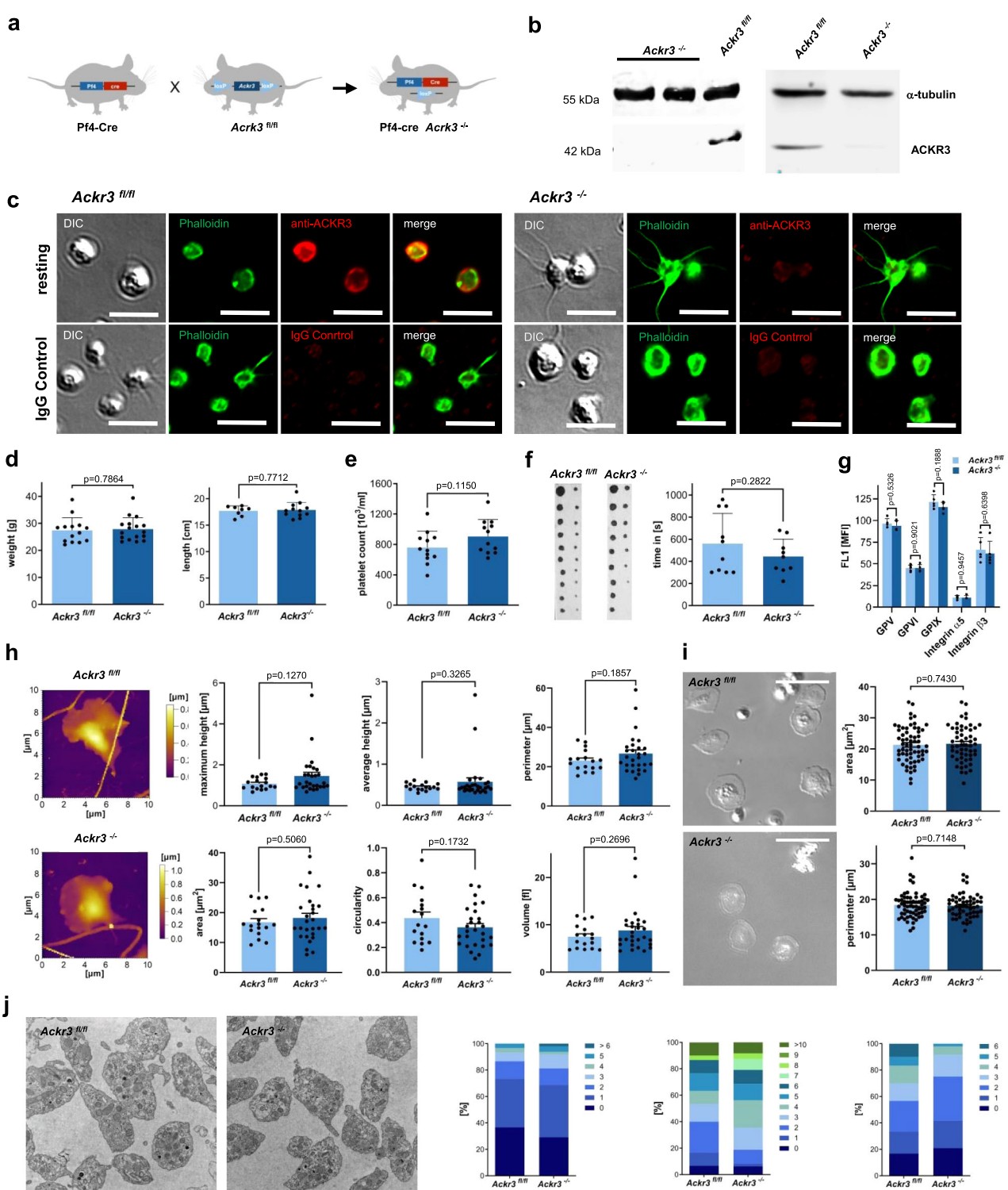

**Fig. 2 Generation and characterization of *Ackr3*⁻/⁻ Pf4-cre⁺ mice. a** Graphical presentation of the *Ackr3* Pf4-cre⁺ knock-out mouse generation. **b** Example of western blots (total *n* = 5) illustrating the loss of ACKR3 in isolated *Ackr3*⁻/⁻ platelets compared to *Ackr3*ᶠˡ/ᶠˡ. α-tubulin was used as loading control. **c** Example DIC image and immunofluorescence staining of *Ackr3*⁻/⁻ and *Ackr3*ᶠˡ/ᶠˡ platelets with anti-ACKR3 antibodies, phalloidin and corresponding IgG control (total *n* = 3; scale bar = 5 μm). **d** Body weight (*Ackr3*ᶠˡ/ᶠˡ *n* = 17; *Ackr3*⁻/⁻ *n* = 14;), body length (*Ackr3*ᶠˡ/ᶠˡ *n* = 12; *Ackr3*⁻/⁻ *n* = 8) compared between *Ackr3*⁻/⁻ (*n* = 14) and *Ackr3*ᶠˡ/ᶠˡ (*n* = 17). Plotted: mean ± S.E.M.; statistics: Student's *t* test (two tailed); 95% confidence interval. **e** Platelet count compared between *Ackr3*⁻/⁻ and *Ackr3*ᶠˡ/ᶠˡ animals (*n* = 12). **f** Bleeding time compared between *Ackr3*⁻/⁻ (*n* = 9) and *Ackr3*ᶠˡ/ᶠˡ (*n* = 10). **g** Statistical analysis of the receptor expression on *Ackr3*⁻/⁻ platelets compared to *Ackr3*ᶠˡ/ᶠˡ (*n* = 5). **h** Representative SICM images from *Ackr3*⁻/⁻ and *Ackr3*ᶠˡ/ᶠˡ platelets. Statistical analysis of the platelet images obtained by SICM of *Ackr3*⁻/⁻ (*n* = 28) and *Ackr3*ᶠˡ/ᶠˡ (*n* = 17) platelets. **i** Representative images of *Ackr3*⁻/⁻ (*n* = 55) and *Ackr3*ᶠˡ/ᶠˡ (*n* = 63) platelets spread on fibrinogen and activated by 1 μg/ml CRP (scale = 10 μm). Statistical analysis of the platelet images obtained by DIC microscopy of *Ackr3*⁻/⁻ and *Ackr3*ᶠˡ/ᶠˡ platelets. **j** Representative TEM images and organelle count in TEM cross-section of murine platelets of *Ackr3*⁻/⁻ and *Ackr3*ᶠˡ/ᶠˡ mice. Left: Dense bodies per platelet cross-section (>2 μm diameter) in murine platelets, *Ackr3*ᶠˡ/ᶠˡ (*n* = 30) vs *Ackr3*⁻/⁻ (*n* = 48). Middle: Alpha granules per platelet cross-section (>2 μm diameter) in murine platelets, *Ackr3*ᶠˡ/ᶠˡ (*n* = 30) vs *Ackr3*⁻/⁻ (*n* = 48). Right: Mitochondria per platelet cross-section (>2 μm diameter) in murine platelets, *Ackr3*ᶠˡ/ᶠˡ (*n* = 30) vs *Ackr3*⁻/⁻ (*n* = 48) (scale = 1 μm).

using a modified Boyden chamber system. We found that migration of monocytes towards supernatant derived from activated *Ackr3*⁻/⁻ platelets (APS) was substantially increased compared to *Ackr3*ᶠˡ/ᶠˡ control (Fig. 4a). Next, we analyzed the CRP-induced secretome with mass spectrometry of the CRP-induced platelet releasates from *Ackr3*⁻/⁻ (*n* = 3) and *Ackr3*ᶠˡ/ᶠˡ mice (*n* = 4). We identified 1.152 proteins in the CRP-induced releasate. Two proteins were significantly upregulated and 21 proteins downregulated in *Ackr3*⁻/⁻ mice (Supplementary Fig. 9). Most of these downregulated proteins are associated with mitochondrial metabolism and microvesicles. Furthermore, monocyte adhesion onto an immobilized layer of *Ackr3*⁻/⁻ platelets was substantially enhanced when compared to wild-type control *Ackr3*ᶠˡ/ᶠˡ platelets both under static and flow dynamic conditions (Fig. 4b, c). Co-aggregate formation of platelets and Ly6G⁺ cells upon activation with CRP-XL (1 μg/ml) in whole blood derived from *Ackr3*⁻/⁻ mice was also substantially enhanced when compared to wild-type littermates (Fig. 4d). Thus, loss of platelet-ACKR3 promotes platelet-dependent monocyte function and thrombo-inflammation.

**Deficiency in platelet-ACKR3 aggravates tissue injury following ischemia/reperfusion.** The remarkable increase in ACKR3 on circulating platelets of patients with acute myocardial infarction[16] and the impact on prognosis (Fig. 1) in patients with CAD implies that platelet-ACKR3 has a functional role in ischemia and reperfusion (I/R) injury. Thus, we investigated the role of platelet-ACKR3 in myocardial injury in *Ackr3*⁻/⁻ mice and their littermate controls using the transient LAD-ligation model[23,24]. Mice deficient in platelet-ACKR3 showed a substantial increase in myocardial injury as verified by an approximately 20% increase of infarct area compared to controls (% infarct of the area at risk (AAR), *Ackr3*ᶠˡ/ᶠˡ versus *Ackr3*⁻/⁻, mean ± S.D.: 31.1 ± 3.2% vs. 39.2 ± 6.3%, *p* < 0.02) (Fig. 5a). Echocardiographic functional parameters (ejection fraction and fractional shortening) were not different in *Ackr3*⁻/⁻ compared to controls (Supplementary Fig. 10). There was a significant accumulation of platelets early after reperfusion in the AAR as shown in immunohistological sections (Fig. 5b). Further, histological sections revealed enhanced numbers of infiltrating nuclear cells in AAR in *Ackr3*⁻/⁻ compared to controls (Fig. 5d). Cellular subtype analysis revealed an increased number of Ly6G⁺ but not CD3⁺, B220⁺, or MHCII⁺ in the AAR of myocardium derived from *Ackr3*⁻/⁻ compared to controls (Fig. 5c and Supplementary Fig. 11). Further, immunoreactivity of TNF alpha and IL-1beta was significantly enhanced in the AAR in *Ackr3*⁻/⁻ versus *Ackr3*ᶠˡ/ᶠˡ mice (Supplementary Fig. 12).

To further assess myocardial inflammation in AAR, we performed NanoString mRNA profiling (Supplementary Fig. 13 and Supplementary Table 3). Hierarchical clustering analysis showed that expression levels of mRNAs were significantly different in AAR in *Ackr3*⁻/⁻ versus *Ackr3*ᶠˡ/ᶠˡ mice. Of the 254 tested mRNA probes, six were upregulated and six were downregulated in *Ackr3*⁻/⁻ versus *Ackr3*ᶠˡ/ᶠˡ mice (Fig. 5e). We found a significant upregulation in AAR of inflammatory key molecules (CXCL5, CCL24, CCR2, TRAF2, STAT1, IRF5) in *Ackr3*⁻/⁻ versus *Ackr3*ᶠˡ/ᶠˡ mice (Fig. 5e). The most prominently downregulated mRNA signals involved key regulators of the toll-like receptor/inflammasome pathway (TLR6, NLRP3, HMGB1) (Fig. 5e). Pathway enrichment analysis (KEGG pathways) of the up-regulated genes in *Ackr3*⁻/⁻ ischemic myocardium identified the protein translation-related categories "chemokine signaling" and "toll-like receptor signaling," "necroptosis," "apoptosis," and "complement/coagulation" (Fig. 5e). Thus, genetic deletion of platelet-ACKR3 results in dramatically enhanced inflammation in the infarcted myocardium.

We also evaluated markers of systemic inflammation following I/R. The plasma of *Ackr3*⁻/⁻ and *Ackr3*ᶠˡ/ᶠˡ mice was obtained 24 h after reperfusion and analyzed for levels of chemokines and platelet/leukocyte interaction. Circulating platelet/macrophages co-aggregates (CD42b/CD14⁺) were significantly enhanced in *Ackr3*⁻/⁻ versus *Ackr3*ᶠˡ/ᶠˡ mice (Supplementary Fig. 14), indicating enhanced interaction of platelets with monocytes in *Ackr3*⁻/⁻ mice following I/R. Further, plasma levels of interleukin 6 (IL-6) were significantly enhanced in *Ackr3*⁻/⁻ versus *Ackr3*ᶠˡ/ᶠˡ mice 24 h after reperfusion. No significant changes were observed for plasma CXCL-12, CCL-2/-5, IL-1alpha/-beta, or TNF alpha (Supplementary Fig. 14). Thus, the data indicate that deficiency in platelet-ACKR3 has an impact on systemic inflammation after I/R.

To further evaluate the functional outcome following I/R we analyzed the extent of myocardial infarction and function after 28 days of I/R. We found that the area of the infarcted myocardium was substantially enhanced in *Ackr3*⁻/⁻ versus *Ackr3*ᶠˡ/ᶠˡ mice (Fig. 6). In *Ackr3*⁻/⁻ mice myocardial function was significantly impaired and sustained over 28 days of I/R compared to *Ackr3*ᶠˡ/ᶠˡ mice as verified by global strain echocardiography and ejection fraction (Fig. 6e, f and Supplementary Fig. 15). Interestingly, whereas in *Ackr3*ᶠˡ/ᶠˡ mice myocardial function almost completely recovered following I/R, the impairment of myocardial function was sustained in *Ackr3*⁻/⁻ mice. Decrease of myocardial function in *Ackr3*⁻/⁻ was paralleled by a substantial enhanced myocardial fibrosis in *Ackr3*⁻/⁻ compared to *Ackr3*ᶠˡ/ᶠˡ mice (Fig. 6c, d). Speckle tracking analysis revealed a loss of global longitudinal function (Fig. 6e).

The data provide evidence that platelet-ACKR3 deficiency results in a sustained and irreversible impairment of myocardial function and enhanced fibrosis, thus has an impact on functional outcome. These long-term analyses (28 days) combined with the enhanced myocardial inflammation early after reperfusion (24 h)

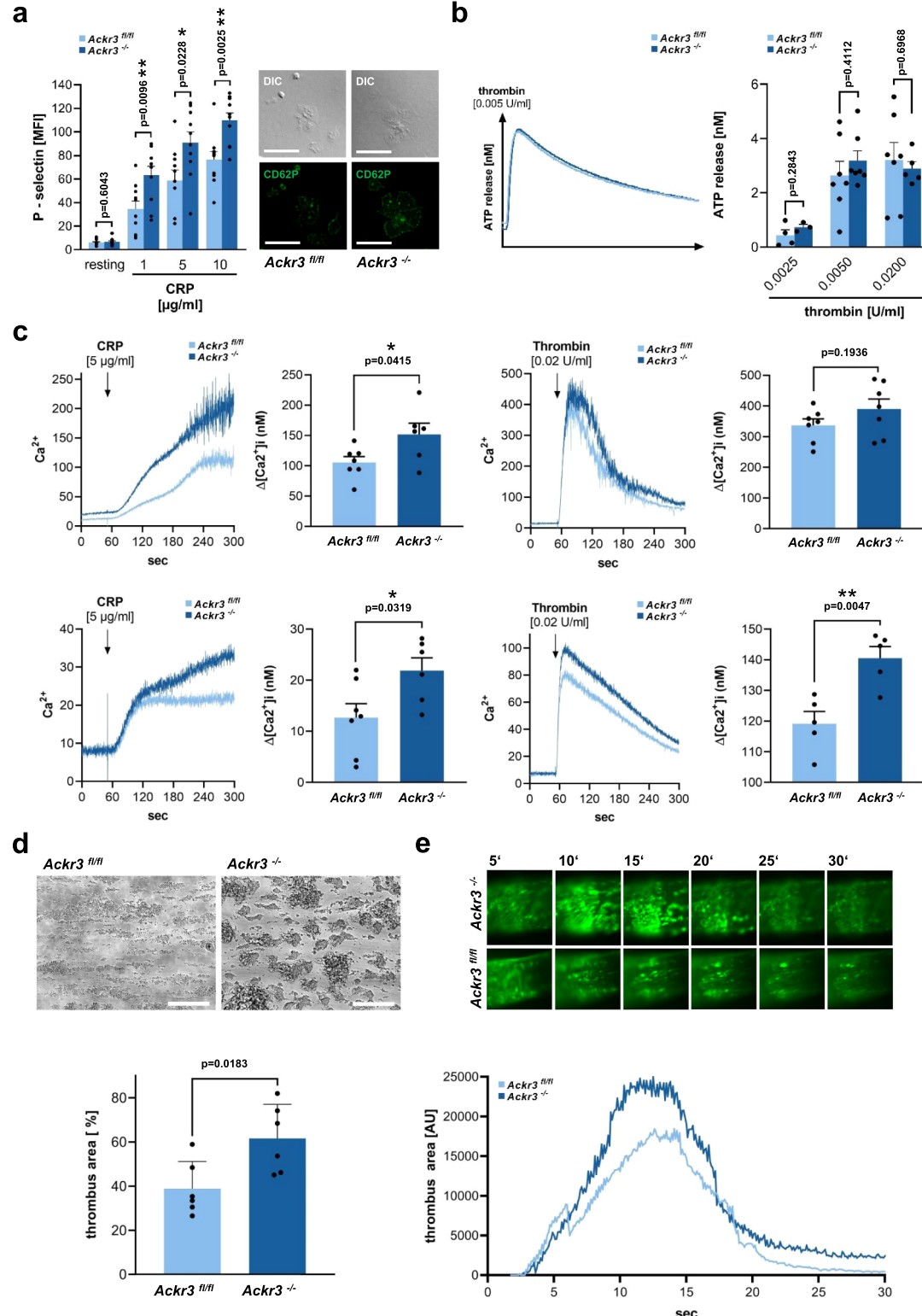

imply that enhanced platelet activation early in the time course of myocardial ischemia is associated with enhanced fibrosis and impairment of myocardium in the long term. Platelet ACKR3 seems to play a prominent role in enhanced platelet activation and triggers fibrosis development and thus disease progression.

Further, we quantified the extent of neovascularization by CD31 immunostaining of the infarct area of $Ackr3^{-/-}$ and $Ackr3^{fl/fl}$ mice after 28 days of reperfusion. We observed clear

neovascularization signals in the wild-type mice but the amount of CD31$^+$ vessel was significantly reduced in $Ackr3^{-/-}$ mice (Fig. 6g). These results indicate an impaired tissue regeneration in the $Ackr3^{-/-}$ animals.

We next analyzed whether loss of platelet-ACKR3 also affects brain injury in I/R. Similarly, to the myocardial I/R experiments, $Ackr3^{-/-}$ and $Ackr3^{fl/fl}$ mice were subjected to transient cerebral ischemia (tMCAO) for 30 min (Fig. 7a). After 24 h of reperfusion, infarct volumes were enhanced in $Ackr3^{-/-}$ compared to

**Fig. 3 Loss of platelet-ACKR3 promotes platelet activation and thrombus formation. a** P-selectin surface expression after stimulation on *Ackr3⁻/⁻* compared to *Ackr3ᶠˡ/ᶠˡ* platelets. resting: *n* = 11; CRP: *n* = 10; Plotted: mean ± S.E.M.; statistics: Student's *t* test (two-tailed); 95% confidence interval. **b** Representative ATP release curve and statistical comparison of *Ackr3⁻/⁻* and *Ackr3ᶠˡ/ᶠˡ* platelets upon activation with 0.005 U/ml thrombin. 0.0025 U/ml Thr: *Ackr3ᶠˡ/ᶠˡ n* = 4, *Ackr3⁻/⁻ n* = 3; 0.005 U/ml Thr: *Ackr3ᶠˡ/ᶠˡ n* = 7, *Ackr3⁻/⁻ n* = 7; 0.02 U/ml Thr: *Ackr3ᶠˡ/ᶠˡ n* = 7, *Ackr3⁻/⁻ n* = 5; Plotted: mean ± S.E.M.; statistics: Student's *t* test (two-tailed); 95% confidence interval. **c** Upper row: Representative calcium measurements and statistical analysis of the extrinsic calcium influx upon activation with 5 µg/ml CRP or 0.02 U/ml thrombin and addition of 1 mM $Ca^{2+}$ to the medium. 5 µg/ml CRP: *Ackr3ᶠˡ/ᶠˡ n* = 7, *Ackr3⁻/⁻ n* = 6; 0.02 U/ml Thr: *Ackr3ᶠˡ/ᶠˡ n* = 7, *v n* = 7; Plotted: mean ± S.E.M.; statistics: two-tailed Student's *t* test; 95% confidence interval. Lower row: Representative calcium measurements and statistical analysis of intrinsic calcium release upon activation with 5 µg/ml CRP or 0.02 U/ml thrombin and $Ca^{2+}$ deprivation by addition of 0.5 mM EGTA to the medium. 5 µg/ml CRP: *Ackr3ᶠˡ/ᶠˡ n* = 7, *Ackr3⁻/⁻ n* = 6, 0.02 U/ml Thr: *Ackr3ᶠˡ/ᶠˡ n* = 5, *Ackr3⁻/⁻ n* = 5; Plotted: mean ± S.E.M.; statistics: two-tailed Student's *t* test; 95% confidence interval. **d** Whole blood from *Ackr3⁻/⁻* and *Ackr3ᶠˡ/ᶠˡ* mice was perfused over a collagen-coated surface (100 µg/ml) for 15 min at a shear rate of 1,000 s⁻¹. Representative phase contrast images (scale = 100 µm) and statistical analysis of thrombus coverage shows a significant increase in coverage in *Ackr3⁻/⁻* animals compared to *Ackr3ᶠˡ/ᶠˡ*, *n* = 6. Plotted: mean ± S.D.; statistics: Student's *t* test (two-tailed); 95% confidence interval. **e** Representative images of the thrombus formation within the left carotid artery after a 5 min ligature and graphical representation of the thrombus area after ligature of the left carotid artery of *Ackr3⁻/⁻* compared to *Ackr3ᶠˡ/ᶠˡ* mice, *n* = 4.

littermates controls (mean ± S.D.: 49.3 ± 24.4 vs. 76.4 ± 65.7 mm³; *p* < 0.05) (Fig. 7a). Enhanced infarct size in *Ackr3⁻/⁻* mice was functionally relevant insofar that the grip test assessing global neurological function coordination was significantly worse after 24 h reperfusion in *Ackr3⁻/⁻* mice than in *Ackr3ᶠˡ/ᶠˡ* controls (grip test, mean ± S.D.: 4.0 ± 0.8 versus 2.7 ± 1.4, respectively; *p* < 0.02) (Fig. 7a).

Additional tMCAO experiments were performed to test whether the observed worse outcome in *Ackr3⁻/⁻* compared to littermates persists over time. Animals were subjected to tMCAO for 25 min to minimize animal loss by the procedure (Fig. 7b) and euthanized after a 7-day reperfusion period. Indeed, we observed the persistence of the difference in infarct size over time between wild-type and *Ackr3⁻/⁻* animals (mean ± S.D.: 14.5 ± 9.0 vs. 29.4 ± 14.9 mm³; *p* < 0.05) (Fig. 7b).

Again, similar to myocardial injury, NanoString mRNA profiling showed significantly upregulated key inflammatory mediators in ischemic brain tissue derived from *Ackr3⁻/⁻* and control littermates exposed to 24 h I/R (Fig. 7c). Of the 254 tested mRNA probes, nine were upregulated and nine were downregulated in *Ackr3⁻/⁻* versus *Ackr3ᶠˡ/ᶠˡ* mice (Supplementary Fig. 13 and Supplementary Table 3). We found a significant upregulation of key inflammatory molecules in ischemic brain tissue (CXCL5, CCL24, CHI3L3, IRF3, MMP9, PLA2G4A, RIPK1, NFATC3) in *Ackr3⁻/⁻* versus *Ackr3ᶠˡ/ᶠˡ* mice (Fig. 7c). The most prominently downregulated mRNA signals involved key regulators of the toll-like receptor/inflammasome pathway (RAC1, MAP2K4, PLCB1, GRB2, MAPK8, RAPGEF2, GRIB1). Pathway enrichment analysis (KEGG pathways) of the upregulated genes in *Ackr3⁻/⁻* identified the protein translation-related categories "metabolic pathways", "chemokine signaling", "toll-like receptor signaling", and "apoptosis". Analysis of individual downregulated genes showed the most significant "MAPK signaling" and "RAS signaling" categories (Fig. 7c). This indicates that similar to myocardial ischemia, loss of platelet-ACKR3 results in substantial cerebral inflammation.

As shown for myocardial injury, immunostaining data of brain tissue shows an enhanced accumulation of Ly6G⁺ cells in ischemic compared to non-ischemic areas after 24 h reperfusion (Fig. 8). Further, CD3⁺ cells were not different in ischemic versus non-ischemic areas similar to the findings in myocardial ischemia. The numbers of astrocytes and the neurons were not different (Fig. 8).

**Activation of ACKR3 inhibits platelet activation, thrombus formation, and myocardial injury following I/R.** Previously, we found that the interaction of MIF with ACKR3 attenuates thrombus formation[15]. To further investigate the role of platelet-ACKR3 on platelet function, we analyzed the effect of ACKR3

agonists (VUF11207, C10) on platelet activation. When isolated platelets were stimulated with ADP (5 µM) or CRP (0.5 µg/ml), the surface expression of P-selectin was significantly reduced compared to controls (Fig. 9a). CRP-induced (1 µg/ml) P-selectin surface expression was inhibited by the tested ACKR3 agonists in a dose-dependent manner with half-maximal effects in the range of 125 to 250 µM (Fig. 9a). No significant effect on CRP-induced platelet P-selectin expression was found in the presence of the control compound C46. Further, collagen-induced platelet aggregation was significantly reduced in the presence of the ACKR3 agonists (VUF11207, C10) compared to control (C46) (Fig. 9b). Additionally, activation of ACKR3 by agonists significantly attenuated flow-mediated thrombus formation on immobilized collagen (Fig. 9c). Similarly, in vivo thrombus formation at the site of arterial injury (ligation carotid injury model) was significantly reduced in the presence of ACKR3 agonist (Fig. 9d). In addition, the ACKR3 agonist (C10) significantly reduces CRP-induced $Ca^{2+}$ signaling compared to an irrelevant small molecule compound (C46) but has no effect on platelet spreading (Supplementary Fig. 16).

The endogenous ligands MIF and CXCL11 but not CXCL12 reduced platelet-dependent aggregate formation on immobilized collagen, similar to the ACKR3-agonist C10 (Supplementary Fig. 17). To test the potential involvement of CXCR4 in P-selectin expression, we pre-incubated platelets with AMD3100 or blocking monoclonal antibody directed against CXCR4. We found that CRP-induced P-selectin surface expression was not affected by the tested antibody/inhibitor (Supplementary Fig. 18). Further, the addition of recombinant CXCL12 to platelets prior to activation does not modulate CRP-induced P-selectin degranulation. When we performed this flow cytometric analysis in the presence of ACKR3-agonist, CRP-induced P-selectin expression was significantly reduced irrespectively of the presence of anti-CXCR4 or AMD3100 (Supplementary Fig. 18). Thus, CXCR4 seems not to be required for ACKR3-agonist-dependent P-selectin degranulation.

To test the specificity of the ACKR3 agonist, the effect of compounds on platelet degranulation (P-selectin) and platelet-mediated thrombus formation on immobilized collagen was analyzed in *Ackr3⁻/⁻* and *Ackr3ᶠˡ/ᶠˡ* platelets. Platelets or blood, respectively, obtained from *Ackr3⁻/⁻* mice did not show a reduction of CRP-induced (5 µg/ml) P-selectin expression (Fig. 9e) or thrombus formation under flow (Fig. 9f) in the presence of agonists compared to controls (*Ackr3ᶠˡ/ᶠˡ*), indicating that the tested agonists are specific for ACKR3.

To test whether systemic administration of an ACKR3-agonist affects myocardial injury following I/R, C57BL/6 J mice were treated intravenously with an ACKR3 agonist or control vehicle 60 min before LAD ligation and induction of ischemia (Fig. 9g). We found that following 24 h reperfusion, myocardial injury was

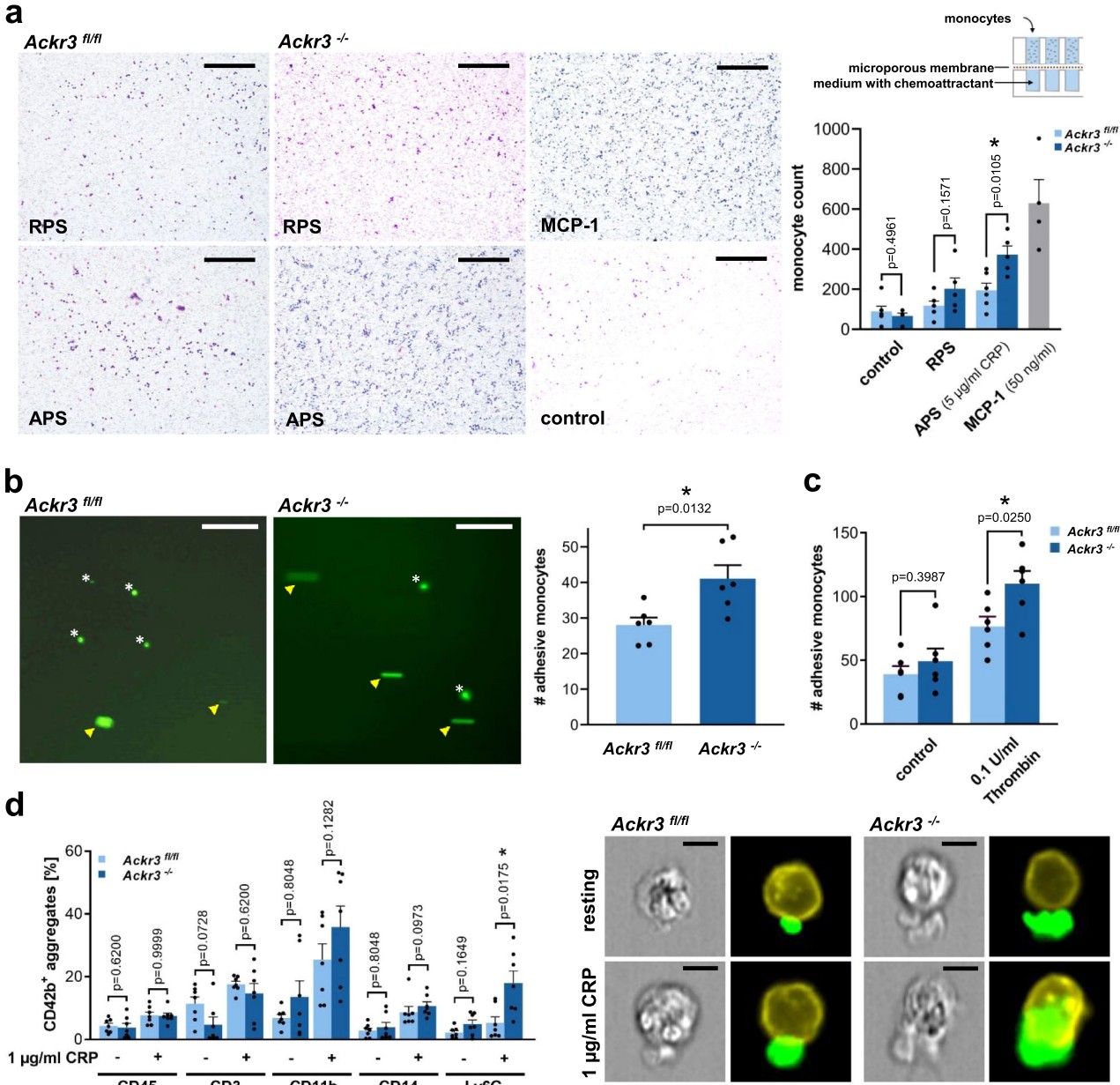

**Fig. 4 Ackr3⁻/⁻ platelets promote thrombo-inflammation in vitro. a** Schematic drawing of a modified Boyden chamber used for the following experiments and representative images (scale bar = 200 μm). Statistical analysis of the monocyte migration toward RPS and APS (5 μg/ml CRP-XL) derived from *Ackr3⁻/⁻* or *Ackr3fl/fl*. The migration of monocytes was enhanced towards APS derived from *Ackr3⁻/⁻* platelets compared to *Ackr3fl/fl* APS. 50 ng/ml MCP-1 was used as a positive control. *Ackr3fl/fl* n = 6, *Ackr3⁻/⁻* n = 5; MCP-1 n = 4; Plotted: mean ± S.E.M.; statistics: Student's *t* test (two-tailed); 95% confidence interval. **b** Representative images of flow chamber experiments were performed with isolated human monocytes. The monocytes were perfused over *Ackr3fl/fl* (left panel) and *Ackr3⁻/⁻* (right panel) platelets spread (1 μg/ml CRP) on fibrinogen-coated cover slides. Yellow arrowheads point at rolling monocytes whereas asterisks indicate adhesive monocytes (scale bar=100 μm). Statistical analysis of the number of rolling monocytes within 40 s of perfusion. An enhanced adhesion between *Ackr3⁻/⁻* platelets and human monocytes compared to *Ackr3fl/fl* platelets was observed, n = 6; Plotted: mean ± S.E.M.; statistics: two-tailed Student's *t* test; 95% confidence interval. **c** Statistical analysis of the static adhesion of *Ackr3⁻/⁻* platelets to murine monocytes compared to wild-type control platelets after 2 h incubation period. An enhanced adhesion between *Ackr3⁻/⁻* platelets and murine monocytes was observed compared to *Ackr3fl/fl* platelets after thrombin (0.1 U/ml) stimulation, n = 6; Plotted: mean ± S.E.M.; statistics: Student's *t* test (two-tailed); 95% confidence interval. **d** Statistical analysis of the spontaneous co-aggregate formation of *Ackr3fl/fl* and *Ackr3⁻/⁻* platelets measured by flow cytometry analysis under unstimulated and stimulated conditions (1 μg/ml CRP-XL). Image stream images from Ly6G⁺/platelet aggregates, n = 7; Plotted: mean ± S.E.M.; statistics: two-tailed Mann–Whitney *U*.

substantially reduced in the presence of the ACKR3-agonist as indicated by an approximately 30% reduction of infarct area compared to controls (% of the AAR, ACKR3-agonist vs. control, mean ± S.D.: 46.7 ± 6.7% vs. 33.2 ± 4.1%, *p* = 0.001) (Fig. 9g and Supplementary Fig. 19). Similar experiments performed in

*Ackr3⁻/⁻* mice revealed that the positive effect of the ACKR3-agonist on the infarct area was completely abolished in ACKR3-deficient mice (% of the AAR, ACKR3-agonist vs. control, mean ± S.D.: 54.8 ± 8.1% versus 58.1 ± 10.6%, *p* = n.s.), proving the specificity of the agonist treatment (Fig. 9g). Histological

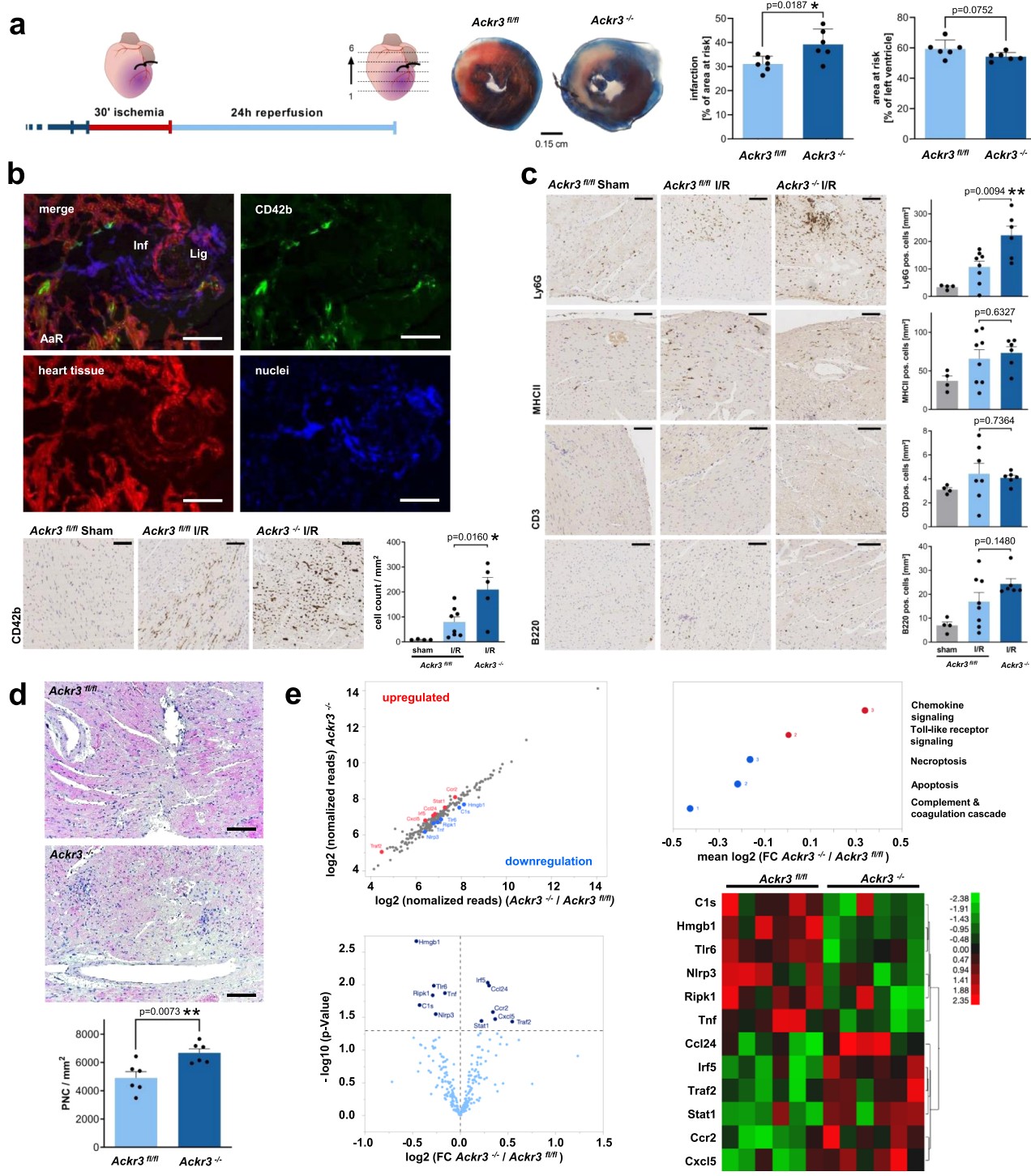

sections revealed a significant reduction of infiltrating nucleated cells in tissue AAR in ACKR3-agonist-treated *Ackr3<sup>fl/fl</sup>* mice compared to *Ackr3<sup>fl/fl</sup>* controls (Fig. 9h). This effect was completely abolished in *Ackr3<sup>−/−</sup>* mice (Fig. 9i). Hence, the results indicate that ACKR3 activation attenuates myocardial inflammation in I/R.

## Discussion
Activation of platelet surface receptors to control platelet activation and thrombo-inflammation has the potential to limit thrombus formation without increasing the risk of bleeding. Here,

we characterized a novel role of the chemokine receptor ACKR3 expressed on platelets for thrombus formation and organ injury following ischemia/reperfusion. Our major findings are (i) expression of ACKR3 on platelets in patients with symptomatic CAD is associated with clinical prognosis, (ii) genetic deficiency of platelet ACKR3 promotes platelet activation and tissue injury in ischemic myocardium and brain; (iii) loss of platelet ACKR3 aggravates tissue inflammation and systemic thrombo-inflammation; (iv) activation of platelet-ACKR3 mediates inhibitory effects on platelet activation and thrombus formation, and (v) attenuates tissue injury in ischemic myocardium and brain. Thus, targeting platelet ACKR may be an attractive strategy to control

**Fig. 5 Deficiency in platelet ACKR3 aggravates myocardial injury and inflammation following I/R. a** Graphic display of I/R of the heart, including representative images of Evans-Blue and TTC staining of heart sections (healthy tissue=blue, infarct=white, area at risk=red/white areas). Statistical comparison of infarction area [%] of left ventricle and area at risk [%] of left ventricle measured in $Ackr3^{-/-}$ and $Ackr3^{fl/fl}$ after I/R, n = 6; Plotted: mean ± S.D.; statistics: two-tailed Student's $t$ test; 95% confidence interval. **b** Representative fluorescence images of infarct area of $Ackr3^{-/-}$Pf4Cre$^+$ROSA animals with ubiquitous dTomato expression, except in Pf4Cre$^+$ platelets (endogenous GFP). Representative images (scale bar=100 µm) and statistical analysis of CD42b specific DAB staining of the infarct area. $Ackr3^{fl/fl}$ Sham: n = 4, $Ackr3^{fl/fl}$ I/R: n = 8, $Ackr3^{-/-}$: n = 5; Plotted: mean ± S.E.M.; statistics: two-tailed Student's $t$ test; 95% confidence interval. **c** Representative images (scale bar=100 µm) and statistical analysis of Ly6G, MHCII CD3, and B220-specific DAB staining of cell migration into infarct area. $Ackr3^{fl/fl}$ Sham: n = 4, $Ackr3^{fl/fl}$ I/R: n = 8 (CD3 $Ackr3^{fl/fl}$ I/R: n = 7), $Ackr3^{-/-}$: n = 6; Plotted: mean ± S.E.M.; statistics: Student's $t$ test; 95% confidence interval. **d** Representative HE staining images (scale bar=200 µm) and statistical analysis of the number of infiltrating cells/mm$^2$ into infarct area, n = 6; Plotted: mean ± S.E.M.; statistics: two-tailed Student's $t$ test; 95% confidence interval. **e** NanoString analysis of infarct area in $Ackr3^{-/-}$ and $Ackr3^{fl/fl}$ after I/R. Upper left: Scatter plot of mRNA expression showing significantly adjusted $p$ values ($p < 0.05$, upregulation=red, downregulation=blue; displayed: log2; n = 6). Upper right: KEGG pathway enrichment analysis of significantly altered mRNAs (downregulation = blue, upregulation=red). The number of different KEGG genes of each group (n = 5) is displayed next to it. Semi-quantitative alteration of genes is presented using log2-transformed fold change of mRNA. Lower left: Volcano plot analysis of significance detecting quantitative changes in mRNA levels. $p < 0.05$=dark blue, n = 6. Lower right: Heat map analysis of significantly regulated mRNAs (upregulation = red, downregulation = green; n = 6). Color density legend on the right indicates plotted variation in z-scores of mRNA expression.

thrombus formation and thrombo-inflammation in the early phase of acute organ ischemia.

ACKR3 (previously termed the RDC1 orphan receptor or C-X-C chemokine receptor type 7 (CXCR7)) is a 7 transmembrane receptor expressed on various cell types, including hematopoietic, vascular, and tumor cells[25,26]. ACKR3 is a receptor for chemokines (CXCL11 and CXCL12) and for the proinflammatory cytokine MIF[27–30]. The function and signaling capabilities of ACKR3 are partially defined and vary between cell types. ACKR3 has been postulated to act as a decoy receptor to clear and degrade its ligands after binding[31,32]. ACKR3 mediates signaling through recruitment of β-arrestin and activation of MAPK and AKT pathways[33,34]. ACKR3 is involved in tumor growth and metastasis[35,36] and cardiac development[19]. Further, ACKR3 regulates atherosclerotic lesion formation[37] and loss of endothelial ACKR3 impairs vascular hemostasis[38].

Previously, we identified ACKR3 on the platelet plasma membrane[13]. ACKR3 is expressed in platelets in substantial amounts as verified by immunoblotting. Most commercially available antibodies detect an immunoreactive band at approximately 55 kDa and a weaker signal at 50 kDa and 42 kDa (native). Molecular mobility is dependent on ubiquitination and glycosylation. The immunoreactive detection is varying regarding the epitope recognition of the specific antibody.

ACKR3 promotes platelet survival via AKT signaling, and ligation of the receptor with MIF limits thrombus formation[15]. To define the role of platelet ACKR3 for thrombus formation and organ ischemia, we generated a megakaryocyte/platelet-specific knock-out mouse. Mice deficient in platelet ACKR3 ($Ackr3^{-/-}$) show normal growth, normal gross morphology, and an inconspicuous phenotype in contrast to the global knock-out mouse (not viable) or the endothelial-specific ACKR3 knock-out (severe organ anomalies)[19]. Platelet count, expression of adhesion receptors, and morphology of platelets in $Ackr3^{-/-}$ mice was normal. However, loss of platelet ACKR3 resulted in substantial hyperreactivity and degranulation of platelets as indicated by enhanced platelet activation and thrombus formation. Further, we found that activation of ACKR3 through specific agonists (VUF11207, C10) inhibits platelet activation and thrombus formation. This indicates that ACKR3 controls and modulates platelet activation in response to the tested platelet agonists CRP, ADP, thrombin, or collagen. For in vitro studies we found a concentration-dependent reduction of CRP-induced P-selectin degranulation with a half-maximal effect between 125 and 250 µM. We are aware that the concentration used in our in vitro and in vivo experiments are comparatively high for specific low molecular weight compounds. However, regarding ACKR3, there

is a lack of well-characterized and easily available molecules for ACKR3 modulation. Both agonists used in our studies show a clear inhibition of platelet function. Despite the fairly high concentration both agonists specifically attenuated platelet thrombus formation in wild-type mice but not $Ackr3^{-/-}$ mice indicating a high specificity for ACKR3. Our data may stimulate the development of highly specific and potent ACKR3 modulators for further studies.

Previous human and animal studies showed that platelets become activated following myocardial infarction[10] and contribute to myocardial reperfusion injury. In the present study, we show that a reduced platelet ACKR3 level is associated with hyperresponsiveness and poor clinical outcome in patients with CAD. In the early phase of ischemia and reperfusion (I/R), platelets become trapped in the marginal ischemic zone and infiltrate the ischemic myocardium[39]. Targeting activated platelets with a bifunctional fusion protein protects against myocardial I/R injury[1]. This indicates that targeting platelets not only prevents thrombosis of the epicardial coronary arteries but also preserves myocardial function in I/R. Thus, we were wondering whether hyperreactive platelets ($Ackr3^{-/-}$ mice) promote I/R organ injury. Interestingly, we found that both in the myocardium and brain, the ischemic injury was substantially enhanced in $Ackr3^{-/-}$ mice compared to $Ackr3^{fl/fl}$ controls. This indicates and confirms previous studies that an enhanced reactivity of platelets is critical for tissue injury and its role in I/R is not organ-specific. Acute organ ischemia is associated with tissue and systemic inflammation. Hyperreactive platelets release potent inflammatory mediators that promote chemotaxis, migration, and differentiation of monocytes[40]. Platelets deficient in ACKR3 ($Ackr3^{-/-}$) show enhanced secretion of α-granules (P-selectin). Further, supernatants derived from activated $Ackr3^{-/-}$ platelets reveal a significantly enhanced promotion of chemotaxis of monocytes, and adhesion of $Ackr3^{-/-}$ platelets to monocytes is substantially increased compared to $Ackr3^{fl/fl}$ controls. Mass spectrometry analysis of activated platelet supernatants revealed only minor changes in the composition of the APS but an enrichment of mitochondrial- and microvesicle-associated proteins within the downregulated proteins in $Ackr3^{-/-}$ APS. This indicates that an altered secretome of $Ackr3^{-/-}$ platelets is associated with the proinflammatory response in $Ackr3^{-/-}$ mice. We found that myocardial inflammation following I/R is dramatically enhanced and prolonged in $Ackr3^{-/-}$ mice compared to wild-type littermates. Ischemic myocardium shows a substantially increased infiltration of PNCs in the AAR most prominently due to infiltrating Ly6G$^+$ cells, similar results were obtained in ischemic brains. The increased tissue inflammation was paralleled by an increase in

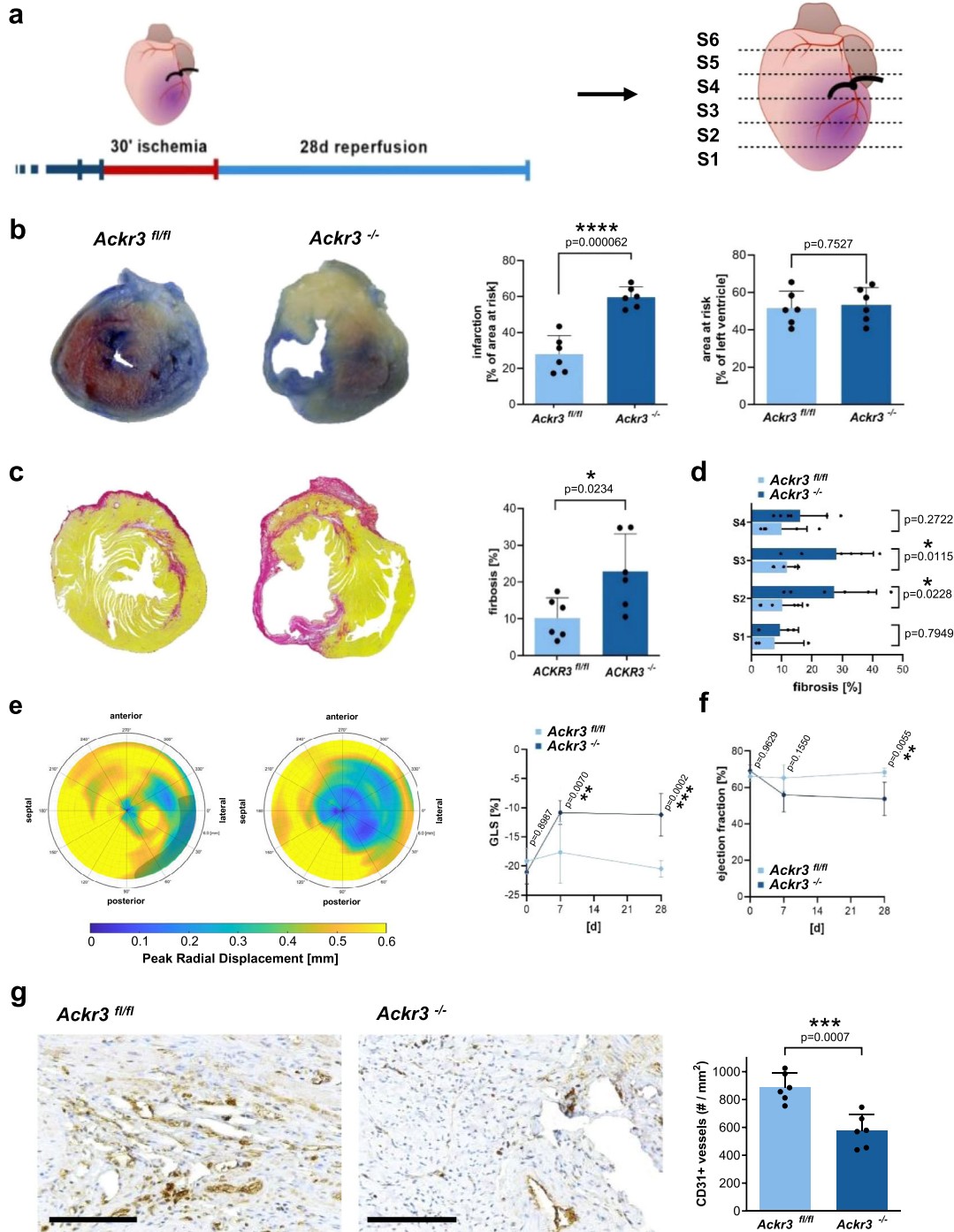

**Fig. 6 30 min ischemia and 28-day reperfusion of the heart in *Ackr3*<sup>−/−</sup> versus *Ackr3*<sup>fl/fl</sup> mice. a** Schematic drawing of the timeline and heart sectioning for long-term functional outcome of MI and reperfusion. **b** Representative TTC/Evans blue staining of the heart section. Statistical analysis of the area at risk of the left ventricle in percent and the infraction of the area at risk in percent, $n = 6$; Plotted: mean ± S.D.; Statistics: Student's *t* test; 95% confidence interval. **c** Representative pictures of Sirius Red staining of the fibrosis within the heart section 4 and statistical analysis of the fibrosis in the whole heart, $n = 6$; Plotted: mean ± S.D.; Statistics: Student's *t* test; 95% confidence interval. **d** Statistical analysis of the fibrosis separated by heart sections S1 to S4. S1: *Ackr3*<sup>fl/fl</sup>: $n = 3$, *Ackr3*<sup>−/−</sup>: $n = 3$; S2: *Ackr3*<sup>fl/fl</sup>: $n = 6$, *Ackr3*<sup>−/−</sup>: $n = 6$; S3: *Ackr3*<sup>fl/fl</sup>: $n = 6$, *Ackr3*<sup>−/−</sup>: $n = 6$; S4: Ackr3<sup>fl/fl</sup>: $n = 5$, *Ackr3*<sup>−/−</sup>: $n = 6$. Plotted: mean ± S.D.; Statistics: two-tailed Student's *t* test; 95% confidence interval. **e** Strain analysis of the heart in *Ackr3*<sup>−/−</sup> versus *Ackr3*<sup>fl/fl</sup> mice, $n = 6$; Plotted: mean ± S.D.; Statistics: two-tailed Student's *t* test; 95% confidence interval. **f** Statistical analysis of the ejection fraction of the heart over time in *Ackr3*<sup>−/−</sup> versus *Ackr3*<sup>fl/fl</sup> mice, $n = 6$; Plotted: mean ± S.D.; Statistics: two-tailed Student's *t* test; 95% confidence interval. **g** Representative images of CD31 staining and statistical analysis CD31<sup>+</sup> vessels in *Ackr3*<sup>−/−</sup> versus *Ackr3*<sup>fl/fl</sup> mice. Scale bar = 50 µm, $n = 6$; Plotted: mean ± S.D.; Statistics: two-tailed Student's *t* test; 95% confidence interval.

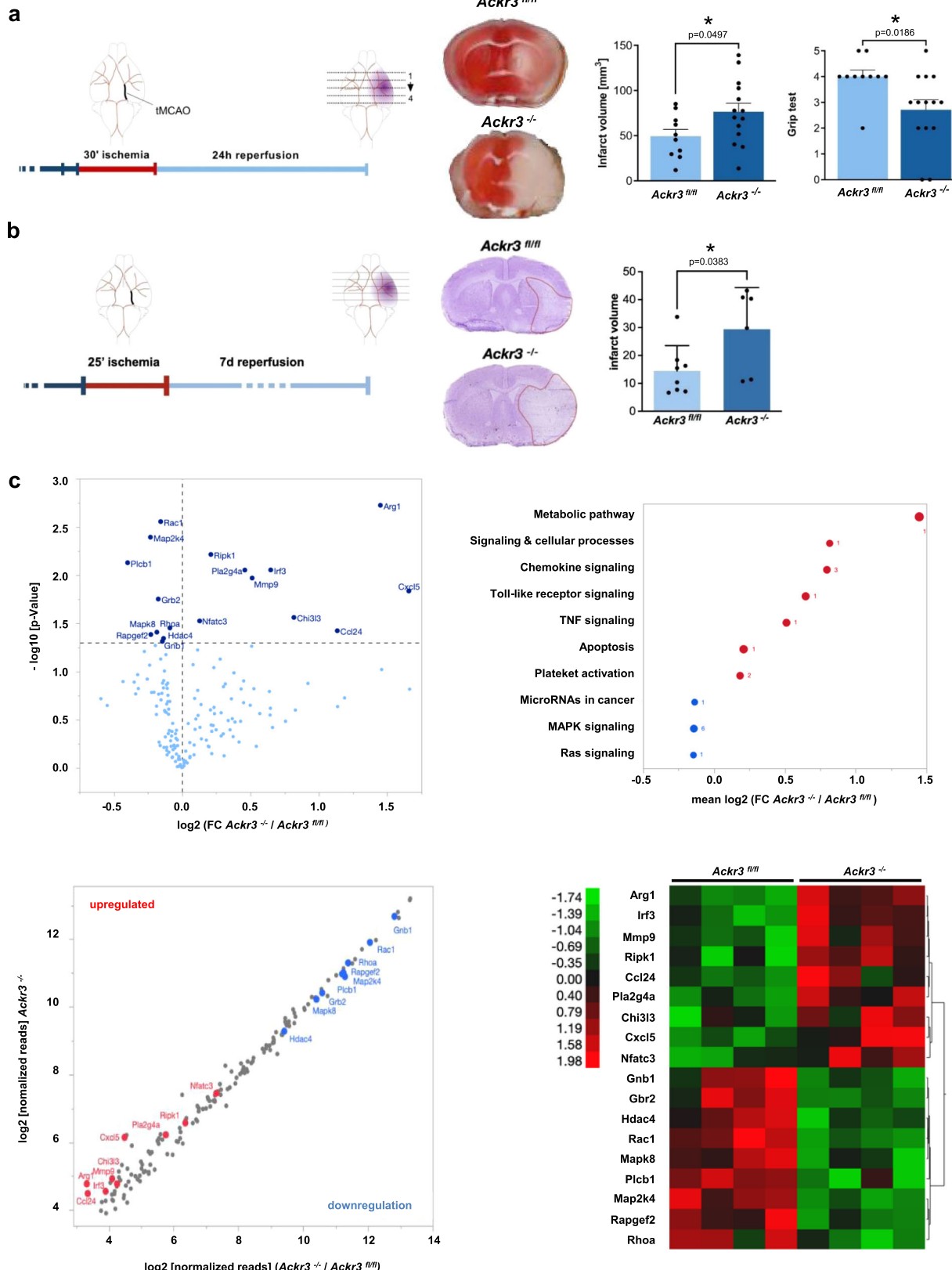

circulating platelet/Ly6G-monocyte co-aggregates in *Ackr3*⁻/⁻ compared to *Ackr3*^fl/fl control mice. Further, analysis of the transcriptome of 254 genes (NanoString) revealed a significant upregulation of mRNAs related to inflammation and chemokine pathways in both ischemic myocardium and brain of *Ackr3*⁻/⁻ mice. Thus, loss of platelet ACKR3 was associated with enhanced

and prolonged tissue injury and inflammation following I/R identifying the ACKR3 receptor on platelets as a critical regulator for thrombo-inflammation in I/R injury.

Next, we addressed whether activation of platelet ACKR3 using receptor-specific agonists affects platelet function. We found that in the presence of ACKR3-agonists ADP- and CRP-induced

**Fig. 7 Deficiency in platelet ACKR3 aggravates brain injury and inflammation following ischemia/reperfusion. a** Diagram of the tMCAO of the brain and representative images. Statistic comparison of the infarct volume and Grip test of *Ackr3*⁻/⁻ and *Ackr3*fl/fl after tMCAO. *Ackr3*fl/fl: *n* = 10, *Ackr3*⁻/⁻: *n* = 14; Plotted: mean ± S.E.M.: statistics: Student's *t* test; 95% confidence interval. **b** Graphic display of 7 days tMCAO of the brain, and representative images of the sections. Statistic comparison of infarct volume of *Ackr3*⁻/⁻ and *Ackr3*fl/fl after tMCAO and 7 d reperfusion. *Ackr3*fl/fl: *n* = 8, *Ackr3*⁻/⁻: *n* = 6; Plotted: mean ± S.D.; statistics: two-tailed Student's *t* test; 95% confidence interval. **c** Analysis of NanoString data obtained from brain section after tMCAO. Further statistical analysis of mRNA data was performed using one-way ANOVA of four different animals per genotype. Upper left: Scatter plot of mRNA expression levels in *Ackr3*⁻/⁻ versus *Ackr3*fl/fl. NanoString data showing adjusted *p* values (*p* < 0.05) of altered mRNAs, up- or downregulation is highlighted in red and blue respectively. Data displayed using log2 of mRNA reads (*n* = 4). Nine mRNA were significantly upregulated whereas eight showed downregulation in *Ackr3*fl/fl. Upper right: Pathway enrichment analysis (KEGG pathway database) of the significantly altered mRNA from the panel on the left. Coloring indicates upregulation or downregulation in the brain section of *Ackr3*⁻/⁻. The semi-quantitative alteration of mRNA is arrayed using log2-transformed fold change comparing *Ackr3*⁻/⁻ versus *Ackr3*fl/fl. The number of different KEGG genes of each group (*n* = 10) is displayed. Lower left: Volcano plot analysis of changes in mRNA levels in the infarct area of *Ackr3*⁻/⁻ and *Ackr3*fl/fl. Significant changes of NanoString data with adjusted *p* values (*p* < 0.05) are displayed in dark blue, *n* = 4. Lower right: Heat map analysis of significantly upregulated or downregulated mRNAs of all eight individuals comparing *Ackr3*⁻/⁻ and *Ackr3*fl/fl. Upregulated mRNA in brain sections are displayed in red, downregulation is shown in green. Color density legend on the right indicates the plotted variation in *z*-scores of mRNA expression.

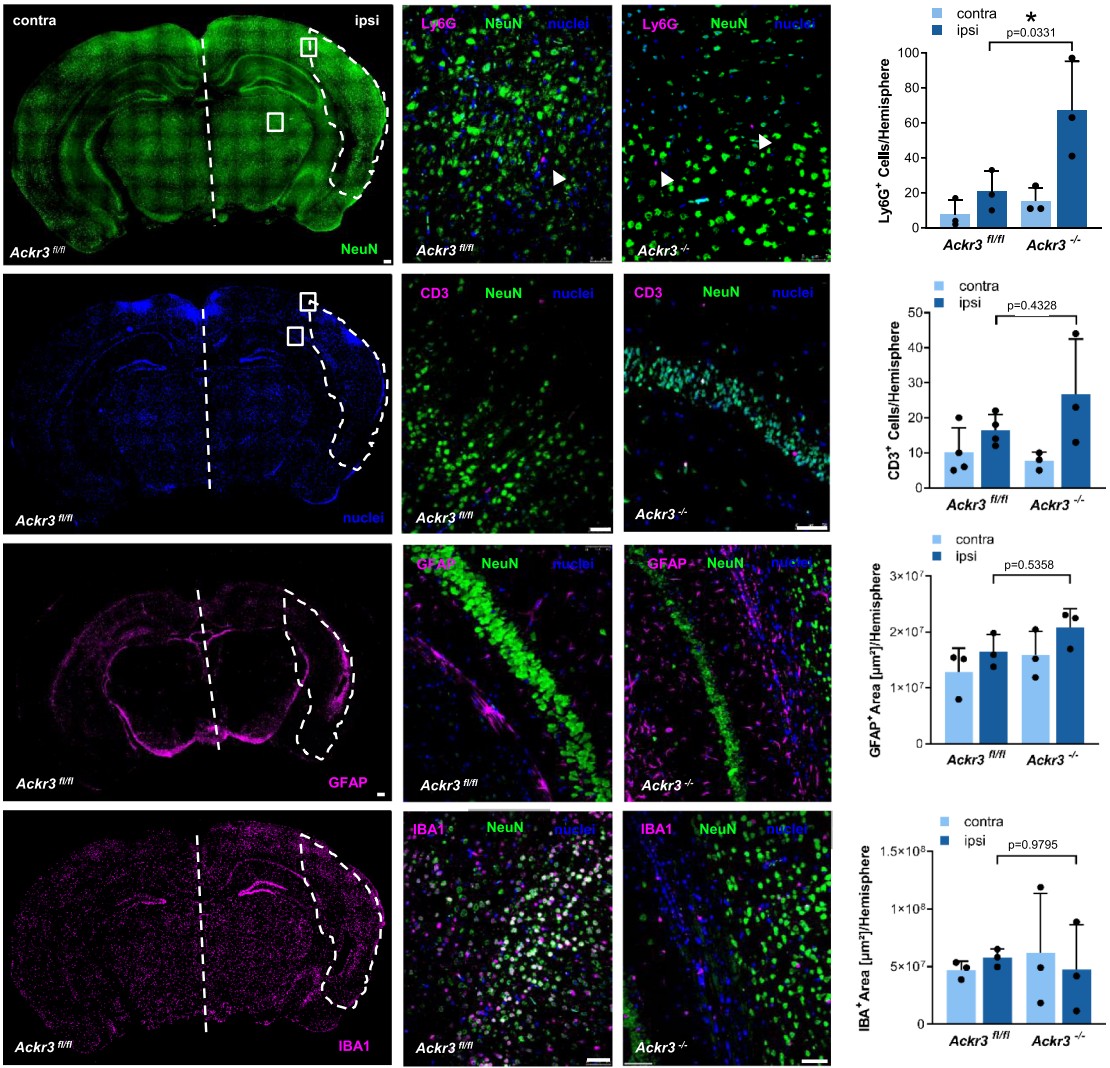

**Fig. 8 Deficiency in platelet ACKR3 results in altered cell migration into the brain following ischemia/reperfusion.** Immunostaining and statistical analysis of tMCAO brain sections. Ly6G⁺ staining, *n* = 3; CD3⁺ staining, *Ackr3*fl/fl: *n* = 4, *Ackr3*⁻/⁻: *n* = 3; GFPA⁺ staining representing activated astrocytes, *n* = 3; IBA⁺ staining, *n* = 3. Plotted: mean ± S.D.; statistics: one-way ANOVA, 95% confidence interval.

degranulation of α-granules (P-selectin), platelet-mediated thrombus formation ex vivo and in vivo was significantly attenuated suggesting that platelet ACKR3 acts as an inhibitory surface receptor. The specificity of the ACKR3-agonists was verified in experiments with *Ackr3*⁻/⁻ platelets. In contrast to wild-type platelets, the presence of ACKR3-agonists did not decrease activation-induced degranulation or thrombus formation under flow. Next, we tested whether systemic administration

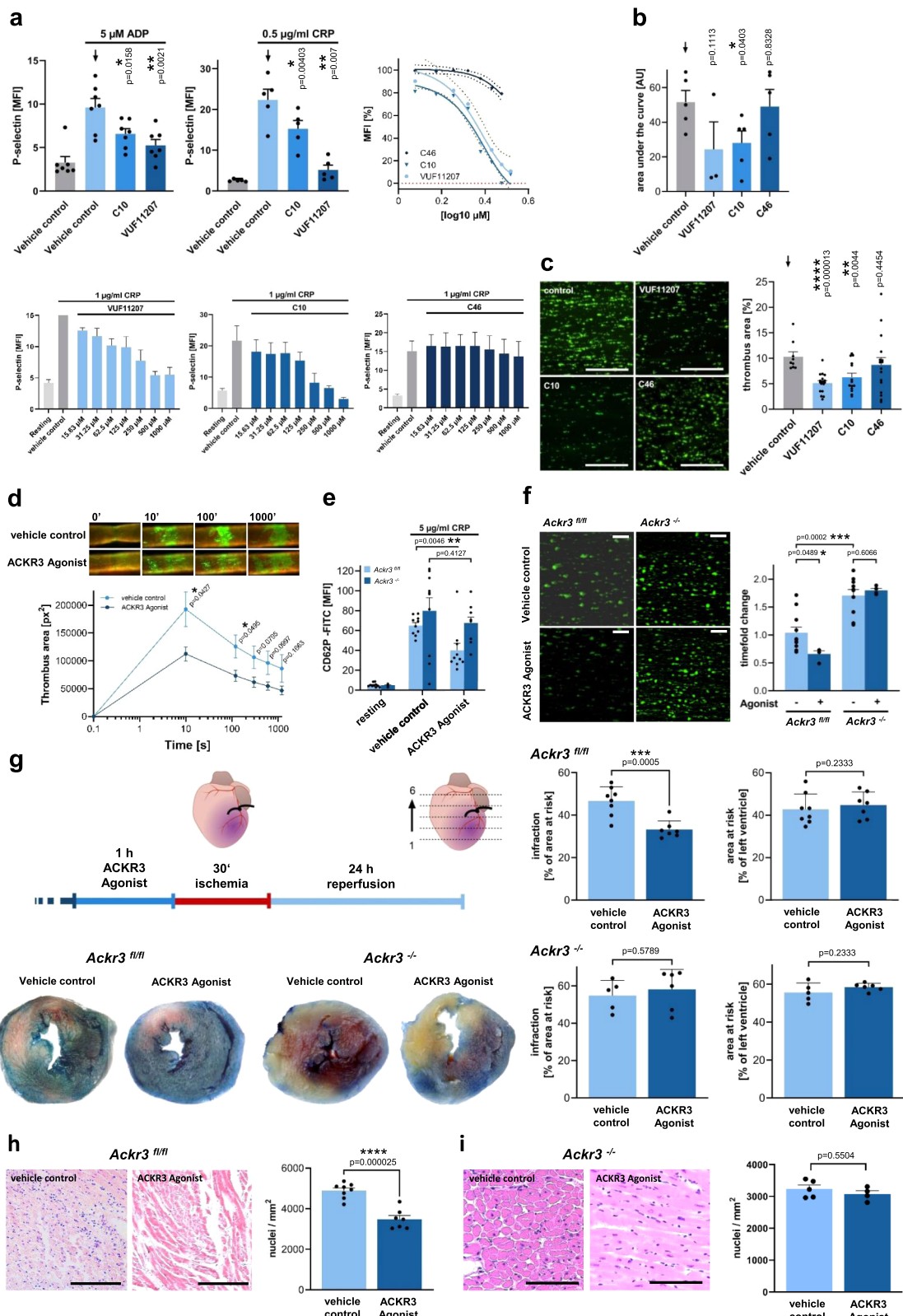

of ACKR3-agonist limits organ injury in I/R. Intriguingly, we found that wild-type mice treated with an ACKR3-agonist before transient ischemia revealed a reduced myocardial injury in I/R and reduced cell infiltration into the infarct area. This effect was completely abolished in $Ackr3^{-/-}$ mice treated with the ACKR3-agonist, indicating a high specificity of the agonist/receptor interaction for the ACKR3 receptor on platelets. Although

ACKR3 is expressed in various cell types, our data show that platelet ACKR3 is primarily involved in controlling I/R injury in the used experimental model.

In summary, our results indicate that platelet ACKR3 is a prominent receptor to control platelet hyperreactivity and organ ischemia in I/R. Recently, the importance of ACKR3 expression in endothelial cells has been shown to be critical for vascular

**Fig. 9 Activation of ACKR3 inhibits platelet activation, thrombus formation, and myocardial injury following I/R. a** Upper left: Statistical analysis of platelet P-selectin (CD62P) flow cytometry signals after treatment with ACKR3 agonists (100 μM). ADP: $n = 7$; CRP: $n = 5$; Plotted: mean ± S.E.M.; statistics: one-way ANOVA; 95% confidence interval. Upper right/lower row: Dose–response curves of P-selectin expression after treatment with ACKR3 agonists acquired by flow cytometry. **b** Impedance platelet aggregometry measurements of treated human blood. Control: $n = 5$; 100 μM VUF11207: $n = 3$; 100 μM C10: $n = 5$; control substance 100 μM C46: $n = 5$; Plotted: mean ± S.E.M.; statistics: Student's t test, 95% confidence interval. **c** Representative images of flow chamber experiments conducted with human blood (1000 s$^{-1}$ on collagen, scale bar = 100 μm) and statistical analysis of ACKR3 agonists effects upon thrombus area [%]. Control: $n = 9$; 100 μM VUF11207: $n = 17$; 100 μM C10: $n = 14$; 100 μM C46: $n = 16$; Plotted: mean ± S.E.M.; statistics: one-way ANOVA, 95% confidence interval. **d** Representative images of thrombus formation and analysis of thrombus area [px$^2$] within the left carotid artery after 5 min ligature of C57BL/6 J treated with ACKR3 agonist, $n = 5$; Plotted: mean ± S.E.M.; statistics: Student's t test. 95% confidence interval. **e** Flow cytometry measurements of murine platelets treated with ACKR3 agonist (100 μM C10; vehicle control: 1% DMSO), $n = 10$; Plotted: mean ± S.E.M.; statistics: Student's t test, 95% confidence interval. **f** Representative images of flow chamber experiments conducted with murine whole blood (1000 s$^{-1}$ on collagen; scale bar=100 μm) and statistical analysis of 100 μM C10 effects upon thrombus area [%]. Ackr3$^{fl/fl}$: $n = 11$; Ackr3$^{-/-}$: $n = 10$; 100 μM C10: $n = 4$; Plotted: mean ± S.E.M.; statistics: Student's t test, 95% confidence interval. **g** Effect of VUF11207 (300 μg) treatment on Ackr3$^{fl/fl}$ and Ackr3$^{-/-}$ upon I/R of the heart with statistical analysis. Ackr3$^{fl/fl}$ control: $n = 8$; Ackr3$^{fl/fl}$ Agonist: $n = 7$; Ackr3$^{-/-}$ control: $n = 5$; Ackr3$^{-/-}$ Agonist: $n = 6$; Plotted: mean ± S.D.; statistics: Student's t test, 95% confidence interval. **h, i** HE staining and statistical analysis of the infarct area **h** Ackr3$^{fl/fl}$ control: $n = 8$; Ackr3$^{fl/fl}$ Agonist: $n = 7$; **i** Ackr3$^{-/-}$ control: $n = 5$; Ackr3$^{-/-}$ Agonist: $n = 5$.

homeostasis and cardiac remodeling after myocardial infarction in mice[38]. Loss of endothelial ACKR3 exacerbated myocardial impairment after chronic occlusion of LAD[38]. Thus, it is tempting to hypothesize that both platelet (early phase of I/R) and endothelial (chronic occlusion and myocardial remodeling) ACKR3 play a critical role in the pathophysiology of myocardial repair following ischemia. Targeting ACKR3 in the early time course of reperfusion might not only limit tissue inflammation but also the subsequent formation of fibrosis and preservation of organ function.

To conclude, high platelet ACKR3 surface expression is independently associated with all-cause mortality in CAD patients. ACKR3 expressed on platelets is of prognostic relevance and regulates thrombosis and thrombo-inflammation. Activation of platelet ACKR3 is a critical mechanism to limit organ ischemia in I/R. Targeting platelet ACKR3 may be a promising novel strategy to control thrombosis and organ injury in I/R without enhancing bleeding.

## Methods

This study complies with all relevant ethical regulations and was approved by the ethics committee of the Medical Faculty of the Eberhard-Karls-University Tuebingen and the University Hospital Tuebingen (270/2011BO1 and 238/2018BO2) and complies with the declaration of Helsinki and the good clinical practice guidelines[41–43].

Animal handling and all animal experiments were performed according to the German animal protection law and were approved by the local authorities (Regierungspräsidium Tübingen M5/17, M20/15, M08/14 and M18/14 – myocardial infarction; University Hospital Essen approval 84-02.04.2017.A106 and Ethics committee of Istanbul Medipol University 16/08/2021-53—stroke).

**Reagents**. The reagents and antibodies used in the present study are summarized in (Supplementary Table 1). As ACKR3 agonist the commercially available high affinity ACKR3 chemical VUF11207 ($C_{27}H_{35}FN_2O_4$) was used. In addition, a non-commercially in-house developed agonist named C10 was used. C10 is an agonist ($C_{16}H_{20}N_6OS$) with an affinity for ACKR3 and less toxicity compared to VUF11207. C46 served as nonspecific control derivate (Supplementary Table 1).

**Statistical analysis of human samples**. For the cohort study, blood samples were collected during percutaneous coronary intervention (PCI) and immediately analyzed for ACKR3 platelet surface expression by flow cytometry and platelet aggregation levels by multiple electrode aggregometry (MEA) (Multiplate® Roche, Germany)[44]. Patients were admitted to the Department of Cardiology of the University of Tübingen, Germany. All subjects gave written informed consent. We included 389 consecutive patients with symptomatic coronary artery disease (CAD; chronic coronary symptom (CCS), $n = 184$; acute coronary symptom (ACS), $n = 205$) (Supplementary Table 2). ACS was defined as acute chest pain patient occurring with or without persistent ST-segment elevation as well as positive, or negative in case of unstable angina, cardiac enzymes[45]. Myocardial infarction (MI) was defined as an acute myocardial injury with clinical evidence of acute myocardial ischemia and with detection of a rise and/or fall of cardiac troponin (cTn) values with at least one value above the 99th percentile upper reference limit and at least one of the following symptoms of myocardial ischemia: new ischemic changes

in electrocardiogram (ECG), development of pathological Q waves in ECG, imaging evidence of new loss of viable myocardium or new regional wall motion abnormality, and identification of a coronary thrombus by angiography[46]. CCS included patients with suspected CAD who presented with stable angina symptoms or newly diagnosed heart failure/left ventricular dysfunction; asymptomatic or symptomatic patients with stabilized CAD < 1 year after ACS or patients with recent revascularization, as well as patients >1 year after initial diagnosis or revascularization; and patients with vasospastic or microvascular angina as well as asymptomatic subjects in whom CAD was detected at screening[47].

**Whole-blood flow cytometry of platelet ACKR3 plasma membrane expression**. Platelets in whole blood were analyzed for ACKR3 platelet surface exposure gating for the platelet-specific marker CD42b. Blood collected in citrate phosphate dextrose adenine was diluted 1:50 with Dulbecco's phosphate-buffered saline (PBS; Sigma Aldrich Co., St. Louis, MO, USA) and incubated with the respective conjugated antibodies, mouse monoclonal anti-human ACKR3-PE (R&D systems, Minneapolis, Minnesota, USA) and mouse anti-human CD42b-FITC (Beckman Coulter, Brea, California, USA) or their respective isotype controls (R&D systems) for 30 min at room temperature (RT). After staining, the cells were fixed with 0.5% formaldehyde and analyzed by flow cytometry (FACS-Calibur flow cytometer Becton-Dickinson, Heidelberg, Germany).

**Impedance platelet aggregometry**. The Multiplate® analyzer (Roche, Germany), a whole blood platelet function assay, was used to study the platelet aggregation level. A 600 μl blood sample acquired in hirudinized tubes (Sarstedt, Nümbrecht, Germany) was obtained to perform TRAP (32 μM) stimulation tests in CAD patients. The blood samples were collected via the catheter sheet. The area under the aggregation curve (AUC) was used as a measure for the overall platelet aggregation. Tests were performed 30 min to 3 h after taking blood.

**Follow-up**. All patients were tracked for all-cause mortality for 1080 days after study inclusion. In 14.7% of patients were lost during follow-up.

**Statistical analysis of data in human study**. All statistical analysis was performed using SPSS version 27.0 (SPSS Inc., Chicago IL). Normally distributed data were compared using Student's t test. Cross-tabulations were performed descriptively to show the number of endpoint distribution between ACKR3 levels 1$^{st}$ tertile (low-ACKR3) versus 2$^{nd}$/3$^{rd}$ tertile (high-ACKR3). Additionally, incidence rates per person-years are given. For censored data, Kaplan-Meier curves, log-rank tests, and hazard ratios were determined. In these analyses, ACKR3 levels were dichotomized between 1$^{st}$ tertile and 2$^{nd}$/3$^{rd}$ tertile. Multiple Cox regression analysis was applied to analyze independent associations of ACKR3 with the endpoints after adjustment for epidemiological factors influencing the cardiovascular outcome.

**Animal studies**. C57BL/6J and C57BL/6-Tg(Pf4-cre)Q3Rsko/J mice were obtained from Jackson Laboratories (The Jackson Laboratories, Bar Harbor, Maine, USA). The Ackr3$^{fl/fl}$ mouse strain was a kind gift of Dr. Fabienne Mackay, Victor Chang Cardiac Research Institute, Darlinghurst, Australia[19].

The Cre-loxP system was used to create a platelet-specific deletion of Ackr3 in an C57BL/6 background (B6.Cg-Thy1a-(Ackr3)Ackr3$^{tm1Fma}$-Tg(Pf4-icre)Q3Rsko/J). Ackr3$^{fl/fl}$ Pf4-Cre$^+$ (Ackr3$^{-/-}$) and Ackr3$^{fl/fl}$ Pf4-Cre$^-$ (Ackr3$^{fl/fl}$) offsprings were bred using this technique. A Ackr3$^{-/-}$ PF4-Cre$^+$ ROSA mouse (B6.Cg-Thy1a-(Ackr3)Ackr3$^{tm1Fma}$-Tg(Pf4-cre)Q3Rsko-Tg(ROSA$^{mT/mG}$)/J), with ubiquitous dTomato and a platelet specific eGFP expression, was generated in a similar manner. Wild-type littermates were used as control animals in all experiments.

Animals were genotyped by polymerase chain reaction using murine tissues samples obtained from the ear during the marking of freshly weaned mice according to the genotyping protocol provided by the Jackson Laboratory (JAX, Bar Harbor, Maine, USA) or with the primer listed in the original publication of the ACKR3 flox strain[19]. All mice experiments described in this study were conducted with female and male animals at the age of 10–12 weeks. Cervical dislocation under isoflurane anesthesia (5 vol%) was used to sacrifice all animals used in this study. Animal handling and all animal experiments were performed according to the German animal protection law and were approved by the local authorities (M5/17, M20/15, M08/14, and M18/14; 84-02.04.2017.A106 and 16/08/2021-53).

**Isolation of human and murine platelets**. Human and murine washed platelets were isolated as previously described[48]. Washed platelets were either lysed in RIPA lysis buffer for Western blot analysis or resuspended in Tyrode's buffer (pH 7.4, supplemented with 1 mM $CaCl_2$) for further experiments. For activation, isolated platelets were stimulated with collagen-related peptide (CRP-XL, CambCol, Cambridge, UK) or thrombin (F. Hoffmann La-Roche AG, Basel, Switzerland) in concentrations specified within the individual experiments at room temperature. Individual concentrations and exposure times are indicated within figures and legends.

**Supernatant from activated platelets**. To generate resting platelet supernatant (RPS), washed platelets were kept under resting conditions by the application of 0.2 U/ml apyrase (Sigma Aldrich Co., St. Louis, MO, USA) and 10 µM Pros-taglandin $I_2$ (Merck, Darmstadt, Germany). To gain activated platelet supernatant (APS), washed platelets were activated by the addition of 5 µg/ml CRP-XL (CambCol, Cambridge, UK) and incubated at room temperature for 30 min each. Afterward, both samples were centrifuged at $340 \times g$ for 5 min and the supernatants were used for further experiments.

**Mass spectrometry**. SDS PAGE short gel purification was run and in-gel digestion with Trypsin was conducted as described previously[49]. Extracted peptides were desalted using C18 StageTips[50] and subjected to LC-MS/MS analysis. LC-MS/MS analyses were performed on an Easy-nLC 1200 UHPLC (Thermo Fisher Scientific) coupled to an QExactive HF Orbitrap mass spectrometer (Thermo Fisher Scientific) as described elsewhere[51]. Peptides were eluted with a 60 min segmented gradient at a flow rate of 200 nl/min, selecting the 20 most intensive peaks for fragmentation with HCD.

Protein identification and quantification were performed as followed. Raw MS files were analyzed by MaxQuant version 1.6.2.1. MS/MS spectra were searched by the Andromeda search engine against human FASTA (October 2020) obtained from UniProt. MaxQuant analysis included an initial search with a precursor mass tolerance of 20 ppm the results of which were used for mass recalibration. In the main Andromeda search, precursor mass and fragment mass had an initial mass tolerance of 6 and 20 p.p.m., respectively. The search included fixed modification of carbamidomethyl cysteine. Minimal peptide length was set to seven amino acids and a maximum of two miscleavages was allowed. The false discovery rate (FDR) was set to 0.01 for peptide and protein identifications. For quantitative comparison between samples, we used label-free quantification (LFQ) with a minimum of two ratio counts to determine the normalized protein intensity. LFQ intensities were assigned to identified proteins by comparing the area under the curve of the signal intensity for any given peptide. Protein IDs were filtered to eliminate the identifications from the reverse database and common contaminants. LFQ values were Log2 transformed. A protein was included if it was identified in at least 50% of samples in at least one group.

**Megakaryocyte isolation**. To isolate megakaryocytes the femur was extracted from dead mice used in other experiments. The femur was cut open at one end and centrifuged for 1 min at $2500 \times g$. The resulting pellet was resuspended in 1 ml DMEM- Medium and centrifuged for 5 min at $300 \times g$. The resulting pellet was again resuspended in 1 ml DMEM-Medium (10% FCS, 1% Pen/Strep, and 1% TPO) filtered through a 70 µm cell mesh. After 5 days of cultivation, the cells were loaded onto a BSA gradient (1.5% and 3% BSA in PBS) and incubated for 30 min. The step was repeated three times and the resulting megakaryocyte pellet was used for further analysis.

**Immunoblot Analysis**. Proteins derived from isolated whole-platelet lysates or megakaryocytes were separated by sodium dodecyl sulfate polyacrylamide gel electrophoresis (SDS-PAGE) and subjected to immunoblotting using standard protocols. In short, samples were lysed by the addition of two volumes of RIPA buffer to 1 volume of sample. Samples were then incubated on ice for 10 min, further homogenized using an ultra-sound homogenizer and centrifuged for 5 min at 4000 r.p.m. The supernatant was collected and the protein content measured using a standard Bradford assay. The samples mixed with beta-mercaptoethanol containing standard 4× loading buffer and cooked for 10 min at 96 °C. Samples were loaded onto the SDS-Page applying an equal amount of total protein onto each lane. The electrophorese was performed for 30 min at 80 V and about 1 h at 120 V. Subsequently, the gel was blotted using a wet Blot set-up. The resulting PVDF membranes were incubated with primary and secondary antibodies (see

Supplementary Table 1). Equal loading of total protein was verified by α-tubulin immunostaining. Membranes were scanned and analyzed with the Licor Odyssey Infrared Imaging System (LI-COR, Bad Homburg, Germany).

**Tail bleeding time assay**. Mice were anesthetized intraperitoneally (100 µg/10 g body weight) using a combination of fentanyl (0.05 mg/kg), medetomidine (0.5 mg/kg) and midazolam (5 mg/kg). A 3 mm segment of the tail tip was removed with a scalpel. Tail bleeding was monitored by gentle absorption of the blood with filter paper at 20 s intervals without contacting the wound.

**Flow chamber assay**. Human whole-blood was incubated with the fluorochrome 3,3'-dihexyloxacarbocyanine iodide (1 mM $DiOC_6$, Sigma Aldrich Co., St. Louis, MO, USA) for 10 min at room temperature. Then 1 ml of the blood was perfused over a collagen-coated surface (100 µg/ml), through a transparent flow chamber with high $(1000 \ s^{-1})$ shear rates. In contrast, murine blood was anti-coagulated with heparin and diluted in Tyrode's buffer, supplemented with 1 mM $CaCl_2$, and was either stained with $DiOC_6$ for 10 min at room temperature or used directly. Afterward, the blood was perfused through a transparent flow chamber over a collagen-coated surface (shear rate = $1000 \ s^{-1}$). During the perfusion 1 min videos were taken (1 s/frame, Nikon Eclipse Ti2-A, ×20 objective) for the murine as well as human animal set-ups. Afterward, the chamber was rinsed, and pictures were taken of five representative areas (Nikon Eclipse Ti2-A, 20x objective). The covered area was analyzed using the NIS-Elements AR software (Nikon, Japan), and the mean percentage of the covered area was determined. In the case of agonist treatment, the blood was divided into samples, which were subsequently incubated for 30 min at room temperature either with an ACKR3 agonist or a vehicle control at the indicated concentrations.

**Platelet spreading**. Isolated murine platelets in Tyrodes buffer (pH 7.4) were supplemented with 1 mM $CaCl_2$, activated with 1 µg/ml CRP-XL (CambCol, Cambridge, UK), and incubated on fibrinogen-coated (100 µg/ml; Sigma Aldrich Co., St. Louis, MO, USA) coverslips for 30 min at room temperature. Afterward, platelets were fixed for 15 min with 4% paraformaldehyde (Sigma Aldrich Co., St. Louis, MO, USA) and washed three times with PBS (Sigma Aldrich Co., St. Louis, MO, USA). The coverslips were mounted onto slides and several images from randomly selected areas were taken (Nikon Eclipse Ti2-A, 100x DIC objective). The images were analyzed with the NIS-Elements AR software (Nikon, Japan). The classification of platelets into spreading stages was performed manually. Platelets on five randomly selected images per condition were counted.

**Scanning ion conductance microscopy**. Isolated washed platelets from $Ackr3^{fl/fl}$ and $Ackr3^{-/-}$ mice were stimulated with 0.1 U/ml thrombin (F. Hoffmann La-Roche AG, Basel, Switzerland) and allowed to adhere for 30 min to collagen-coated culture plates (100 µg/ml; Sigma Aldrich Co., St. Louis, MO, USA). Platelets were then fixed with 4% formaldehyde (Sigma Aldrich Co., St. Louis, MO, USA) and mounted into a custom-built scanning ion conductance microscopy set-up for imaging[52]. The microscope was operated in backstep/hopping mode using bor-osilicate glass nanopipettes with an opening radius of typically 50 nm. Topography images were recorded with a scan area of $10 \times 10 \ \mu m^2$ and a pixel size of 100 nm/px. Platelet morphology was analyzed with software custom-written in Igor Pro (Wavemetrics, Lake Oswego, Oregon, USA). Briefly, the shape of a cell was determined using a height threshold of 50 nm within a manually selected region of interest (to exclude other platelets or collagen fibers). Morphology parameters (area $A$, volume, mean and maximum height, and circularity $C = 4\pi A/P^2$ with perimeter $P$) were calculated from the shape.

**Transmission electron microscopy**. Resting murine platelets were isolated as described above under the presence of 0.2 U/ml apyrase (Sigma Aldrich Co., St. Louis, MO, USA) and 10 µM Prostaglandin $I_2$ (Merck, Darmstadt, Germany). The platelets were incubated for 1 h at room temperature in Karnovsky's solution and then stored at 4 °C. For TEM studies, cells were embedded in agarose at 37 °C, coagulated, cut in blocks, fixed again in Karnovsky's solutions, postfixed in osmium tetroxide, and embedded in glycid ether. Cutting of postfixed samples was per-formed with an ultramicrotome (Ultracut Reichert, Vienna, Austria). Ultrathin sections of 30 nm were mounted on copper grids and visualized using a Zeiss LIBRA 120 transmission electron microscope (Carl Zeiss, Jena, Germany).

**Measurement of ATP Release**. ATP release was triggered as described before[53]. In short, to measure ATP-release from activated platelets, washed platelets were adjusted to a cell count of 200,000 cells/µl in Tyrode's buffer (pH 7.4; supplemented with 1 mM $CaCl_2$). The samples were incubated for 2 min with luciferase (ChronoLume, Probe & Go Labordiagnostika GmbH). Afterward, thrombin was added in indicated concentrations and ATP release was measured using Chron-oLume luciferin assay (ChronoLog) with a lumino-aggregometer (ChronoLog, model 700) for 10 min (1000 r.p.m., 37 °C)[54].

**Measurement of cytosolic $Ca^{2+}$ concentration**. To measure the effect of the ACKR3 agonist on the cytosolic calcium concentration, experiments were

performed in washed human platelets loaded with 5 nmol/ml fura-2 acetoxymethylester (Invitrogen, Carlsbad, California, USA) containing 0.2 μg/ml Pluronic F-127 (Biotium, Hayward, California, USA) for 30 min in the presence of 100 μM C10 or 100 μM C46. Subsequently, loaded platelets were activated with 5 μg/ml CRP-XL and calcium responses were measured under stirring condition with a spectrofluorometer (LS 55; PerkinElmer) at alternate excitation wavelengths of 340 and 380 nm (37 °C). The 340/380-nm ratio values were converted into concentrations of $Ca^{2+}$.

**Isolation of human monocytes**. Human monocytes were isolated as described previously[55]. Briefly, monocytes were isolated by centrifugation of citrate-phosphate-dextrose-adenine (CPDA) coagulated blood (S-Monovettes, Sarstedt, Nümbrecht, Germany) on a Ficoll-Paque gradient (Merck, Darmstadt, Germany). Leukocytes were cultured overnight at 37 °C, 5% $CO_2$ in a cell culture flask in RPMI (Roswell Park Memorial Institute) 1640 medium (Merck, Darmstadt, Germany) supplemented with 10% fetal calf serum (Fisher Scientific, Waltham, Massachusetts, USA) and 1% Penicillin/Streptomycin (Sigma-Aldrich, St. Louis, MO, United States).

**Isolation of murine monocytes**. Murine monocytes were isolated from the spleen. Spleen tissue of sacrificed mice was squeezed through a 40 μm mesh filter and the filter was subsequently washed with PBS (Sigma Aldrich Co., St. Louis, MO, USA). The obtained cell suspension was processed through differential gradient centrifugation in Ficoll gradient (922 × g, 18 min). The resulting white intermediate layer was resuspended in PBS (Sigma Aldrich Co., St. Louis, MO, USA) and centrifuged (535 × g, 10 min) again. To lyse erythrocytes, the obtained cell pellet was resuspended in distilled water for 30 s. The addition of PBS (Sigma Aldrich Co., St. Louis, MO, USA) and centrifugation (535 × g, 10 min) was followed by adhesion depletion on a plastic surface. Monocytes were cultured in RPMI-1640 medium supplemented with 10% fetal calf serum, 100 U/ml penicillin, 100 μg/ml streptomycin, and 2 mM L-glutamine at 37 °C and 5% $CO_2$ in a humidified atmosphere. Non-adherent cells were removed by gentle washing with PBS on the following day, and the remaining adherent cells were harvested.

**Monocyte migration**. Migration of monocytes toward RPS or APS was performed with a modified Boyden chamber as described previously[56]. In short, isolated monocytes were loaded onto the upper chamber. Control medium, resting (RPS), activated (APS), or MCP-1 was loaded in the lower chamber. After incubation for 4 h at 37 °C, the membrane was fixed with 100% ethanol and stained with May-Gruenwald/Giemsa. The membrane was mounted on glass slides, and five randomly selected images were taken (Nikon Eclipse Ni-U, ×20 objective). The number of migrated cells was counted for each well in several microscopic fields per condition using ImageJ software to quantify the cell count (ImageJ, National Institutes of Health, USA)[57].

**Transient monocyte adhesion**. Washed murine platelets were adjusted to a cell count of $4 \times 10^7$ cells/ml in Tyrode's buffer (pH 7.4; supplemented with 1 mM $CaCl_2$). The platelets were activated with 1 μg/ml CRP-XL and incubated on fibrinogen-coated (100 μg/ml) coverslips for 30 min at 37 °C. Isolated human monocytes were adjusted to a cell count of $1 \times 10^6$ cells/ml in RPMI 1640 medium (supplemented with 10% fetal calf serum) and stained with 1 μM $DiOC_6$ for 10 min at 37 °C. The platelet-coated coverslips were mounted to a transparent flow chamber and perfused with the pre-stained monocytes for 5 min. Experiments were performed at shear rates of 15 ml/h. Four 40 s long videos (1 frame/sec) of randomly selected areas were taken (Nikon Eclipse Ti2-A, ×20 objective). The images were analyzed with the NIS-Elements AR software (Nikon, Japan)[58].

Washed murine platelets were adjusted to a cell count of $1 \times 10^8$ cells/ml in Tyrode´s buffer (pH 7.4 supplemented with 1 mM $CaCl_2$). The platelets were seeded in triplicates for all conditions onto a 96-well culture plate and incubated for 1 h at 37 °C. Afterward, the monolayers were washed with Tyrode´s buffer. The platelets were activated with 5 μg/ml CRP-XL or 0.1 U/ml thrombin for 20 min at room temperature. Isolated murine monocytes were harvested and adjusted to a cell count of $2 \times 10^4$ cells/ml. The monocytes were added to each well and incubated for 1 h at room temperature. Then, wells were washed three times with PBS and five images of randomly selected areas were taken (Nikon Eclipse Ti2-A, ×20 objective). The number of adherent monocytes was counted for each well in several microscopic fields per condition using ImageJ software to quantify the cell count (ImageJ, National Institutes of Health, USA).

**Platelet aggregometry**. Human platelet aggregation was assessed by impedance platelet aggregometry according to standard procedures. ACKR3 agonists and collagen were used as indicated. The light transmittance aggregometry was performed for ex vivo experiments according to the standard protocol[53].

**Flow cytometry**. Flow cytometry experiments of blood were performed as described before[23]. In brief, relevant fluorophore-labeled antibodies together with platelet-stimulating agonists at indicated concentrations were added to whole blood or platelet-rich plasma samples diluted with PBS supplemented with $CaCl_2$ and $MgCl_2$. After incubation, samples were analyzed on a FACS-Calibur flow cytometer. Mean fluorescence intensity (MFI) was used as a quantitative measurement of platelet protein surface expression.

Immunophenotyping of spleen obtained from MI-operated mice was performed. The spleen samples were ground against a 30-μm cell strainer then underwent erythrocyte lysis. All samples were stained for 15 min at room temperature with zombie aqua fixable viability dye (BioLegend, San Diego, USA) for live/dead discrimination. Samples were then washed and resuspended in FACS buffer (PBS containing 1% BSA, 0.1% sodium azide, and 1 mM EDTA). Surface staining was performed in the presence of FC-blocking antibody (anti-CD16/CD32, clone 2.4G2, BD Pharmingen). For intracellular cytokine staining of splenocytes, samples were restimulated with Cell Stimulation Cocktail plus protein transport inhibitors (eBioscience™) for 3 h in vitro prior to staining. The complete list of commercially available antibody clones that were used is provided in Supplementary Table 1. Flow cytometry measurements were performed using an Attune-NxT machine (Thermo Scientific, Darmstadt, Germany), and the data were analyzed with the FlowJo software.

Bone marrow cells were harvested from femurs and tibias. Total cell numbers were determined by trypan blue exclusion. Before staining, cells were incubated for 15 min at 4 °C with hybridoma supernatant from 2.4G2 cell line producing anti-FcgRII/III mAb. For flow cytometry, staining with monoclonal Abs for 20 min at 4 °C (a list of all antibodies used is shown in Supplementary Table 1) was performed in FACS buffer (PBS containing 1% FBS and 0.09% $NaN_3$ (Sigma-Aldrich) and 2 mM EDTA (Merck)). Dead cells were excluded using Live/dead blue (Invitrogen) according to the manufacturer's instruction. At least 1xs106 cells were acquired on an Aurora spectral flow cytometer (Cytek) and further analyzed using the OMIQ Data Science Platform.

**Carotid ligation and in vivo thrombus formation**. The common carotid artery of 8–10-week-old mice ($Ackr3^{fl/fl}$ and $Ackr3^{-/-}$) was injured by ligation as described previously[23]. Mice were anesthetized by injection of midazolam (5 mg/kg body weight), medetomidine (0.5 mg/kg body weight), and fentanyl (0.05 mg/kg body weight). After incision, both carotid arteries were exteriorized and ligated with an 7-0 threat for 5 min to induce a localized endothelial injury and denudation. Platelet accumulation at the site of carotid injury was monitored for 20 min with an intravital microscope (Nikon, Japan) using a 10x objective. Platelets were visualized by injecting an anti-GPIbβ-FITC monoclonal antibody. Digital images were recorded, and the samples were analyzed with the Nikon NIS-Elements AR software (Nikon, Japan)[20]. In the case of agonist treatment, the animals were injected with 100 μl solution, containing either 300 μg agonist or a vehicle control, 1 h prior to the carotid artery occlusion.

**Stroke model (tMCAO)**. All stroke experiments were performed as described before[59]. Mice were anesthetized intraperitoneally (100 μg/10 g body weight) using a combination of fentanyl (0.05 mg/kg), medetomidin (0.5 mg/kg), and midazolam (5 mg/kg). Focal cerebral ischemia was prompted by tMCAO using the intraluminal filament technique as described[59]. Following a midline skin incision in the neck, the proximal common carotid artery and the external carotid artery were ligated, and standardized silicon rubber-coated 6.0 nylon monofilament was inserted and advanced via the right internal carotid artery to occlude the origin of the middle cerebral artery. Occlusion and reperfusion were verified by laser-doppler-flowmetry (LDF). The reperfusion period was sustained for 24 h or 7 days under buprenorphine analgesia (0.1 mg/kg body weight, administered every 8 h), where after the mice were sacrificed by cervical dislocation under isoflurane anesthesia (5% v/v). The brain was immediately removed and fixated. The in situ Cell Death Detection Kit, TMR red was used to analyze for apoptosis after 24 h I/R according to the manufacturer's instructions.

**Myocardial infarction I/R model**. All myocardial infarction experiments were performed as described before[60]. Mice were anesthetized by injection of midazolame (5 mg/kg body weight), medetomidine (0.5 mg/kg body weight), and fentanyl (0.05 mg/kg body weight). The occlusion of the left anterior descending artery (LAD) and thereby ischemia was maintained for 30 min. The reperfusion period was sustained for 24 h or 28 days under buprenorphine analgesia (0.1 mg/kg body weight, administered every 8 h) during the first three days after I/R. Baseline ultrasound examinations and imaging of the heart were performed prior to the I/R. In the case of the 24 h reperfusion experiments, a second ultrasound analysis was performed prior to the LAD occlusion and shortly before the sacrifice of the mouse. During the 28 days reperfusion period ultrasounds examinations and imaging of the heart were performed 7 days post I/R and at day 28. Strain analysis was performed offline using VevoStrain Software inside VisualSonics VevoLab 3.0.5 (FUJIFILM VisualSonics Inc., Toronto, Canada). Polar plots of peak displacement were created by exporting Strain Analysis Data into a custom in-house Matlab script (MathWorks Inc., Natick, MA).

After the reperfusion period, the mice were sacrificed by cervical dislocation under isoflurane anesthesia (5%v/v), and blood samples were taken. The heart and spleen were extracted after the re-ligation of the LAD and prepared for flow cytometry analysis or the mice were perfused with Evans blue dye solution and the heart was extracted for histological analysis. The heart was divided from epical to basal into six sections of equal thickness. The sections were counterstained with 1%

TTC dye, fixated with 4% formaldehyde, and imaged (ViewPix 700, biobtep, Burkhardtsdorf, Germany). ImageJ software was used to analyze the Evans Blue/TTC staining (ImageJ, National Institutes of Health, USA). For hematoxylin and eosin stainings (H&E), de-paraffinized 5 μm sections were stained in Hematoxylin (Sigma-Aldrich) for 5 min, washed for 15 min under tap water, and consequently stained in Eosin (Carl Roth) for 3 min. Fibrosis staining was performed using Picrosiriusred staining with 1.2% Picrosiriusred (Merck) for 60 min.

In the case of agonist treatment, the animals were injected with 100 μl solution, containing either 300 μg agonist or vehicle control, 1 h prior to the LAD occlusion.

**Immunofluorescence.** Platelets were isolated and fixed in 8-well chamber slides (Ibidi, Gräfelfing, Germany). Murine heart or brain sections from mice obtained after ischemic tissues were embedded into paraffin, and cut into 3–5 μm sections using a microtome (RM 2045, Leica Mircosystems, Wetzlar, Germany). The slices were treated with Roticlear (Roth, Karlsruhe, Germany) and rehydrated in a decreasing alcohol series (100% to A. dest). Afterward, the samples were cooked in citrate buffer and washed with PBS. The samples were blocked with normal donkey serum and BSA (EMD Millipore, Burlington, Massachusetts, USA). Finally, samples were stained with various primary antibodies and AlexaFluor-labeled secondary antibody as recommended by the distributor (Supplementary Table 1). DAPI staining was applied where necessary. Confocal images of the infarction area within the heart and brain tissues were obtained using a Zeiss LSM 5 EXCITER confocal laser scanning microscope with an A-plan ×40/0.65 objective (Carl Zeiss, Jena, Germany). Platelet samples images were taken with a Nikon Eclipse Ti2-A microscope (×100 DIC oil objective). 2D deconvolution was applied to the platelet sample images using the NIS-Elements AR software (Nikon, Japan).

**Immunohistochemistry.** Immunohistochemical staining for various antibodies was undertaken using HRP-DAB staining systems (R&D Systems/BioTechne, USA) according to the manufacturer's protocol. Three images of the infarction area were taken per section (Nikon Eclipse Ti2-A, ×20 objective). The number of DAB-positive cells was manually counted for each image using ImageJ software (ImageJ, National Institutes of Health, USA). In the case of the more diffuse IL-1beta and TNFalpha staining of heart tissues, a Histoscore was used to rate the DAB signal in a double-blind procedure.

**NanoString® nCounter assay.** Total RNA was extracted from 6 to 15 sections of formalin-fixed paraffin-embedded tissue sections using the RNeasy FFPE Kit (Qiagen, Hilden, Germany) according to the manufacturer's instructions. RNA yield and purity were assessed using the NanoDrop ND-1000 spectrophotometer (NanoDrop Technologies, Rockland, USA). mRNA expression of 254 genes was measured with the NanoString nCounter Analysis System (NanoString Technologies, Seattle, USA) using 100 ng of total RNA. A custom nCounter CodeSet and three reference genes (GAPDH, HPRT1, RPL19) were hybridized to total RNA for 18 h at 65 °C and nCounter Prep Station loading as well as expression quantification with the nCounter Digital Analyzer was performed as recommended by the manufacturer. The expression data were analyzed utilizing the NanoString nSolver Analysis Software v3.0. Quality control of the data was performed using the default settings within the nSolver software and by analyzing the positive and negative control, housekeeper and total (excluding controls) counts as well as the binding densities in each sample. Agglomerative cluster heat map analysis using the Pearson Correlation distance measure was performed with the nSolver software according to the manufacturer's instructions. A complete list of genes is supplied in Supplementary Table 3.

**Statistics.** Graphs for murine experiments were created with the GraphPad Prism software (GraphPad Software, Inc., La Jolla, CA, USA) and represent biological replicates. Data are presented as dot plots with mean ± S.E.M. or mean ± S.D. All data were tested for significance using GraphPad Prism software (GraphPad Software, Inc., La Jolla, CA, USA) setting statistical significance at $p < 0.05$ for an unpaired two-tailed Student's $t$ test or a Mann–Whitney $U$-test (nonparametric distribution) to compare two sets of data, with a 95% confidence interval. A one-way analysis of variance (ANOVA) using Tukey post hoc test was used to compare multiple data sets. Results with values of $p < 0.05$ were considered statistically significant. Results from the NanoString nCounter Analysis were submitted to one-way ANOVA using JMP® version 14.2.0 to identify mRNA varying significantly ($p < 0.05$). Fold change of $Ackr3^{-/-}$ and $Ackr3^{fl/fl}$ was calculated and log2-transformed using means of 6 individuals per genotype in myocardial infarction assays and 4 individuals per genotype in brain section experiments. Classification of Nanostring data analysis was performed using KEGG pathway enrichment (Kyoto Encyclopedia of Genes and Genomes database). Heat maps were created using hierarchical clustering with plotted z-scores. Color gradient legends display significantly varying mRNA levels.

**Reporting summary.** Further information on research design is available in the Nature Research Reporting Summary linked to this article.

## Data availability

Detailed demographic characterizations of patients are available upon reasonable request. Source data are provided with this paper.

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

## Acknowledgements

We thankfully acknowledge the work of Birgit Fehrenbacher for her expert technical assistance in transmission electron microscopy. We thank Kristine Schmidt for proof-reading the manuscript. We thank Sarah Gekeler for technical assistance with the experiments. This project was supported by the Deutsche Forschungsgemeinschaft (DFG, German Research Foundation) KFO-274—project number 190538538 grant to M.G. (clinical aspects), by the Deutsche Forschungsgemeinschaft (DFG, German Research Foundation)—project number 374031971—TRR 240 grant to M.G. (molecular aspects) and project number BO3786/3-1 grant to O.B.

## Author contributions

M. Gawaz, A.-K.R., J.S., K.K., M.-C.M., S.v.U.-S., S.G., M.C., and M.Z. designed and planned the experiments. D.R. was responsible for patient data collection and measurements. J.S. and A.-K.R. phenotype characterization of the *Ackr3⁻/⁻* mice. J.S. and M.Z. performed the ischemia and reperfusion of the heart, including ultrasound. A.B. performed platelet spreading experiments. J.S., M.H., and K.S. performed the immunohistochemistry staining of heart sections. J.S. prepared samples for NanoString analysis. J.S., M.C., and K.K. performed the flow cytometry experiments. J.S. and K.K. performed the flow chamber experiments. P.S. and P.M. performed mass spectrometry analysis. A.-K.R. and M.K. prepared platelets for spreading, SICM, and electron microscopy. H.v.E. and T.E.S. performed SICM measurements. O.B. and S.G. performed calcium measurements and ATP release. M.B. and M.-C.M. performed carotid artery ligation experiments. A.-K.R., J.S., K.K., T. H., C.L., T.D., and D.R. analyzed the data including statistical analysis. J.S. and V.D. performed the migration assays. S.v.U.-S. and A.-K.R. performed monocyte adhesion experiments. J.S. performed ELISA. C.K., S.M., F.L., M.C.B., and E.K. performed the tMCAO. D.A., S.A., M. Günter, G.R. performed multicolor flow cytometry analysis. S.L. performed agonist analysis. All the authors participated in the interpretation and discussion of the results. A.-K.R., J.S., K.K., T.H., M.S., C.L., T.D., D.H., D.R., K.A.L.M., D.M.H., E.K., R.S., M.L., H.F.L., and M. Gawaz analyzed the data including statistical analysis. A.-K.R. figure design. M. Gawaz, A.-K.R., and T.C. wrote the article.

## Funding

## Competing interests

The authors declare no competing interests.
