## [Peer Review File · Nature Communications]

ACKR3 regulates platelet activation and ischemia-reperfusion tissue injuryREVIEWER COMMENTS

Reviewer #1 (Remarks to the Author):

This study investigated the role of platelet CXCR7 in ischemic/reperfusion injury in the heart and brain. First, they obtained data from the cardiac clinic that showed a correlation between higher levels of platelet CXCR7 and a better prognosis after myocardial infarction. Second, they generated a CXCR7 knockout mouse line allowing platelet specific deletion of the gene. Third, they showed that the CXCR7 knockout mice had aggravated ischemic/reperfusion injury to the heart and brain, respectively, after artery ligation. Finally, they suggested that the CXCR7 deficiency-afforded injury aggravation is due to platelet aggregation, and increased thrombus formation and inflammation.

The scientific topic is very interesting, and the working hypotheses to be tested in the current study are also innovative and promising. However, the results presented in the manuscript are quite superficial, missing important outcome assessments of the models, and lacking in depth mechanistic analysis.

Specific points:

1. The authors characterized the megakaryocyte/platelet-specific CXCR7 knockout mice. Most results were from isolated platelets (Figure 2 & 3), focusing on platelets themselves. However, the likely impact of platelet-deficiency in CXCR7 on other immune cells was not studied. Since the knockout mice are not inducible one, the chronic effect of CXCR7 deficiency on the development and functional alteration of bone marrow immune cells *in vivo*, such as monocytes and lymphocytes, could be substantial. As the authors are aware of and indeed mentioned in the manuscript, platelets release a number of cytokines and chemokines that could substantially influence many cell types. Many alterations in the functional properties of circulating immune cells critically contribute to both CAD and CVD.
2. The authors showed that deficiency in platelet CXCR7 aggravates myocardial injury and inflammation following I/R (Figure 5). The data presented are quite superficial and missing any functional outcomes. The histological data suggest that myocardial infarction is increased and inflammatory cell infiltration aggravated in heart tissues after I/R. However, the mechanism underlying the increased immune infiltration was not studied. Moreover, immunostaining was used to characterize immune cell infiltration; the specificity of the various antibodies was not verified. A much better methodology for quantitative analysis would be using flow cytometry.
3. The authors showed that deficiency in platelet CXCR7 aggravates brain injury and inflammation following ischemia/reperfusion (Figure 6). Again, the data are extremely superficial. Only acute stroke outcomes were assessed (24 hours). No quality controls for the MCAO model were done. In current stroke research, one needs to assess functional outcomes for several weeks, as what happens in the acute stage of stroke (24-72 hours after MCAO) does not always dictate the long-term stroke outcomes. In a stroke clinic, the standard outcome assessment is at 90 days after the onset of stroke.
4. What is the mechanism by which platelet CXCR7 deficiency aggravates ischemic brain injury?
5. In Figure 7, the authors demonstrated the beneficial effect of CXCR7 agonists. A problem with these results is that CXCR7 is not only expressed in platelet but also in other cell types. It's highly likely that the effect of CXCR7 agonists *in vivo* may be derived from their actions on all cell types in the heart and brain that express CXCR7.

Reviewer #2 (Remarks to the Author):

The manuscript of Rohlfing and coworkers describe an interesting and novel role of the chemokine receptor CXCR7 on platelet function and its consequences for thrombo-inflammation and ischaemic tissue injury. The authors generated and characterized a new mouse strain deficient in platelet CXCR7 and show that loss of CXCR7 results in enhanced platelet activation and platelet-dependent thrombus formation in vitro and in vivo. Rohlfing et al. then analysed the effect of CXCR7 deficiency in platelets on ischaemia/reperfusion injury in myocardium and brain. Further, the authors show that CXCR7 agonists reduce platelet-dependent thrombus formation in vitro and in vivo and result in reduction of myocardial ischaemia/reperfusion injury in mice. Finally, Rohlfing et al. show in patients with coronary artery disease that surface expression of CXCR7 is associated with clinical prognosis.

This is an interesting basic science manuscript with direct translational relevance in a highly topical clinical area. The authors provide an extensive array of experimental evidence, including knockout experiments, pharmacological and clinical data. The data are very well presented and well discussed.

To improve the manuscript the authors should address the following points of critique:

General points to address:

1. The authors show that loss of platelet CXCR7 enhances myocardial inflammation 24 h after ischaemia/reperfusion. The authors should also analyse a longer time interval e.g. 28 days after transient ischaemia/reperfusion to characterize whether this short term effect has long-term consequences for myocardial function and inflammation. Further, it would be of interest to see the long-term effects on myocardial fibrosis and neovascularization.
2. The authors mention that loss of platelet CXCR7 has proinflammatory effects. This is an interesting aspect, however the authors miss to provide data e.g. on markers of systemic inflammation following ischaemia/reperfusion in their genetic mouse model.
3. The authors show that loss of CXCR7 results in enhanced platelet degranulation. It would be of interest to characterize the release of inflammatory mediators in platelets derived from CXCR7-deficient and WT mice.
4. The authors mention that CXCR7 is a novel inhibitory receptor on platelets. This is an interesting aspect, however, the authors should discuss the potential relevance of their findings for ischaemic tissue injury.
5. The authors should introduce the CXCR7 agonists VUF11207 and C10. They discuss the interesting pharmacological approach to activate CXCR7 therapeutically. The authors should further elaborate on this important translational perspective, including the discussion on the potential impact of agonist-induced internalisation of the receptor.

Specific points to address:

1. The clinical study present data that support the concept of using surface expression of receptors on platelets for clinical prognosis. The authors should provide some further data on potential cofounders that are associated with platelet CXCR7 surface expression (e.g. co-medication, LV-function, cardiovascular risk factors etc.). Also in the abstract and conclusions the authors state there is an association with clinical prognosis. Rather than providing this unclear statement, the authors should state the actual association.
2. Figure 2: Endothelial deficiency in CXCR7 has significant impact of aortic malformation. The authors should provide further gross anatomical phenotyping of the aorta and greater arteries. Do the platelet-specific CXCR7-KO mice have obvious malformations (e.g. aortic branch, valve, heart weight)?
3. The authors show enhanced P-selectin expression in CXCR7-KO mice. It would be of interest to see data derived from the TEM studies in terms of numbers of alpha-granules and dense-granules

in CXCR7-deficient mice compared to WT mice. Also CD62P immunostaining of single platelets would help to exclude that enhanced P-selectin expression is due to a higher number of alpha-granules.

4. Figure 2: In the adhesion experiments performed by the authors, is spreading different between platelets derived from CXCR7-deficient mice compared to WT mice?

5. Can the authors provide data on systemic inflammation following transient myocardial or brain ischaemia in CXCR7-deficient mice compared to WT mice.

6. Figure 3: The authors show enhanced degranulation in CXCR7-deficient platelets. Is this specific for CRP as agonist? What about other platelet agonists, such as ADP or thrombin?

7. Figure 7: The authors should further characterise the effect on CXCR7 agonists on platelet function. Do CXCR7 modulate Ca²⁺ signalling in platelets? What about platelet adhesion and spreading?

8. CXCR7 has been suggested as a survival receptor. Apoptosis in platelets has recently attracted major attention. The authors should provide data on the potential role of CXCR7 on platelet apoptosis.

9. Although the manuscript overall seems carefully prepared, it needs further editing. There are a few spelling (line 341: mic instead of mice) and grammar mistakes throughout the manuscript.

Reviewer #3 (Remarks to the Author):

The group of M. Gawaz has previously reported the expression of ACKR3 on human platelets. Here they show that elevated expression of ACKR3 on platelets is a positive prognostic marker in coronary artery disease. In a new mouse model, where ACKR3 is selectively deleted in megakaryocytes using a cre recombinase under the Pf4 promoter, they recapitulate the findings made in humans showing aggravated heart and brain tissue damage after reperfusion in mice with platelets deficient in ACKR3.

The mouse model is essential for the study, but is poorly characterized. The breeding strategy is clear, but genotyping and ACKR3 deletion remain doubtful. Although megakaryocytes are rare in circulation and flushed bone marrow samples, evidence of the receptor in the cells should be documented with the available ACKR3-GFP reporter. Moreover, GFP expression in platelets could be shown. Previous studies, however, have shown that ACKR3 expression in leukocytes is solely restricted to the B cell compartment.

Alternatively, real-time PCR on the transcripts for ACKR3 and internal controls are needed to confirm expression in megakaryocytes.

A FACS analysis with complete gating strategy from human and mouse blood is missing, starting from whole blood or platelet enriched plasma.

Figure 2 needs additional explanations. B: the comparison of "CXCR7^{fl/fl}" and "CXCR7^{-/-}" was performed how? Which cells? Whole blood? What animals? The strong 450 bp band in CXCR7^{-/-} animals suggests some enrichment over the 200 bp band. What does the 227bp band corresponds to? The used primers should give a different amplicon according to GeneBank

Are the ACKR3^{fl/fl} mice homozygous for Pf4-cre? Can it be excluded that the observed effects are due to cre expression?

2C: the western blot needs better controls. The authors reported previously a diffuse 55kD for human ACKR3, here the sharp 42kD contrasts and is not convincing. Trustworthy commercial antibodies (e.g 7mantibodies.com) indicate similar Mw for human and mouse ACKR3 (~55kD) with a larger Mw than reported here. In addition, entire lanes would be more preferred. Were the samples boiled before loading?

Figure 3: Stimulation with CRP induces larger P-selectin expression on ACKR3⁺ cells (from intracellular granules), but no enhanced ATP secretion. What about CXCL12? This is critical since CXCL12 is a ligand of ACKR3 and could have a feedback effect (e.g. measurements in the presence of AMD 3100). This is also relevant for the statement on line 175 were apparently only Ly6G

myelocytes, but not lymphocytes were differentially attracted depending on platelet ACKR3 expression. Continuing line 178 the role of CXCR4 positive neutrophils, which play a critical role during reperfusion, is not addressed.

Figure 4: What is the active compound in APS-stimulated migration? Is it AMD3100 sensitive? Is the phenotype of migrated monocytes different from the input? Are there differences in the phenotype of migrated monocytes depending on the presence of ACKR3 on the platelets (APS)?

Figure 7a: the authors analyzed surface expression of P-selectin in platelets (wt and ko) stimulated with ACKR3 agonists. The authors should show corresponding flow cytometry data. Stimulation of platelets should also be performed with endogenous ligands (CXCL11, CXCL12 and MIF).

The conclusion that ACKR3 is a repressing surface receptor on platelets is conceivable, but in the lack of any presented evidence, the statement is circumstantial and alternative mechanisms should be considered.

Minor: line 268 the authors state that previous names for CXCR7 were RDC1 and ACKR3. This is simply wrong. The IUPHAR nomenclature, that should be used throughout the paper, is ACKR3 and has replaced CXCR7, RDC1 and CMKOR1.

Reviewer #4 (Remarks to the Author):

The paper of Roling et al is interesting paper with substantial data and experiments showing that platelet CXCR7 regulates thrombosis and inflammation.

I have several remarks or suggestions

1 Despite CXCR7 is associated with thrombosis, the high and low platelet CXCR7 are associated with all mortality and not with thrombotic events like stroke or MI. Since only 36 events occur I understand this choice, however, it would be good to add # thrombotic events to the suppl. data and a statement that association of CXCR7 with outcome should be taken with caution. The TRAP data are convincing.

2 Do the authors have MIF data as the ligand of CXCR7?

3 In the results the introduction of I/R injury starts with the following sentence:

61 The remarkable increase in CXCR7 on circulating platelets of patients with acute
162 myocardial infarction¹⁴ and the impact on prognosis (Figure 1) in patients with CAD
163 implies that platelet-CXCR7 has a functional role in ischemia and reperfusion (I/R)
164 injury.

This sentence is very confusing as I do not find baseline levels before ACS in fig 1 so this is not an increase

Secondly the impact on prognosis implies a functional role in IR Injury. So a future all mortality event implies an impact on acute damage after an infarct This reasoning is very unclear and needs explanation or rephrasing. Is the paper not mainly on CXCR7 on the crossroads between coagulation and inflammation and is the IR Injury models not merely an example of this.

4 The nano string profiling shows clear up AND down regulation of inflammatory genes in both brain and heart. This includes pro and anti-inflammatory genes which can be expected and it results in influx of more inflammatory cells suggesting that the up regulation of inflammatory genes is towards the pro-inflammatory site. This is hard to determine from these data now as when more chemokine are stored in the cell the mRNA are often down and after release the respective mRNA goes up. So what is the net effect on protein level Is this also towards pro-inflammatory genes so is there indeed an upregulation from eg TNFalpha or IL1B in the tissue (or blood)

Nature Communications NCOMMS-20-50883**Title: ACKR3 regulates platelet activation and ischemia - reperfusion tissue injury****Corresponding Author: Meinrad Gawaz****Response to the Reviewers**

We thank Reviewers for their positive assessment of our manuscript. In the following we answer their question step by step. Please note, that as suggested by Reviewer 3 and in correspondence with the IUPHAR nomenclature **ACKR3 instead of CXCR7** is now used throughout the manuscript and this letter.

Comments of Reviewer #1:

This study investigated the role of platelet CXCR7 in ischemic/reperfusion injury in the heart and brain. First, they obtained data from the cardiac clinic that showed a correlation between higher levels of platelet CXCR7 and a better prognosis after myocardial infarction. Second, they generated a CXCR7 knockout mouse line allowing platelet specific deletion of the gene. Third, they showed that the CXCR7 knockout mice had aggravated ischemic/reperfusion injury to the heart and brain, respectively, after artery ligation. Finally, they suggested that the CXCR7 deficiency-afforded injury aggravation is due to platelet aggregation, and increased thrombus formation and inflammation.

The scientific topic is very interesting, and the working hypotheses to be tested in the current study are also innovative and promising. However, the results presented in the manuscript are quite superficial, missing important outcome assessments of the models, and lacking in depth mechanistic analysis.

Response to Reviewer #1:

We thank the reviewer for his/her favorable and constructive comments. In the following we would like to address the comments and concerns point-by-point.

1. The authors characterized the megakaryocyte/platelet-specific CXCR7 knockout mice. Most results were from isolated platelets (Figure 2 & 3), focusing on platelets themselves. However, the likely impact of platelet-deficiency in CXCR7 on other immune cells was not studied. Since the knockout mice are not inducible one, the chronic effect of CXCR7 deficiency on the development and functional alteration of bone marrow immune cells in vivo, such as monocytes and lymphocytes, could be substantial. As the authors are aware of and indeed mentioned in the manuscript, platelets release a number of cytokines and chemokines that could substantially influence many cell types. Many alterations in the functional properties of circulating immune cells critically contribute to both CAD and CVD.

Response to Reviewer #1:

We thank the reviewer for this valuable and justified comments. In the previous manuscript we provided data derived from immunohistochemistry that in *Ackr3*^{-/-} mice (we renamed the mouse strains according to suggestions of reviewer 3, *Ackr3* = *Cxcr7*), myocardial cell infiltration was substantially enhanced in *Ackr3*^{-/-} versus *Ackr3*^{fl/fl} mice. As suggested we performed additional studies to characterize bone marrow immune cells derived from bone marrow and spleen by multi-panel flow

cytometry. We found that the number of immune cells derived from bone marrow was not altered (neutrophils and neutrophil progenitors, monocytes and monocyte progenitors, B cells, NK T-cells, CD8⁺ T-cells, CD4⁺ T cells, CD4⁺/CD25⁺ T-cells, NK cells, CD64⁺ macrophages, cDC1, cDC2 and pDCs) (**supplemental Figure 2/3**). In addition, we measured the receptor expression of bone marrow derived immune cells. No significant differences were observed for the tested surface markers on various cell types between *Ackr3*^{-/-} and *Ackr3*^{fl/fl} mice (**supplemental Figure 2/3**). Thus, platelet-specific deletion of ACKR3 does not have an effect on development and number of bone marrow immune cells. Furthermore, we analyzed immune cells of the spleen (Ly6G⁺ neutrophils, Ly6C⁺/CD11b⁺ monocyte-derived macrophages). Again, no significant differences were observed in *Ackr3*^{-/-} versus *Ackr3*^{fl/fl} mice (**supplemental Figure 2/3**).

Although we did not provide data that platelet-ACKR3-deficiency alters the function of cells that infiltrate the myocardium following I/R, we favor the conclusion that enhanced platelet secretion of inflammatory mediators of ACKR3-deficient platelets is the prominent cause of enhanced myocardial inflammation and cell infiltration especially of Ly6G⁺ cells (**please see Figure 5c**).

Supplemental Figure 2: Quantification of immune cells derived in bone marrow and spleen from *Ackr3*^{-/-} and *Ackr3*^{fl/fl} mice. a/b Immunes cell count in percent of total cells within the bone marrow comparing *Ackr3*^{-/-} and *Ackr3*^{fl/fl} mice. **c** CD11b expression on immune cells. **d** MHCII expression on immune cells. **e** CD25 expression on immune cells. **f** CD3 expression on immune cells. **g** CD45R/B220 and CD19 expression on B cells. **h** Immune cell count per mg spleen tissue in *Ackr3*^{-/-} and *Ackr3*^{fl/fl} mice. Plotted: Mean±SEM; n > 7; Statistics: Student's t-test; n.s. = not significant, * p < 0.05.

Supplemental Figure 3: Gating strategy for the quantification of immune cells derived in bone marrow and spleen from *Ackr3*^{-/-} and *Ackr3*^{fl/fl} mice. a bone marrow b spleen.

Figure 5: Deficiency in platelet ACKR3 aggravates myocardial injury and inflammation following I/R. **a** Graphic display of the I/R procedure of the heart, including representative images of Evans-Blue and TTC staining of heart sections, (healthy tissue = blue, infarct = white, area at risk = red/white areas). Statistical comparison of infarction area [%] of left ventricle and area at risk [%] of left ventricle measured in *Ackr3*^{-/-} and *Ackr3*^{fl/fl} after I/R. Plotted: mean±S.D.; statistics: Student's t-test. n = 6. **b-d** Various comparisons of the infarct area of *Ackr3*^{-/-} and *Ackr3*^{fl/fl} after I/R. **b** Analysis of platelet migration into the infarct area. Representative fluorescence images of the infarct area of *Ackr3*^{-/-}Pf4Cre⁺ROSA animals expressing dTomato in all cells except for Pf4Cre⁺ platelets (endogenous GFP). Representative images (scale bar = 100 μm) and statistical analysis of CD42b specific DAB staining of infarct area. n ≥ 4. **c** Representative images (scale bar = 100 μm) and statistical analysis of Ly6G, MHCII, CD3 and B220 specific DAB staining of cell migration into infarct area. n ≥ 4. **d** Representative HE staining images (scale bar = 200 μm) and statistical analysis of the number of infiltrating cells/mm² into infarct area. n = 6. **e** Nanostring analysis of infarct area in *Ackr3*^{-/-} and *Ackr3*^{fl/fl} after I/R. Upper left Scatter plot of mRNA expression showing significantly adjusted p-values (p < 0.05, upregulation = red, downregulation = blue; displayed: log₂; n = 6). Upper right KEGG pathway enrichment analysis of significantly altered mRNAs (downregulation = blue, upregulation = red). The number of different KEGG genes of each group (n = 5) is displayed next to it. Semi-quantitative alteration of genes is presented using log₂-transformed fold change of mRNA. Lower left Volcano plot unveiling analysis of significance detecting quantitative changes in mRNA levels. p < 0.05 = dark blue. n = 6. Lower right Heat map analysis of significantly regulated mRNAs (Upregulated = red, downregulation = green; n=6). Color density legend on the right indicates plotted variation in z-scores of mRNA expression. **b-d** Plotted: mean±S.E.M.; statistics: Student's t-test; n.s. = not significant; * p < 0.05; ** p < 0.01.

2. The authors showed that deficiency in platelet CXCR7 aggravates myocardial injury and inflammation following I/R (Figure 5). The data presented are quite superficial and missing any functional outcomes. The histological data suggest that myocardial infarction is increased and inflammatory cell infiltration aggravated in heart tissues after I/R. However, the mechanism underlying the increased immune infiltration was not studied. Moreover, immunostaining was used to characterize immune cell infiltration; the specificity of the various antibodies was not verified. A much better methodology for quantitative analysis would be using flow cytometry.

Response to Reviewer #1:

We thank the reviewer for this valuable and justified comments. In our previous submitted manuscript, we described that 24h after reperfusion myocardial function (ejection fraction, fractional shortening) was reduced in both mouse strains equally, although inflammation and infarct size was enhanced in *Ackr3*^{-/-} mice (**please see Figure 5 and supplemental Figure 10**). The lack of short-term functional differences between *Ackr3*^{-/-} and *Ackr3*^{fl/fl} mice despite of differences in the area at risk might be due to myocardial stunning which is known to occur early (24 hours) after I/R.

As suggested, to further evaluate the functional outcome following I/R we performed additional experiments. I/R experiments were performed in *Ackr3*^{-/-} and *Ackr3*^{fl/fl} mice and the extent of myocardial infarction and function was analyzed after **28 days**. We found that the area of the infarcted myocardium was substantially enhanced in *Ackr3*^{-/-} versus *Ackr3*^{fl/fl} mice (**please see new Figure 6**). The extent of the infarcted myocardium has also functional consequences. In *Ackr3*^{-/-} mice myocardial function was significantly impaired and sustained over 28 days of I/R compared to *Ackr3*^{fl/fl} mice as verified by global strain echocardiography and injection fraction (**new Figure 6**). Interestingly, whereas in *Ackr3*^{fl/fl} mice myocardial function almost recovered following I/R, the loss of myocardial function was sustained in *Ackr3*^{-/-} mice. Loss of myocardial function in *Ackr3*^{-/-} was paralleled by a substantial enhanced myocardial fibrosis in *Ackr3*^{-/-} compared to *Ackr3*^{fl/fl} mice (**new Figure 6**). Speckle tracking analysis revealed a loss of global longitudinal function (**new Figure 6**).

The data of these comprehensive new experiments provide evidence that platelet-ACKR3 deficiency results in a sustained and irreversible loss of myocardial function and enhanced fibrosis, thus has an impact on functional outcome. These long-term analysis (28 days) combined with the enhanced myocardial inflammation early after reperfusion (24 hours) imply that enhanced platelet activation early in the time course of myocardial ischemia is associated with enhanced fibrosis and loss of myocardium in the long-term. Platelet ACKR3 seems to play a prominent role in enhanced platelet activation and triggers fibrosis development and thus disease progression.

We agree with the reviewer that at present we cannot defined further molecular mechanism responsible for the increase in myocardial inflammation observed in *Ackr3*^{-/-} and *Ackr3*^{fl/fl} mice. However, we describe a novel mechanism that hyperresponsiveness and increased degranulation of platelets (which is dependent on platelet ACKR3) determines the inflammatory and functional response in I/R myocardium. We are currently following strategies to address and define potential critical platelet-derived inflammatory mediators and responsive immune cells that are involved in our ischemia model. However, this requires a major amount of work and time including establishment of various novel mouse strains and is not foreseeable to be successfully achieved in a reasonable time frame. Thus, we would like to ask the reviewer to accept our data and conclusion as currently presented.

Figure 6: 30 min ischemia and 28-day reperfusion of the heart in *Ackr3*^{-/-} versus *Ackr3*^{fl/fl} mice. **A Schematic drawing of the time line and heart sectioning of the for long term functional outcome of MI and reperfusion. **b** Representative TTC/Evans blue staining of the heart section **S4** **b** Statistical analysis of the area at risk of the left ventricle in percent and the infarction of the area at risk in percent. Plotted: Mean±SD; n > 7; Statistics: Student's t-test; ns = not significant, **** p < 0.0001. **c** Representative pictures of Sirius Red staining of the fibrosis within the heart section 4 and statistical analysis of the fibrosis in the whole heart. Plotted: Mean±SEM; n > 7; Statistics: Student's t-test; ns = not significant, * p < 0.05. **d** Statistical analysis of the fibrosis separated by heart sections S1 to S4. Plotted: Mean±SEM; n > 7; Statistics: Student's t-test; ns = not significant, * p < 0.05. **e** Strain analysis of the heart in *Ackr3*^{-/-} versus *Ackr3*^{fl/fl} mice. Plotted: Mean±SEM; n > 7; Statistics: Student's t-test; ** p < 0.01, *** p < 0.001. **f** Statistical analysis of the ejection fraction of the heart over time in *Ackr3*^{-/-} versus *Ackr3*^{fl/fl} mice. **g** Representative images of CD31 staining and statistical analysis CD31+ vessels in *Ackr3*^{-/-} versus *Ackr3*^{fl/fl} mice. Plotted: Mean±SEM; n > 7; Statistics: Student's t-test; *** p < 0.001.**

As suggested, we have added our IgG controls for the immunostainings to verify the specificity of our antibodies (**please see supplemental Figure 11**). The stainings have been performed in cooperation with Prof. Heikenwalder (DKFZ, Heidelberg, Germany). His group previously published the staining procedure and similar stainings in per-reviewed journals.¹⁻³

rat IgG Control

rabbit IgG Control

Supplemental Figure 11: IgG Control for the antibodies used in the DAB staining in Figure 5 of the manuscript.

As recommended we performed additional analysis of myocardial infiltration of immune cells following I/R at time 24 hours. Ischemic myocardial tissue derived from *Ackr3*^{-/-} and *Ackr3*^{fl/fl} mice was digested and analyzed by flow cytometry as described.⁴ We found that the number (count/mg) of immune cells was not different between the tested mouse groups (CD64, CD11b, CCR2, MHCII, CD11b+) (**please see Figure I and II for Reviewer 1**). Also, the numbers of Ly6G⁺ monocytes were not significantly different in ischemic *Ackr3*^{-/-} versus *Ackr3*^{fl/fl} mice (**Figure I and II for Reviewer 1**). This seems to be somehow contrasting the observed localized increase in Ly6G⁺ cells as documented in our immunostainings (**please see Figure 5c**). At present we cannot easily explain these two observations. However, we observed a significant local accumulation and clustering of Ly6G⁺ cells especially within the area at risk. This may explain that localized Ly6G⁺ cell clustering will follow enhanced platelet activation and accumulation although the total cell count of infiltrating cells does not differ between the two mouse groups. Further, isolation of single separated immune cells from digested myocardium may not fully represent the total cell populations that are present within the inflamed tissue. In addition, we found similar enhanced infiltration of Ly6G⁺ cells in ischemic brain tissue, which is now shown in new immunostainings of brain sections from tMCOA mice (**please see new Figure 7c**).

In summary, it seems unlikely that deletion of ACKR3 in platelets results in changes of number and function of immune cells that may be responsible for the observed enhanced inflammatory response in *Ackr3*^{-/-} mice.

Figure I for Reviewer 1: Immune cell infiltrates in the heart after 30 min ischemia and 24 h reperfusion of the heart in *Ackr3*^{-/-} versus *Ackr3*^{fl/fl} mice. Plotted: Mean±SEM; n = 6; Statistics: Student's t-test; n.s. = not significant.

Figure II for Reviewer 1: Gating strategy for immune cell infiltrates in the heart after 30 min ischemia and 24 h reperfusion of the heart in *Ackr3*^{-/-} versus *Ackr3*^{fl/fl} mice.

Figure 7: Deficiency in platelet ACKR3 aggravates brain injury and inflammation following ischemia/reperfusion. C Immunostaining and statistical analysis of tMCAO brain sections. First panel: Ly6G+ staining, Second Panel: CD3+ staining, Third panel: GFAP+ staining representing activated astrocytes, Fourth panel: IBA+ staining. $n = 3$; Plotted: mean \pm S.E.M.; statistics: student's t-test; n.s. = not significant; * $p \geq 0.05$.

References:

1. Malehmir M et al, Platelet GPIIb α is a mediator and potential interventional target for NASH and subsequent liver cancer, *Nat Med.* 2019 Apr;25(4):641-655. doi: 10.1038/s41591-019-0379-5.
2. Dudek M et al, Auto-aggressive CXCR6+ CD8 T cells cause liver immune pathology in NASH. *Nature.* 2021 Apr;592(7854):444-449. doi: 10.1038/s41586-021-03233-8.
3. Pfister D et al, NASH limits anti-tumour surveillance in immunotherapy-treated HCC. *Nature.* 2021 Apr;592(7854):450-456. doi: 10.1038/s41586-021-03362-0.
4. Rieckmann M et al, Myocardial infarction triggers cardioprotective antigen-specific T helper cell responses. *J Clin Invest.* 2019 Aug 13;129(11):4922-4936. doi: 10.1172/JCI123859.

3. The authors showed that deficiency in platelet CXCR7 aggravates brain injury and inflammation following ischemia/reperfusion (Figure 6). Again, the data are extremely superficial. Only acute stroke outcomes were assessed (24 hours). No quality controls for the MCAO model were done. In current stroke research, one needs to assess functional outcomes for several weeks, as what happens in the acute stage of stroke (24-72 hours after MCAO) does not always dictate the long-term stroke outcomes. In a stroke clinic, the standard outcome assessment is at 90 days after the onset of stroke.

Response to Reviewer #1:

We agree with the reviewer that in terms of the clinical situation, the long-term clinical outcome is of significance as shown above for myocardial ischemia. The current stroke data shows that similar to the experiments of myocardial ischemia, loss of platelet-ACKR3 results in enhanced cerebral ischemia and tissue inflammation 24 hours after I/R. We agree that data on the long-term effects of cerebral ischemia would be of interest. However, these long-term experiments are not easily performed due to the high mortality and morbidity of the animals in the tMCAO model. Due to reservation of the regional board for animal experiments, regarding the even high mortality and the major strain for the animals expected from long-term tMCAO experiments, requires a major time frame with uncertain permission. In addition, the regional board for animal experiments (Regierungspräsidium Baden-Württemberg) has prolonged processing time on account of the ongoing Covid-19 pandemic. Thus, we would like to ask the reviewers for understanding not to have data in a reasonable time frame available to answer the reviewer's question. To elucidate and characterize further aspects of brain inflammation we performed new experiments as shown below.

4. What is the mechanism by which platelet CXCR7 deficiency aggravates ischemic brain injury?**Response to Reviewer #1:**

As shown in **Figure 3** loss of platelet-Ackr3 results in enhanced platelet activation (enhanced P-selectin surface expression and rise of intracellular Ca^{2+}). Activated platelets release inflammatory mediators such as chemokines and interact with circulating immune cells.⁵ Thus, the hyperreactivity of *Ackr3*^{-/-} platelets results in enhanced platelet-mediated inflammatory response as indicated by an increase migration response and co-aggregate formation primarily with Ly6G-positive cells (*see new Figure 4d*). Thus, it is tempting to speculate that an enhanced release reaction of ACKR3 deficient platelets results in enhanced chemotaxis and co-aggregate formation with preferentially Ly6G⁺ cells. As shown for myocardial injury our new immunostaining data of brain tissue shows an enhanced accumulation of Ly6G positive cells in ischemic compared to non-ischemic areas (*please see new Figure 7c*). Further, CD3⁺ cells were not different in ischemic versus non-ischemic areas similar to the findings in myocardial ischemia. Numbers of astrocytes and the neurons were not different (*please see new Figure 7c*)

Figure 4: *Ackr3*^{-/-} platelets promote thrombo-inflammation in vitro. d Statistical analysis of spontaneous co-aggregate formation of *Ackr3*^{fl/fl} and *Ackr3*^{-/-} platelets measured by flow cytometry analysis under unstimulated and stimulated conditions (1 µg/ml CRP-XL). Image stream images from Ly6G⁺/platelet aggregates. n = 7. Plotted: mean±S.E.M.; statistics: student's t-test; ns = not significant, * p < 0.05.

Figure 7: Deficiency in platelet ACKR3 aggravates brain injury and inflammation following ischemia/reperfusion. C Immunostaining and statistical analysis of tMCAO brain sections. First panel: Ly6G+ staining, Second Panel: CD3+ staining, Third panel: GFAP+ staining representing activated astrocytes, Fourth panel: IBA+ staining. $n = 3$; Plotted: mean \pm S.E.M.; statistics: student's t-test; n.s. = not significant; * $p \geq 0.05$.

References:

- Coppinger JA et al, Characterization of the proteins released from activated platelets leads to localization of novel platelet proteins in human atherosclerotic lesions. *Blood*. 2004 Mar 15;103(6):2096-104. doi: 10.1182/blood-2003-08-2804. Epub 2003 Nov 20. PMID: 14630798.

5. In Figure 7, the authors demonstrated the beneficial effect of CXCR7 agonists. A problem with these results is that CXCR7 is not only expressed in platelet but also in other cell types. It's highly likely that the effect of CXCR7 agonists *in vivo* may be derived from their actions on all cell types in the heart and brain that express CXCR7.

Response to Reviewer #1:

We fully agree with this comments that ACKR3 is expressed in various cells including platelets. As shown in **Figure 8 e/f** we found that the effect of CXCR7-agonists on platelet activation and thrombus formation was virtually absent in platelets obtained from *Ackr3*^{-/-} compared to *Ackr3*^{fl/fl} mice. Thus, in our *in vitro* experiments platelet-ACKR3 is a primary target receptor for the tested CXCR7-agonists. Further we have performed *in vivo* I/R experiments in *Ackr3*^{-/-} and *Ackr3*^{fl/fl} mice treated with a CXCR7 agonist. We found that myocardial ischemia (infarction of area at risk) and cell infiltration was reduced in *Ackr3*^{fl/fl} mice treated with CXCR7-agonist. In contrast, no significant effect of the CXCR7-agonist was observed in ACKR3-deficient mice (**please see Figure 8**). Both effects were completely abolished

in *Ackr3*^{-/-} mice compared to *Ackr3*^{fl/fl} control animals (**Figure 8**). Thus, we conclude that platelet ACKR3 is primarily involved in the CXCR7 agonist dependent observed effects on myocardial ischemia. This, however, does not exclude that other ACKR3 positive cells may be additionally involved. We have addressed this point in the discussion section.

Figure 8: Activation of ACKR3 inhibits platelet activation, thrombus formation and myocardial injury following I/R. a Upper left Statistical analysis of platelet P-selectin (CD62P) flow cytometry signals after treatment with ACKR3 agonists (100 μ M; vehicle control: 1 % DMSO) and activation. **Upper right / lower row** Dose-response curves of P-selectin expression after treatment with increasing concentrations of ACKR3 agonists (VUF11207 or C10; control: C46) acquired by flow cytometry. Plotted: mean \pm S.E.M.; n = 6; statistics: one-way ANOVA; **b** Impedance platelet aggregometry measurements of treated human blood (Agonists:100 μ M VUF11207 or 100 μ M C10; control substance: 100 μ M C46; vehicle control: 1 % DMSO). Plotted: mean \pm S.E.M.; n \geq 3; statistics: Student's t-test. **c** Representative images of flow chamber experiments conducted with human blood (1000 s⁻¹ on collagen, scale bar = 100 μ m) and statistical analysis of ACKR3 agonists effects (100 μ M VUF11207 or 100 μ M C10; control: 100 μ M C46) upon thrombus area [%]. Plotted: mean \pm S.E.M.; n \geq 3; statistics: one-way ANOVA. **d** Representative images of thrombus formation and analysis of thrombus area [%] within the left carotid artery after 5 min ligature of C57BL/6J treated with ACKR3 agonist (300 μ g VUF11207; vehicle control: 1% DMSO). Plotted: mean \pm S.E.M.; n \geq 3; statistics: Student's t-test. **e** Flow cytometry measurements of murine platelets from *Ackr3*^{-/-} and *Ackr3*^{fl/fl} treated with ACKR3 agonist (100 μ M C10; vehicle control: 1% DMSO). Plotted: mean \pm S.E.M.; n \geq 3; statistics: Student's t-test. **f** Representative images of flow chamber experiments conducted with murine whole blood from *Ackr3*^{-/-} and *Ackr3*^{fl/fl} (1000 s⁻¹ on collagen; scale bar = 100 μ m) and statistical analysis of the effect of an ACKR3 agonist (100 μ M C10) upon thrombus area [%]. Plotted: mean \pm S.E.M.; n \geq 3; statistics: Student's t-test. **g** Effect of ACKR3 agonist (300 μ g VUF11207, 1 hour before ischemia) treatment on *Ackr3*^{fl/fl} and *Ackr3*^{-/-} mice upon I/R of the heart, with statistic comparison analysis of infarct area [%] of the AAR and the AAR [%] of the left ventricle. Plotted: mean \pm S.D.; n \geq 5; statistics: student's t-test. **h/i** HE staining and statistical analysis of the infarct area **h** *Ackr3*^{fl/fl} **i** *Ackr3*^{-/-}. **a-i** tested against: \blacktriangledown ; n.s. = not significant; * p < 0.05; ** p < 0.01, *** p<0.001, **** p<0.0001.

Comments of Reviewer #2:

The manuscript of Rohlfing and coworkers describe an interesting and novel role of the chemokine receptor CXCR7 on platelet function and its consequences for thrombo-inflammation and ischemic tissue injury. The authors generated and characterized a new mouse strain deficient in platelet CXCR7 and show that loss of CXCR7 results in enhanced platelet activation and platelet-dependent thrombus formation in vitro and in vivo. Rohlfing et al. then analyzed the effect of CXCR7 deficiency in platelets on ischemia/reperfusion injury in myocardium and brain. Further, the authors show that CXCR7 agonists reduce platelet-dependent thrombus formation in vitro and in vivo and result in reduction of myocardial ischemia/reperfusion injury in mice. Finally, Rohlfing et al. show in patients with coronary artery disease that surface expression of CXCR7 is associated with clinical prognosis.

This is an interesting basic science manuscript with direct translational relevance in a highly topical clinical area. The authors provide an extensive array of experimental evidence, including knockout experiments, pharmacological and clinical data. The data are very well presented and well discussed.

To improve the manuscript the authors should address the following points of critique:

Response to Reviewer #2:

We thank the reviewer for his/her favorable and constructive comments. In the following we would like to address the comments and concerns point-by-point.

General points to address:

1. The authors show that loss of platelet CXCR7 enhances myocardial inflammation 24 h after ischemia/reperfusion. The authors should also analyze a longer time interval e.g. 28 days after transient ischemia/reperfusion to characterize whether this short-term effect has long-term consequences for myocardial function and inflammation. Further, it would be of interest to see the long-term effects on myocardial fibrosis and neovascularization.

Response to Reviewer #2:

We thank the reviewer for this valuable and justified comments. To further evaluate the impact of *Ackr3*-deficiency in platelets on functional outcome we performed additional experiments. I/R experiments were performed in *Ackr3*^{-/-} and *Ackr3*^{fl/fl} mice and the extent of myocardial infarction and dysfunction was analyzed after **28 days**. We found that the extent of the infarcted myocardium was substantially enhanced in *Ackr3*^{-/-} and *Ackr3*^{fl/fl} mice (**please see new Figure 6**). The extent of the infarcted myocardium has also functional consequences. In *Ackr3*^{-/-} mice myocardial function was significantly impaired at 28d I/R compared to *Ackr3*^{fl/fl} mice (**see Figure 6**) (ejection fraction, fractional shortening, echocardiographic assessment). Interestingly, whereas in *Ackr3*^{fl/fl} mice myocardial function almost recovered following I/R, the loss of myocardial function was sustained in *Ackr3*^{-/-} mice. Loss of myocardial function in *Ackr3*^{-/-} was paralleled by a substantial enhanced myocardial fibrosis in *Ackr3*^{-/-} compared to *Ackr3*^{fl/fl} mice (**see Figure 6**). Fibrosis was prominently found predominantly in midventricular section of the left ventricle (**see Figure 6**). Speckle tracking analysis revealed a loss of global longitudinal but not a circumferential strain function (**see Figure 6**).

Further, we quantified the extend of neovascularization by CD31 immunostaining of the infarct area of *Ackr3*^{-/-} and *Ackr3*^{fl/fl} mice after 28 days of reperfusion. We observed clear neovascularization signals in the wild type mice but the amount of CD31⁺ vessel was significantly reduced in *Ackr3*^{-/-} mice (**Figure 6**). These results indicate an impaired tissue regeneration in the *Ackr3*^{-/-} animals.

The data of the new experiments provide evidence that platelet-ACKR3 deficiency results in a sustained and irreversible loss of myocardial function, reduced regeneration of the infarct area and enhanced fibrosis. These long-term analysis (28 days) combined with the enhanced myocardial inflammation early after reperfusion (24 hours) imply that enhanced platelet activation early in the time course of myocardial ischemia is associated with enhanced fibrosis and loss of myocardial function in the long term. Platelet ACKR3 seems to play a prominent role in disease progression.

Figure 6: 30 min ischemia and 28-day reperfusion of the heart in *Ackr3*^{-/-} versus *Ackr3*^{fl/fl} mice. **a** Schematic drawing of the time line and heart sectioning of the for long term functional outcome of MI and reperfusion. **b** Representative TTC/Evans blue staining of the heart section S4 **b** Statistical analysis of the area at risk of the left ventricle in percent and the infarction of the area at risk in percent. Plotted: Mean±SD; n > 7; Statistics: Student's t-test; ns = not significant, **** p < 0.0001. **c** Representative pictures of Sirius Red staining of the fibrosis within the heart section 4 and statistical analysis of the fibrosis in the whole heart. Plotted: Mean±SEM; n > 7; Statistics: Student's t-test; ns = not significant, * p < 0.05. **d** Statistical analysis of the fibrosis separated by heart sections S1 to S4. Plotted: Mean±SEM; n > 7; Statistics: Student's t-test; ns = not significant, * p < 0.05. **e** Strain analysis of the heart in *Ackr3*^{-/-} versus *Ackr3*^{fl/fl} mice. Plotted: Mean±SEM; n > 7; Statistics: Student's t-test; ** p < 0.01, *** p < 0.001. **f** Statistical analysis of the ejection fraction of the heart over time in *Ackr3*^{-/-} versus *Ackr3*^{fl/fl} mice. **g** Representative images of CD31 staining and statistical analysis CD31+ vessels in *Ackr3*^{-/-} versus *Ackr3*^{fl/fl} mice. Plotted: Mean±SEM; n > 7; Statistics: Student's t-test; *** p < 0.001.

2. The authors mention that loss of platelet CXCR7 has proinflammatory effects. This is an interesting aspect; however, the authors miss to provide data e.g. on markers of systemic inflammation following ischemia/reperfusion in their genetic mouse model.

Response to Reviewer #2:

Thank you for your observation. As suggested we evaluated markers of systemic inflammation following I/R. Plasma of *Ackr3*^{-/-} and *Ackr3*^{fl/fl} mice were obtained 24 hours after reperfusion and analyzed for levels of chemokines and platelet/leukocyte interaction. Circulating platelet/macrophages co-aggregates (CD42/CD14⁺) were significantly enhanced in *Ackr3*^{-/-} versus *Ackr3*^{fl/fl} mice (please see supplemental Figure 14), indicating enhanced interaction of platelets with monocytes in *Ackr3*^{-/-} mice following I/R. Further, plasma levels of interleukin 6 (IL-6) was significantly enhanced in *Ackr3*^{-/-} versus *Ackr3*^{fl/fl} mice 24h after reperfusion. No significant changes were observed for plasma CXCL-12, CCL-2/-5, IL-1alpha/-beta, or TNF alpha (supplemental Figure 14). Thus, the data indicate that deficiency in platelet-ACKR3 has an impact on systemic inflammation after I/R.

Supplemental Figure 14: Platelet and plasma parameter after I/R. **a** Platelet aggregates in whole blood 24 h post I/R. **b** The chemokine concentrations in the plasma after ischemia and reperfusion on the heart in *Ackr3*^{-/-} and *Ackr3*^{fl/fl} animals are altered. A Statistical analysis of ELISA data of Plasma derived from *Ackr3*^{-/-} and *Ackr3*^{fl/fl} animals reveal a significant increase of IL-6 and CXCL5 plasma levels in *Ackr3*^{-/-} animals after I/R. Plotted: Mean±SEM; n > 7; Statistics: Student's t-test; ns = not significant, * p < 0.05, ** p < 0.01.

3. The authors show that loss of CXCR7 results in enhanced platelet degranulation. It would be of interest to characterize the release of inflammatory mediators in platelets derived from CXCR7-deficient and WT mice.

Response to Reviewer #2:

As suggested we analyzed the CRP-induced secretome in platelets derived from *Ackr3*^{-/-} and *Ackr3*^{fl/fl} mice. We performed mass spectrometry of the CRP-induced platelet releasate from *Ackr3*^{-/-} (n=3) and *Ackr3*^{fl/fl} mice (n=4). We identified 1.152 proteins in the CRP-induced releasate. Two proteins were significantly upregulated and 21 proteins downregulated in *Ackr3*^{-/-} mice (**please see supplemental Figure 9**). Most interestingly among the downregulated proteins there was an enrichment towards proteins associated with mitochondrial metabolism and microvesicles. Thus, although we cannot define an exact molecular mechanism at present, it is tempting to speculate that variance in the protein composition of the releasate between the two mouse strains may affect at site of tissue ischemia platelet-dependent inflammation. We are currently evaluating this issue, however at present we cannot provide conclusive evidence how this changes of the platelet secretome modulates tissue inflammation following I/R. Conventional inflammatory chemokines/cytokines were such as CXCL12, IL, TNFalpha were not different between the mouse strains (data not shown).

Supplemental Figure 9: Mass spectrometry analysis of activated platelets supernatants (APS). **a** Volcano blot presentation of the significantly changed proteins in APS. **b** Heat map presentation of the significantly changed proteins in APS.

4. The authors mention that CXCR7 is a novel inhibitory receptor on platelets. This is an interesting aspect; however, the authors should discuss the potential relevance of their findings for ischemic tissue injury.

Response to Reviewer #2:

We thank the reviewer for this interesting suggestion. Our data implies that loss of ACKR3 or agonist-induced activation results in modulation of platelet activation. Thus, targeting ACKR3 reduced platelet activation and secretion and thus platelet-dependent thrombo-inflammation and tissue inflammation following I/R. Therefore, augmenting the ACKR3 activity by agonists may reduce platelet activation and platelet-dependent thrombo-inflammation and tissue inflammation early in the time course of I/R which may have a favorable aspect to control inflammation and myocardial fibrosis. We have this issue discussed in the revised manuscript (please see revised discussion).

5. The authors should introduce the CXCR7 agonists VUF11207 and C10. They discuss the interesting pharmacological approach to activate CXCR7 therapeutically. The authors should further elaborate on

this important translational perspective, including the discussion on the potential impact of agonist-induced internalization of the receptor.

Response to Reviewer #2:

As suggested we have introduced CXCR7 agonists (please see revised introduction and discussion). As already described above targeting ACKR3 reduced platelet activation and secretion and thus platelet-dependent thrombo-inflammation and tissue inflammation following I/R. Thus, augmenting the ACKR3 activity by agonists may reduce platelet activation and platelet-dependent thrombo-inflammation and tissue inflammation early in the time course of I/R which may have a favorable aspect to control inflammation and myocardial fibrosis, thus my results in preservation of myocardial function. We have addressed this issue in the revised manuscript.

Specific points to address:

1. The clinical study present data that support the concept of using surface expression of receptors on platelets for clinical prognosis. The authors should provide some further data on potential cofounders that are associated with platelet CXCR7 surface expression (e.g. co-medication, LV-function, cardiovascular risk factors etc.). Also, in the abstract and conclusions the author's state there is an association with clinical prognosis. Rather than providing this unclear statement, the authors should state the actual association.

Response to Reviewer #2:

We thank the reviewer for this helpful comment. In the supplementary data section, we previously provided a multivariate Cox-regression analysis including all-cause mortality as independent variable and age, LVEF%, hyperlipidemia, ASA, clopidogrel, ACE inhibitors, ARBs, beta blockers, statins and ACKR3 as covariates. We decided to furthermore include gender, diabetes mellitus and smoking into the multivariate analysis. No patient without arterial hypertension deceased during the follow-up period. Thus, we did a subgroup analysis for hypertensive patients only. When we compared hypertensive patients and those without arterial hypertension we could not find a significant difference for CXCR7 platelet surface exposure levels (median MFI 26.4, 25th/75th percentile 17.8/33.5 vs 26.7, 25th/75th percentile 17.3/53.0, p=0.415).

Supplemental table 6

Variable	HR (All-cause mortality) (95% CI)	p-value
Age	1.10 (1.05-1.15)	<0.001
Gender	2.09 (0.93-4.69)	0.073
LVEF%	0.95 (0.92-0.98)	<0.001
Hyperlipidaemia	2.46 (0.94-6.42)	0.067
Diabetes mellitus	2.13 (1.03-4.39)	0.041
Smoking	1.60 (0.67-3.80)	0.287

ASA	0.90 (0.39-2.07)	0.805
Clopidogrel	0.88 (0.34-2.27)	0.791
ACE inhibitors	0.79 (0.29-2.08)	0.627
ARBs	0.74 (0.23-2.37)	0.614
Beta blockers	1.43 (0.51-4.02)	0.503
Statins	1.12 (0.46-2.77)	0.801
CXCR7 1 st vs 2 nd and 3 rd tertile	0.31 (0.14-0.69)	0.004

We furthermore agree that the statement of ACKR3 being associated with clinical prognosis is unclear. We thus rephrased the statement as follows: *“High platelet ACKR3 surface expression is significantly and independently associated with all-cause mortality in CAD patients”*.

2. Figure 2: Endothelial deficiency in CXCR7 has significant impact of aortic malformation. The authors should provide further gross anatomical phenotyping of the aorta and greater arteries. Do the platelet-specific CXCR7-KO mice have obvious malformations (e.g. aortic branch, valve, heart weight)?

Response to Reviewer #2:

We thank the reviewer for these comments. As suggested and in light of the described vascular malformations of endothelial deficiency in ACKR3, we have extensively phenotyped our *Ackr3*^{-/-} mouse strain (n=6). As shown in **new supplemental Figure 4** we did not observe any obvious changes in the circulatory gross morphology.

Supplemental Figure 4: Phenotypical comparison of organs from *Ackr3^{-/-}* and *Ackr3^{fl/fl}* mice. a-e representative images a heart b Aorta c Spleen d Kidney e *In situ* overview of the heart and aorta f-h Statistical analysis of organ weight and size f heart g Spleen h Kidney. Plotted: Mean±SEM.

3. The authors show enhanced P-selectin expression in CXCR7-KO mice. It would be of interest to see data derived from the TEM studies in terms of numbers of alpha-granules and dense-granules in CXCR7-deficient mice compared to WT mice. Also, CD62P immunostaining of single platelets would help to exclude that enhanced P-selectin expression is due to a higher number of alpha-granules.

Response to Reviewer #2:

As suggested we quantified the number of alpha as well as dense granules and mitochondria in our mouse strains and did not find significant changes (*please see Figure 2j*). Further, we performed CD62-staining of adherent platelets and did not find differences in the P-selectin staining pattern. Immunoblotting of platelet lysates of *Ackr3^{-/-}* shows similar CD62-positive immunoreactive bands compared to wild type *Ackr3^{fl/fl}* mice. Thus, the enhanced surface expression of *Ackr3^{-/-}* platelets may not be explained by an enhanced number or content of alpha granules (*supplemental Figure 7*).

Figure 2: Generation and characterization of *Ackr3*^{-/-} Pf4-cre⁺ mice. **a** Graphical presentation of the *Ackr3* Pf4-cre⁺ knock-out mouse generation. **b** Western blot sample illustrating the loss of ACKR3 protein in isolated *Ackr3*^{-/-} platelets compared to *Ackr3*^{fl/fl}. α -tubulin was used as loading control. **c** DIC image and Immunofluorescence staining of *Ackr3*^{-/-} and *Ackr3*^{fl/fl} platelets with anti-ACKR3 antibodies, Phalloidin and a corresponding IgG control. **d** Body weight, body length compared between *Ackr3*^{-/-} animals and *Ackr3*^{fl/fl}. $n \geq 8$. **e** Platelet count compared between *Ackr3*^{-/-} and *Ackr3*^{fl/fl} animals. $n = 12$. **f** Bleeding time compared between *Ackr3*^{-/-} and to *Ackr3*^{fl/fl} animals. $n \geq 9$. **g** Statistical analysis of the receptor expression on *Ackr3*^{-/-} platelets compared to *Ackr3*^{fl/fl}. $n = 5$. **h** Representative SICM images from *Ackr3*^{-/-} and *Ackr3*^{fl/fl} platelets. Statistical analysis of the platelet images obtained by SICM of *Ackr3*^{-/-} and *Ackr3*^{fl/fl} platelets. $n \geq 17$. **i** Representative images of *Ackr3*^{-/-} and *Ackr3*^{fl/fl} platelets spread on fibrinogen and activated by 1 μ g/ml CRP (scale = 10 μ m). Statistical analysis of the platelet images obtained by DIC microscopy of *Ackr3*^{-/-} and *Ackr3*^{fl/fl} platelets. $n \geq 55$. **j** Representative TEM images and organelle count in TEM cross section of murine platelets of from *Ackr3*^{-/-} and *Ackr3*^{fl/fl} mice. **Left:** Dense bodies per platelet cross section (> 2 μ m diameter) in murine platelets, *Ackr3*^{fl/fl} ($n = 30$) vs *Ackr3*^{-/-} ($n = 48$). **Middle:** Alpha granules per platelet cross section (> 2 μ m diameter) in murine platelets, *Ackr3*^{fl/fl} ($n = 30$) vs *Ackr3*^{-/-} ($n = 48$). **Right:** Mitochondria per platelet cross section (> 2 μ m diameter) in murine platelets, *Ackr3*^{fl/fl} ($n = 30$) vs *Ackr3*^{-/-} ($n = 48$). (scale = 1 μ m). **b-j** Plotted: mean \pm S.E.M.; statistics: student's t-test; n.s. = not significant.

Supplemental Figure 7: P-Selectin expression. **a** P-Selectin expression upon activation with 5 µg/ml CRP or 0.02 U/ml Thrombin. Plotted: Mean±SEM; n > 7; Statistics: Student's t-test; ns = not significant, * p < 0.05. **b** Western blot analysis of two sets of platelet lysates generated from resting platelets isolated from *Ackr3^{fl/fl}* and *Ackr3^{-/-}* animals. The specific P-Selectin signal at approx. 140 kDa is marked with an arrow. **c** P-Selectin immunofluorescence staining of platelets isolated from *Ackr3^{fl/fl}* and *Ackr3^{-/-}*. Staining was performed with Phalloidin-Rhodamin and CD62P – ALEXA488 of stimulated and unstimulated platelets (scale = 10 µm).

4. Figure 2: In the adhesion experiments performed by the authors, is spreading different between platelets derived from CXCR7-deficient mice compared to WT mice?

Response to Reviewer #2:

As suggested we have analyzed platelet adhesion and spreading on immobilized fibrinogen of platelets derived from *Ackr3^{-/-}* and *Ackr3^{fl/fl}* mice. As shown in the **supplemental Figure 5**, we didn't observe significant differences between the groups.

Supplemental Figure 5: Spreading analysis of *Ackr3^{fl/fl}* versus *Ackr3^{-/-}* animals. **a-d** Representative images of the various stages of spreading platelets. **a** unspread **b** filopodia **c** small lamellipodia **d** large lamellipodia **e** Statistical analysis of the observed spreading stages after 30 min activation with 1 µg/ml CRP. Plotted: Mean±SEM; n ≥ 4; Statistics: Student's t-test; ns = not significant.

5. Can the authors provide data on systemic inflammation following transient myocardial or brain ischemia in CXCR7-deficient mice compared to WT mice.

Response to Reviewer #2:

Thank you for your observation. As suggested we evaluated markers of systemic inflammation following I/R. Plasma of *Ackr3*^{-/-} and *Ackr3*^{fl/fl} mice were obtained 24 hours after reperfusion and analyzed for levels of chemokines and platelet/leukocyte interaction. Circulating platelet/macrophages co-aggregates (CD42/CD14⁺) were significantly enhanced in *Ackr3*^{-/-} versus *Ackr3*^{fl/fl} mice (please see supplemental Figure 14), indicating enhanced interaction of platelets with monocytes in *Ackr3*^{-/-} mice following I/R. Further, plasma levels of interleukin 6 (IL-6) was significantly enhanced in *Ackr3*^{-/-} versus *Ackr3*^{fl/fl} mice 24h after reperfusion. No significant changes were observed for plasma CXCL-12, CCL-2/-5, IL-1alpha/-beta, or TNF alpha (supplemental Figure 14). Thus, the data indicate that deficiency in platelet-ACKR3 has an impact on systemic inflammation after I/R.

Supplemental Figure 14: Platelet and plasma parameter after I/R. a Platelet aggregates in whole blood 24 h post I/R. **b** The chemokine concentrations in the plasma after ischemia and reperfusion on the heart in *Ackr3*^{-/-} and *Ackr3*^{fl/fl} animals are altered. A Statistical analysis of ELISA data of Plasma derived from *Ackr3*^{-/-} and *Ackr3*^{fl/fl} animals reveal a significant increase of IL-6 and CXCL5 plasma levels in *Ackr3*^{-/-} animals after I/R. Plotted: Mean±SEM; n > 7; Statistics: Student's t-test; ns = not significant, * p < 0.05, ** p < 0.01.

6. Figure 3: The authors show enhanced degranulation in CXCR7-deficient platelets. Is this specific for CRP as agonist? What about other platelet agonists, such as ADP or thrombin?

Response to Reviewer #2:

As suggested we evaluated P-selectin surface expression induced by thrombin (0.02 U/ml). We did not find a statistically significant difference between both groups in contrast to CRP. This implies that the observed enhanced P-selectin expression in *Ackr3*^{-/-} platelets is primarily dependent on GPVI-dependent alpha-degranulation (please see supplemental Figure 7a).

Supplemental Figure 7: P-Selectin expression. a P-Selectin expression upon activation with 5 µg/ml CRP or 0.02 U/ml Thrombin. Plotted: Mean±SEM; n > 7; Statistics: Student's t-test; ns = not significant, * p < 0.05.

7. Figure 7: The authors should further characterize the effect on CXCR7 agonists on platelet function. Do CXCR7 modulate Ca²⁺ signaling in platelets? What about platelet adhesion and spreading?

Response to Reviewer #2:

As suggested we evaluated the effect of CXCR7 agonist on Ca²⁺ signaling in human platelets. We found, that the CXCR7 agonist (C10) significantly reduces CRP-induced Ca²⁺ signaling compared to an irrelevant small molecule compound (C46) (*please see supplement Figure 16*). The effect of the CXCR7 agonist on platelet spreading raised. Therefore, we performed spreading experiments on fibrinogen and CRP activation (1 µg/ml). Samples were treated with 100 µM CXCR7 Agonist and categorized after 30 min in to four defined stages of platelet spreading. No significant effects of CXCR7 agonists on platelet spreading were observed.

Supplemental Figure 16: Effect of the CXCR7 Agonists on Calcium measurements and platelet spreading. a/b Representative image and statistical analysis of the Calcium increase upon activation with 5 µg/ml CRP in washed platelets treated with 100 µM C10 or C46. c Spreading of platelets on fibrinogen after 1 µg/ml CRP activation under treatment with 100 µM C10 or C46. Plotted: Mean±SEM; n > 7; Statistics: Student's t-test; n.s. = not significant * p < 0.05.

8. CXCR7 has been suggested as a survival receptor. Apoptosis in platelets has recently attracted major attention. The authors should provide data on the potential role of CXCR7 on platelet apoptosis.

Response to Reviewer #2:

We thank the reviewer for this valuable comment. As suggested we have analyzed apoptosis (Annexin-V binding, TMRE) in platelets derived from *Ackr3*^{-/-} mice. We found significantly enhanced Annexin-V

binding and reduced TMRE fluorescence signals in *Ackr3*^{-/-} compared to control mice (***please see supplemental Figure 8***). Further, treatment of platelets with CXCR7 agonists reduces Annexin V binding and prevents platelets from loss of mitochondrial membrane potential (TMRE). This further strengthen the conclusion, that ACKR3 prevents platelets from activation and modulates survival.

Supplemental Figure 8: Statistical analysis of flow cytometry measurements. **a** Annexin V expression after stimulation with 1 µg/ml CRP. **b** Annexin V expression after treatment with 20 µM ABT737. Plotted: Mean±SEM; n > 7; Statistics: Student's t-test; n.s. = not significant, * p < 0.05. **c** Gating strategy.

9. Although the manuscript overall seems carefully prepared, it needs further editing. There are a few spelling (line 341: mic instead of mice) and grammar mistakes throughout the manuscript.

Response to Reviewer #2:

We apologize for the spelling and grammar mistakes and have revised our manuscript accordingly.

Comments of Reviewer #3:

The group of M. Gawaz has previously reported the expression of ACKR3 on human platelets. Here they show that elevated expression of ACKR3 on platelets is a positive prognostic marker in coronary artery disease. In a new mouse model, where ACKR3 is selectively deleted in megakaryocytes using a cre recombinase under the Pf4 promoter, they recapitulate the findings made in humans showing aggravated heart and brain tissue damage after reperfusion in mice with platelets deficient in ACKR3.

We thank the reviewer for his interest and valuable suggestions and comments. In the following we will answer and discuss all comments point-by-point.

1. The mouse model is essential for the study, but is poorly characterized. The breeding strategy is clear, but genotyping and ACKR3 deletion remain doubtful. Although megakaryocytes are rare in circulation and flushed bone marrow samples, evidence of the receptor in the cells should be documented with the available ACKR3-GFP reporter. Moreover, GFP expression in platelets could be shown. Previous studies, however, have shown that ACKR3 expression in leukocytes is solely restricted to the B cell compartment. Alternatively, real-time PCR on the transcripts for ACKR3 and internal controls are needed to confirm expression in megakaryocytes.

Response to Reviewer #3:

We agree with this justified comment. However, the ACKR3-GFP reporter strain C57BL/6-Ackr3^{tm1Litt/J} is not readily available. We contacted Prof. Littmann at the New York University Medical Center. However, his strain is currently in cryopreservation. We although discussed to obtain animals from Jackson Laboratory (Bar Harbor, Maine, USA; stock No: 008591), which do have cryo stocks of the strain. Recovery and breeding from cryo stock will require an extensive time, in both cases the retrieval of the strain would have taken at least 12 to 16 weeks. In addition, due to the current pandemic situation the international shipping animals is very complicated at present.

Furthermore, our University animal facility has decreased their capacity to comply with COVID-19 regulations and the regional board for animal experiments (Regierungspräsidium Baden-Württemberg) has prolonged processing time. Hence, we would like to ask the reviewer to accept our decision not to follow his/her suggestion at the time being.

However, as suggested as alternative, we show performed additional immunoblot analysis of washed platelet using ACKR3 specific antibodies, the polyclonal rabbit anti-CXCR7 antibody Abcam 72100 and the polyclonal rabbit anti-CXCR7 antibody Novus NBP1-31309 (**Figure 1 a/b for Reviewer 3**). Furthermore, we had have previously performed a immunoblot analysis for ACKR3 expression on isolated murine megakaryocytes, using the polyclonal rabbit anti-CXCR7 antibody Abcam 72100 (**Figure 1 c for Reviewer 3**). In isolated platelets as well as megakaryocytes a clear ACKR3 protein signal is visible in *Ackr3*^{fl/fl} animals, whereas the signal is missing in *Ackr3*^{-/-} samples.

Moreover, we performed immunostainings of platelets using the polyclonal rabbit anti-CXCR7 antibody Abcam 72100. We found specific positive fluorescence signals in *Ackr3*^{fl/fl} but not *Ackr3*^{-/-} platelets (**Figure 1 d for Reviewer 3**). ACKR3-specific Histogreen staining of murine megakaryocytes isolated from wildtype mice resulted in a specific staining signal from ACKR3 (**Figure 1 e for Reviewer 3**).

In addition, we freshly isolated murine megakaryocytes from bone marrow of *Ackr3^{fl/fl}* or *Ackr3^{-/-}* animals by three consecutive applications of a BSA gradients (**Figure 1 f for Reviewer 3**). From these samples and tissues samples from the spleen and liver we isolated the mRNA and performed a RT-qPCR for *Ackr3* and *gapdh*. For the RT-qPCR we used the qPCR primer published in the original publication of the ACKR3 flox strain.⁶ and applied equal amounts of cDNA (25 ng) per reaction. We determined the relative expression of *Ackr3* mRNA in *Ackr3^{-/-}* to *Ackr3^{fl/fl}* using *gapdh* for correction. The relative mRNA expression of *Ackr3* mRNA in *Ackr3^{-/-}* was 2.6 percent of the wild type control. These residual *Ackr3* expression in the knock out can be explained by a few remaining contaminating cells or a weak leakage of the PF4 Cre system. As the Cre recombinase system is based on an enzymatic reaction not a traditional knock out, leakage can occur, but we still have 97.4 percent knock down of *Ackr3* mRNA in *Ackr3^{-/-}* animals.

We also included a 2 percent agarose gel depicting the RT-PCR products from spleen, liver and megakaryocytes mRNA samples. The megakaryocyte wildtype band is comparable in intensity to the signals obtained in the spleen and liver. The mean CT for the samples were spleen 27.74, liver 29,45 and wild type megakaryocytes 27.57 (**Figure 1 f for Reviewer 3**).

Figure 1 for Reviewer 3: Analysis of ACKR3 expression in *Ackr3^{fl/fl}* vs *Ackr3^{-/-}* mice platelets and megakaryocytes. a 12 % SDS-Page with following immunoblot analysis using the polyclonal Rabbit anti-CXCR7 antibody Abcam 72100. A clear ACKR3 protein band (42 kDa; black arrow) is visible in the *Ackr3^{fl/fl}* platelets. This protein band is missing in the *Ackr3^{-/-}* platelets. a-Tubulin (55 kDa, white arrow head) expression

was used as loading control. **b** 8 % SDS-Page with following immunoblot analysis using the polyclonal Rabbit anti-CXCR7 antibody Novus NBP1-31309. A clear ACKR3 protein band (42 kDa; black arrow) is visible in the *Ackr3^{fl/fl}* platelets. This protein band is missing in the *Ackr3^{-/-}* platelets. An asterisk (*) marks a residual alpha-Tubulin signal in the ACKR3 staining. a-Tubulin expression (55 kDa, white arrow head) was used as loading control. **c** Immunoblot analysis using the polyclonal Rabbit anti-CXCR7 antibody Abcam 72100. A clear ACKR3 protein band (42 kDa; black arrow) is visible in the *Ackr3^{fl/fl}* megakaryocytes. This protein band is clearly missing in the *Ackr3^{-/-}* megakaryocytes. a-Tubulin expression (55 kDa, white arrow head) was used as loading control. **d** Immunofluorescence staining of *Ackr3^{fl/fl}* vs *Ackr3^{-/-}* mice platelets using the polyclonal Rabbit anti-CXCR7 antibody Abcam 72100 and Phalloidin. **e** Histogram Staining of a megakaryocyte using the polyclonal Rabbit anti-CXCR7 antibody Abcam 72100. **f** Isolated megakaryocytes from *Ackr3^{fl/fl}* and *Ackr3^{-/-}* mice (Scale bar = 50 μ m). Relative expression of *Ackr3* in isolated megakaryocytes from *Ackr3^{fl/fl}* and *Ackr3^{-/-}* mice measured by RT-qPCR using *gpdh* as internal control. 2% agarose gel depicting the results of an *Ackr3* specific RT-qPCR using mRNA samples from murine spleen and liver as well as isolated megakaryocytes from wildtype mice. The *Ackr3* specific band is indicated by an arrow. Scale bar = 50 μ m

References:

- Sierro F et al, Disrupted cardiac development but normal hematopoiesis in mice deficient in the second CXCL12/SDF-1 receptor, CXCR7. Proc Natl Acad Sci U S A. 2007 Sep 11;104(37):14759-64. doi: 10.1073/pnas.0702229104.

2. A FACS analysis with complete gating strategy from human and mouse blood is missing, starting from whole blood or platelet enriched plasma.

Response to Reviewer #3:

As suggested we describe complete gating strategies for all our FACS experiments performed with whole blood or tissue in the supplemental part of our manuscript (*please see supplemental Figures 6 and 8c*). The standard gating strategy for platelets in whole blood is depicted below. In case washed platelets were used for flow cytometry experiments, gating for platelets was not necessary as the purity of our platelet preparations were established previously (*supplemental Figures 6*).⁷

Supplemental Figure 6: Example for our gating strategy for CD42⁺ platelets in whole blood, secondary platelet populations were analyzed for P-Selection or JONA expression.

Supplemental Figure 8: c Example for the gating strategies in isolated platelets. In this case only the target signal hat to be differentiated between positive and negative.

References:

- Witte A et al, The chemokine CXCL14 mediates platelet function and migration via direct interaction with CXCR4. *Cardiovasc Res.* 2021 Feb 22;117(3):903-917. doi: 10.1093/cvr/cvaa080.

3. Figure 2 needs additional explanations. B: the comparison of "*CXCR7^{fl/fl}*" and "*CXCR7^{-/-}*" was performed how? Which cells? Whole blood? What animals? The strong 450 bp band in *CXCR7^{-/-}* animals suggests some enrichment over the 200 bp band. What does the 227bp band corresponds to? The used primers should give a different amplicon according to GeneBank

Response to Reviewer #3:

We apologize for being unprecise. For the western blot we used washed murine platelets (**Figure 2b**). Platelet preparation was performed according to well-defined and established isolation methods. In our experiments, we verified purity of our platelet preparations by flow cytometry and microscopic analysis (Nageotto counting chamber). We verified the absence of nucleated cells in these preparations and found a high purity of platelets with neglectable evidence of nucleated cells (less than 0.00011 %), a fact that is in accordance with the literature.⁷ Sample types were included in the manuscript.

The 1 % agarose gel depicting the genotyping PCR for Pf4-Cre positive mice (450 bp Pf4-Cre ^{+/+} and 200 bp PCR control band), Pf4-Cre negative mice (Pf4-Cre ^{-/-}; 200 bp control band) and a *A. dest* control in

Figure 2b was performed with murine tissues samples obtained from the ear during marking of freshly weaned mice according to the genotyping protocol provided by the Jackson Laboratory (JAX, Bar Harbor, Maine, USA) for strain C57BL/6-Tg(Pf4-icre)Q3Rsko/J, given at the time of purchase. The 200 bp band represents a control signal for the PCR reaction. The only function of these band is the validation of the PCR reaction itself, absence of these band does indicate a failed PCR reaction. But the band does not impact the 450 bp signal or the results gained from the presence or absence of the 450 bp band. The image was moved to **supplement Figure 1a** and now the entire gel including a 1kb base pair maker is presented (**supplement Figure 1 a-b**) and additional information about the sample and sample preparation were added to the manuscript.

Supplemental Figure 1: Phenotyping of *Ackr3*^{fl/fl} vs *Ackr3*^{-/-} mice. **a** Representative image of an agarose gel showing the genotyping PCR done with tissues samples from the ear of recently weaned mice for *Ackr3* flox homozygous (fl/fl, 347 bp), heterozygote (fl/-; 347 bp and 277 bp) and wildtype (-/-; 277 bp). **b** Representative image of a 1 % agarose gel depicting the genotyping PCR for Pf4-Cre positive mice (450 bp; Pf4-Cre^{+/+} and 200 bp PCR control band), Pf4-Cre negative mice (Pf4-Cre^{-/-}; 200 bp control band) and a *A. dest* control.

The old **Figure 2a** depicted a representative image of an 1 % agarose gel loaded with the genotyping PCR for the *Ackr3* flox/flox sides. Again, murine tissues samples from the ear were used. For these PCR we used the primer listed in the original publication of the ACKR3 flox strain.⁶ The resulting pattern are flox homozygous 347 bp (fl/fl), wildtype 277 bp (-/-) and heterozygote 347 bp and 277 bp (fl/-). Unfortunately, our manuscript contained a typing error stating a 227 bp band instead of 277 bp for the wildtype. We corrected this error and included the complete gel including 1kb base pair maker in the new **supplement Figure 1b**.

References:

6. Sierro F et al, Disrupted cardiac development but normal hematopoiesis in mice deficient in the second CXCL12/SDF-1 receptor, CXCR7. *Proc Natl Acad Sci U S A*. 2007 Sep 11;104(37):14759-64. doi: 10.1073/pnas.0702229104.
7. Witte A et al, The chemokine CXCL14 mediates platelet function and migration via direct interaction with CXCR4. *Cardiovasc Res*. 2021 Feb 22;117(3):903-917. doi: 10.1093/cvr/cvaa080.

4. Are the ACKR3fl/fl mice homozygous for Pf4-cre? Can it be excluded that the observed effects are due to cre expression?

Response to Reviewer #3:

We agree with this justified comment. In the revised manuscript we have added data from 24 hour I/R experiments using C57BL/6-Tg(Pf4-icre)Q3Rsko/J animals (Pf4 Cre^{+/+}). The mice were wildtype or heterozygous for Pf4-cre in correspondence to the original experiments performed with PF4 Cre^{-/-}; *Ackr3*^{fl/fl} and PF4 Cre^{+/+}; *Ackr3*^{-/-} animals. We did not find a significant difference in wild type versus

Pf4-Cre^{+/−} on myocardial ischemia and area at risk (*please see Figure 1f for Reviewer 3*). Therefore, the transgene Tg(Pf4-icre)Q3Rsko itself is unlikely to induce the findings described in our manuscript. Hence, our findings are in all probability caused by the *Ackr3* depletion.

Figure II for Reviewer 3: 30 min ischemia and 24 h reperfusion in Pf4-cre^{+/−} mice. a Time line depicting the performed procedure and exemplary TTC/Evans Blue staining 24 h post MI. **b** Statistical analysis of the infarct in percent of area at risk and the area at risk in percent of the left ventricle. **c** Baseline and 24 h post MI ultra sound examples and statistical analysis of the ejection fraction and fractional shortening in percent.

5. 2C: the western blot needs better controls. The authors reported previously a diffuse 55kD for human ACKR3, here the sharp 42kD contrasts and is not convincing. Trustworthy commercial antibodies (e.g 7tmantibodies.com) indicate similar Mw for human and mouse ACKR3 (~55kD) with a larger Mw than reported here. In addition, entire lanes would be more preferred. Were the samples boiled before loading?

Response to Reviewer #3:

We apologize for this inconsistency. The citation of our previous work, where we reported a 55 kDa signal is unclear. We indeed have depicted a immunoblot for immunoprecipitation samples detected with ACKR3 antibodies,⁸ which shows a prominent signal at 55 kDa. But this signal is labeled as a cross-reaction with the heavy chain of the IgG control, the actual ACKR3 signal in this western blot is less prominent, also lower in molecular weight and correctly labeled.

The calculated molecular weight of murine ACKR3 is 41.64 kDa or 41.49 kDa for human ACKR3 respectively (www.uni-prot.org or www.expasy.org - Compute pI/MW). ACKR3 has two glycosylation sites, which could indeed increase the molecular weight of a protein significantly. These modifications could also be cell type specific, which would explain the variation in immunoblot signals given by various companies for their individual ACKR3 antibodies, that range from 40 kDa to 55 kDa. To verify our signal, we used various antibodies. Examples for two antibodies, the polyclonal rabbit anti-CXCR7 antibody Abcam 72100 and the polyclonal rabbit anti-CXCR7 antibody Novus NBP1-31309, are shown in **Supplemental Figure 1 c-d**. Both companies are commonly known for producing high quality

antibodies. With both antibodies we have clear signal in *Ackr3^{fl/fl}* animals, which are absent in *Ackr3^{-/-}* samples. In **Figure I for Reviewer 3** (Question 1; reviewer 3) we also show immunoblots with megakaryocyte samples and immunofluorescence staining of platelets. In both cases specific signals are present and in accordance with the murine phenotype.

We also developed our own highly specific antibodies against the extracellular N-Terminus of human ACKR3. In the **Figure III for Reviewer 3**, three candidates out of 42 are depicted. Even though these antibodies are still in the developmental stage, all three give a strong signal at approximal 42 kDa in human and in murine platelets. In all preparations the 42 kDa signal is the most prominent signal in each lane. The weaker signal in murine samples can be explained by the amount of protein loaded onto the gel, which in case of murine samples was half the amount used for human samples, as less material was available. Although only three candidates are shown we have similar results for all of our 39 other candidates. All candidates were further verified by various experimental techniques. Therefore, we are confident, that the observed signal at 42 kD indeed represents ACKR3 in platelets.

Supplemental Figure 1: Phenotyping of *Ackr3^{fl/fl}* vs *Ackr3^{-/-}* mice. **c** 12 % SDS-Page with following western blot analysis using the polyclonal rabbit anti-CXCR7 antibody Abcam 72100. A clear ACKR3 protein band (42 kDa; black arrow) is visible in the *Ackr3^{fl/fl}* platelets. This protein band is missing in the *Ackr3^{-/-}* platelets. An asterisk (*) marks a residual alpha-Tubulin signal from a previous staining of the membrane with alpha tubulin within the ACKR3 detection. alpha-Tubulin (55 kDa, white arrow head) expression was used as loading control. **d** 8 % SDS-Page with following western blot analysis using the polyclonal Rabbit anti-CXCR7 antibody Novus NBP1-31309. A clear ACKR3 protein band (42 kDa; black arrow) is visible in the *Ackr3^{fl/fl}* platelets. This protein band is missing in the *Ackr3^{-/-}* platelets. a-Tubulin expression (55 kDa, white arrow head) was used as loading control.

Figure III for Reviewer 3: Western blot analysis of ACKR3 protein expression in *Ackr3^{fl/fl}* vs *Ackr3^{-/-}* mice platelets. a 12 % SDS-Page with following western blot analysis using the polyclonal rabbit anti-CXCR7 antibody Abcam 72100. A clear ACKR3 protein band (42 kDa; black arrow) is visible in the *Ackr3^{fl/fl}* platelets. This protein band is missing in the *Ackr3^{-/-}* platelets. An asterisk (*) marks a residual alpha-Tubulin signal from a previous staining of the membrane with alpha tubulin within the ACKR3 detection. alpha-Tubulin (55 kDa, white arrow head) expression was used as loading control. **b** 8 % SDS-Page with following western blot analysis using the polyclonal Rabbit anti-CXCR7 antibody Novus NBP1-31309. A clear ACKR3 protein band (42 kDa; black arrow) is visible in the *Ackr3^{fl/fl}* platelets. This protein band is missing in the *Ackr3^{-/-}* platelets. alpha-Tubulin expression (55 kDa, white arrow head) was used as loading control. **c** 10 % SDS-Page with following western blot analysis using three different antibodies raised against a N-terminal epitope on human ACKR3 using hybridoma cultures. A clear ACKR3 protein band (42 kDa; black arrow) is visible in HEK293 cells samples, human platelets and murine platelets.

References:

8. Chatterjee M et al, Macrophage migration inhibitory factor limits activation-induced apoptosis of platelets via CXCR7-dependent Akt signaling. *Circ Res.* 2014 Nov 7;115(11):939-49. doi: 10.1161/CIRCRESAHA.115.305171.

6. Figure 3: Stimulation with CRP induces larger P-selectin expression on ACKR3+ cells (from intracellular granules), but no enhanced ATP secretion. What about CXCL12? This is critical since CXCL12 is a ligand of ACKR3 and could have a feedback effect (e.g. measurements in the presence of AMD 3100). This is also relevant for the statement on line 175 were apparently only Ly6G myelocytes, but not lymphocytes were differentially attracted depending on platelet ACKR3 expression. Continuing line 178 the role of CXCR4 positive neutrophils, which play a critical role during reperfusion, is not addressed.

Response to Reviewer #3:

We agree with the reviewer that release of CXCL12 upon CRP-activation may affect P-selectin degranulation via feedback mechanisms on either CXCR4 or ACKR3. To test the potential involvement of CXCR4 for P-selectin expression, we pre-incubated platelets with blocking mAB directed against CXCR4⁷ or AMD3100, as suggested. We found that, CRP-induced P-selectin surface expression was not affected by these tested antibody/compound (*please see supplemental Figure 18*). Further, addition of recombinant CXCL12 to platelets prior to activation does not modulate CRP-induced P-selectin degranulation. When we performed this flow cytometric analysis in the presence of CXCR7-agonist, CRP-induced P-selectin expression was significantly reduced irrespectively of the presence of anti-CXCR4 or AMD3100 (*please see Figure XXVI below supplemental Figure 18*). We found, that CXCR4 is not required for CXCR7-agonist-dependent P-selectin degranulation. Thus, we conclude that CXCL12/CXCR4 feedback mechanisms is not the major contributor of ACKR3-dependent P-selectin degranulation.

Supplemental Figure 18: Specificity of the ACKR3 Agonist effect. Inhibition of the CXCR4 receptor by AMD3100 or an inhibiting CXCR4 antibody did not alter the significantly reduced CD62P surface expression measured by flow cytometry upon CRP activation after preincubation with the ACKR3 agonist VUF11207. Plotted: Mean±SEM; n > 6; Statistics: Student's t-test; ns = not significant, ** p < 0.01.

References:

- Witte A et al, The chemokine CXCL14 mediates platelet function and migration via direct interaction with CXCR4. *Cardiovasc Res.* 2021 Feb 22;117(3):903-917. doi: 10.1093/cvr/cvaa080.

7. Figure 4: What is the active compound in APS-stimulated migration? Is it AMD3100 sensitive? Is the phenotype of migrated monocytes different from the input? Are there differences in the phenotype of migrated monocytes depending on the presence of ACKR3 on the platelets (APS)?

Response to Reviewer #3:

This is a very justified and interesting research question. When we evaluated the aspect of platelet secretome derived from our two mouse strains, *Ackr3^{fl/fl}* and *Ackr3^{-/-}*, we first tested whether the content of known chemokines/cytokines are different. We did not find differences in concentrations of platelet-derived chemokines/cytokines including CXCL12, MIF, CXCL14, CXCL11, TNFalpha or IL-1 (ELISA). Mass spectroscopy of the platelet secretome of *Ackr3^{fl/fl}* and *Ackr3^{-/-}* revealed some significant differences in protein levels which are not related to known chemotactic compounds. Some of the regulated proteins are related to mitochondrial metabolism (**please see Figure 4 and response to Reviewer 2**). The impact on these changes on chemotaxis and migration of monocytes is not clear yet and we are currently trying to explore this issue. Further, we tested various neutralizing antibodies directed to CXCR4, CXCR7, CXCL12, CXCL14, and MIF on the effect of monocyte migration and did not find a clear association to APS derived from *Ackr3^{fl/fl}* or *Ackr3^{-/-}* platelets. What we consistently observed, however, was that monocyte migration of APS derived from *Ackr3^{-/-}* platelets was enhanced. Thus, we decided to start to build up a large-scale migration assay that allows to test various concentrations, inhibitors and incubation times and input of monocyte input. Unfortunately, our present results do not clearly answer the underlying mechanisms and we are currently intensively addressing the question, but we do not think that we can present conclusive data in a reachable time frame. Thus, we would like to ask the reviewer to accept our explanation and results at this stage of our research.

Figure 4: *Ackr3^{-/-}* platelets promote thrombo-inflammation *in vitro*. **a** Schematic drawing of a modified Boyden chamber used for the following experiments and representative images (scale bar = 200 µm). Statistical analysis of the monocyte migration toward RPS and APS (5 µg/ml CRP-XL) derived from *Ackr3^{-/-}* or *Ackr3^{fl/fl}*. The migration of monocytes was enhanced towards APS derived from *Ackr3^{-/-}* platelets compared to *Ackr3^{fl/fl}* APS. 50 ng/ml MCP-1 was used as positive control. n = 4. **b** Representative images of flow chamber experiments were performed with isolated human monocytes. The monocytes were perfused over *Ackr3^{fl/fl}* (left panel) and *Ackr3^{-/-}* (right panel) platelets spread (1 µg/ml CRP) on fibrinogen-coated cover slides. Yellow arrow heads point at rolling monocytes whereas asterisks indicate adhesive monocytes. (scale bar: 100 µm). Statistical analysis of the number of rolling monocytes within 40 sec of perfusion. An enhanced adhesion

between *Ackr3*^{-/-} platelets and human monocytes compared to *Ackr3*^{fl/fl} platelets was observed. n = 6. **c** Statistical analysis of the static adhesion of *Ackr3*^{-/-} platelets to human monocytes compared to wild type control platelets after a 2 h incubation period. An enhanced adhesion between *Ackr3*^{-/-} platelets and human monocytes was observed compared to *Ackr3*^{fl/fl} platelets after thrombin (0.1 U/ml) stimulation. n > 6. **d** Statistical analysis of spontaneous co-aggregate formation of *Ackr3*^{fl/fl} and *Ackr3*^{-/-} platelets measured by flow cytometry analysis under unstimulated and stimulated conditions (1 µg/ml CRP-XL). Image stream images from Ly6G⁺/platelet aggregates. n = 7 **a-d** Plotted: mean±S.E.M.; statistics: student's t-test; ns = not significant, * p < 0.05.

8. Figure 7a: the authors analyzed surface expression of P-selectin in platelets (wt and ko) stimulated with ACKR3 agonists. The authors should show corresponding flow cytometry data. Stimulation of platelets should also be performed with endogenous ligands (CXCL11, CXCL12 and MIF). The conclusion that ACKR3 is a repressing surface receptor on platelets is conceivable, but in the lack of any presented evidence, the statement is circumstantial and alternative mechanisms should be considered.

Response to Reviewer #3:

Thank you for the interesting question. We included additional flow cytometry gating strategies were applicable.

In addition, we performed flow chamber and flow cytometry experiments with the endogen ACKR3 agonists CXCL11, CXCL12 and MIF in concentration established in previous publications.⁸ The CXCR7 agonist C10 was used as control. We used human platelets for the experiments as they are readily available in large quantities. Murine samples would have been needed to be approved by the regional board for animal experiments, which would have been outside of the time frame of this revision. The experiments revealed, that for endogen agonist flow seems to be an auxiliary factor, as flow chamber experiments did indeed show a significant effect of the ACKR3 specific agonists CXCL11 and MIF comparable to the effect induced by the agonist C10 (***please see supplemental Figure 17***). The flow chamber experiments are further support our theory that ACKR3 is a repressing receptor in platelets.

Supplemental Figure 17: Effect of endogen agonists on platelets. **a** Representative images of flow chamber experiments with endogen agonist. **b** Statistical analysis of flow chamber experiments with endogen agonists. Plotted: Mean±SEM; n ≥ 4; Statistics: Student's t-test; n.s. = not significant * p < 0.05, ** p < 0.01, *** p < 0.001, **** p < 0.0001. Scale bar: 100 µm

References:

8. Chatterjee M et al, Macrophage migration inhibitory factor limits activation-induced apoptosis of platelets via CXCR7-dependent Akt signaling. *Circ Res.* 2014 Nov 7;115(11):939-49. doi: 10.1161/CIRCRESAHA.115.305171.

9. Minor: line 268 the authors state that previous names for CXCR7 were RDC1 and ACKR3. This is simply wrong. The IUPHAR nomenclature, that should be used throughout the paper, is ACKR3 and has replaced CXCR7, RDC1 and CMKOR1.

Response to Reviewer #3:

We apologize for the mistake and corrected the manuscript accordingly. ACKR3 or *Ackr3* are now used throughout the manuscript.

Comments of Reviewer #4:

The paper of Rohlfing et al. is interesting paper with substantial data and experiments showing that platelet CXCR7 regulates thrombosis and inflammation.

I have several remarks or suggestions:

1. Despite CXCR7 is associated with thrombosis, the high and low platelet CXCR7 are associated with all mortality and not with thrombotic events like stroke or MI. Since only 36 events occur I understand this choice, however, it would be good to add # thrombotic events to the suppl. data and a statement that association of CXCR7 with outcome should be taken with caution. The TRAP data are convincing.

Response to Reviewer #4:

This is a justified comment raised by the reviewer. Hence, we created a new **supplementary table 7** showing number of events and corresponding incidence rates per 100 person years stratified according to CXCR7 platelet surface exposure in the 1st tertile vs 2nd/3rd tertile.

Supplemental Table 7

Event	No. of events (1 st tertile vs 2 nd /3 rd tertile)	IR/100py	P
All-cause mortality	19/17	5.8/2.6	0.008
Myocardial infarction	21/25	6.4/3.8	0.052
Ischemic stroke	1/6	0.3/0.9	0.284

Furthermore, in the revised manuscript we stated that the association between CXCR7 platelet surface exposure and outcome should be taken with caution.

2 Do the authors have MIF data as the ligand of CXCR7?**Response to Reviewer #4:**

We appreciate this comment. Indeed, we have data on MIF platelet surface exposure, circulating MIF as well as MIF SNPs (rs2070767, rs755622, rs1007888) in the current CAD cohort. We could not find a significant correlation between MIF and CXCR7 platelet surface exposure levels (n=108, rho=0.107, p=0.271). Furthermore, we could not show a significant correlation between circulating MIF and CXCR7 platelet surface exposure levels in a very small subgroup of the current cohort (n=8, rho=0.071, p=0.867).

Finally, we analyzed CXCR7 platelet surface exposure between different MIF SNPs. We could neither find significant differences in CXCR7 surface exposure for rs2070767 (n=198, p=0.976), rs755622 (n=203, p=0.871) nor rs1007888 (n=204, p=0.416), respectively.

3 In the results the introduction of I/R injury starts with the following sentence:

“The remarkable increase in CXCR7 on circulating platelets of patients with acute myocardial infarction¹⁴ and the impact on prognosis (Figure 1) in patients with CAD implies that platelet-CXCR7 has a functional role in ischemia and reperfusion (I/R) injury.”

This sentence is very confusing, as I do not find baseline levels before ACS in fig 1 so this is not an increase. Secondly the impact on prognosis implies a functional role in IR Injury. So, a future all

mortality event implies an impact on acute damage after an infarct This reasoning is very unclear and needs explanation or rephrasing. Is the paper not mainly on CXCR7 on the crossroads between coagulation and inflammation and is the IR Injury models not merely an example of this.

Response to Reviewer #4:

We fully agree with the reviewer's comments. The hypothesis of the clinical study was to evaluate the association of platelet-CXCR7 surface expression on clinical prognosis. We found in this cohort analysis that patients with platelet-CXCR7 surface expression in the upper range have a more favorable clinical outcome. We do not have baseline levels for these patients because the analysis was performed at time of hospital admission and onset of symptoms. The clinical data is hypothesis generating and reflects a potential relationship of platelet-CXCR7 expression, a pro-thrombotic and pro-inflammatory disease state and a potential effect of myocardial recovery after ischemia. Indeed, we found in a previous clinical study that platelet-CXCR7 expression is associated with recovery of left ventricular function following infarction in humans.⁹ Our animal studies support this hypothesis and in the revised manuscript we present data on the long-term effect on myocardial function of platelet-CXCR7 deficiency (*please see new Figure 6*). Thus, we allow us to conclude that platelet-CXCR7 is somehow protective for adverse clinical events in patients at risk. We have addressed this issue in the discussion.

Figure 6: 30 min ischemia and 28-day reperfusion of the heart in *Ackr3*^{-/-} versus *Ackr3*^{fl/fl} mice. **a** Schematic drawing of the time line and heart sectioning of the for long term functional outcome of MI and reperfusion. **b** Representative TTC/Evans blue staining of the heart section S4 **b** Statistical analysis of the area at risk of the left ventricle in percent and the infarction of the area at risk in percent. Plotted: Mean±SD; n > 7; Statistics: Student's t-test; ns = not significant, **** p < 0.0001. **c** Representative pictures of Sirius Red staining of the fibrosis within the heart section 4 and statistical analysis of the fibrosis in the whole heart. Plotted: Mean±SEM; n > 7; Statistics: Student's t-test; ns = not significant, * p < 0.05. **d** Statistical analysis of the fibrosis separated by heart sections S1 to S4. Plotted: Mean±SEM; n > 7; Statistics: Student's t-test; ns = not significant, * p < 0.05. **e** Strain analysis of the heart in *Ackr3*^{-/-} versus *Ackr3*^{fl/fl} mice. Plotted: Mean±SEM; n > 7; Statistics: Student's t-test; ** p < 0.01, *** p < 0.001. **f** Statistical analysis of the ejection fraction of the heart over time in *Ackr3*^{-/-} versus *Ackr3*^{fl/fl} mice. **g** Representative images of CD31 staining and statistical analysis CD31+ vessels in *Ackr3*^{-/-} versus *Ackr3*^{fl/fl} mice. Plotted: Mean±SEM; n > 7; Statistics: Student's t-test; *** p < 0.001.

References:

- Rath D et al, Expression of stromal cell-derived factor-1 receptors CXCR4 and CXCR7 on circulating platelets of patients with acute coronary syndrome and association with left ventricular functional recovery. *Eur Heart J.* 2014 Feb;35(6):386-94. doi: 10.1093/eurheartj/eh448.

4. The nano string profiling shows clear up AND down regulation of inflammatory genes in both brain and heart. This includes pro and anti-inflammatory genes which can be expected and it results in influx of more inflammatory cells suggesting that the up regulation of inflammatory genes is towards the pro-inflammatory site. This is hard to determine from these data now as when more chemokine are stored in the cell the mRNA are often down and after release the respective mRNA goes up. So what is the net effect on protein level. Is this also towards pro-inflammatory genes so is there indeed an upregulation from eg TNFalpha or IL1B in the tissue (or blood)

Response to Reviewer #4:

We agree with the reviewers with this justified comments and concern. The reason to analyze part of the tissue transcriptome was to further characterize tissue inflammation in our two mouse strains. As suggested we further analyzed our myocardial tissue samples for protein expression of TNFalpha and IL-1beta. We found a positive staining for TNF alpha and IL-1 beta in both mouse tissue. However, the levels of TNFalpha and IL-1beta were significantly enhanced in the area at risk in *Ackr3*^{-/-} versus *Ackr3*^{fl/fl} mice (*please see supplemental Figure 12*).

Supplemental Figure 12: DAB immunostaining of the infarct area of heart samples from *Ackr3*^{-/-} to *Ackr3*^{fl/fl} mice 24 h post MI. a Representative image of IL-1beta staining and statistical analysis of the signal. **b** Representative images of TNF alpha staining and statistical analysis of the signal. **c** Representative image of an appropriate IgG Control. Plotted: Mean±SEM; n > 6; Statistics: Student's t-test; * p < 0.05. Scale bar: 200 μm.

REVIEWER COMMENTS

Reviewer #1 (Remarks to the Author):

This is a revised submission of a potentially interesting manuscript that this reviewer raised many concerns about. I give the authors a lot of credit for making substantial efforts to address some of the critical issues. They have performed additional experiments to characterize the CXCR7 (now called Ackr3) knockout mice in terms of the possible impact on bone marrow-derived immune cells. Their new data showed that Ackr3 knockout had no obvious effect on the number of bone marrow-derived immune cells or receptor expression on these cells. Thus, an acceptable conclusion from the new data set is that Ackr3 knockout does not appear to affect the development of bone marrow-derived immune cells. As the authors acknowledged, a remaining issue is the lack of data showing that the functions of bone marrow-derived immune cells are not altered under the Ackr3 knockout background, especially after myocardial or cerebral ischemia. This is a relatively minor concern.

A major concern about the study was the missing long-term outcome data for both heart and brain ischemia models. The authors have performed 30-min ischemia and 28 days of reperfusion in the heart to address this concern and showed both histological and functional outcomes. These efforts are appreciated. However, the remaining issue is the missing long-term outcome data from the tMCAO brain ischemia model. The authors explained that the mortality rate for the tMCAO model is too high. In this reviewer's opinion, such difficulty can be overcome by: 1) finding a stroke collaborator; 2) reducing the duration of MCAO. Obtaining long-term (>7 days) stroke outcomes has become a gold standard for basic science stroke research in the field. At 24 hours after tMCAO, the infarct and the neurological functions of the stroke mice are still evolving. Accordingly, the conclusion on stroke outcomes in the current study is not considered to be fully supported. This reviewer is not trying to be unreasonably picky; rather, they state the minimal requirement for reporting animal stroke research data nowadays.

Reviewer #2 (Remarks to the Author):

The authors have addressed my comments very well and have substantially improved and strengthened the manuscript.

Reviewer #3 (Remarks to the Author):

General:

The authors have extensively revised the manuscript and added substantially more data. Unfortunately, the main weaknesses of the message were not addressed. The manuscript remains superficial and descriptive. Any attempt to elaborate the mechanism how ACKR3 could act as inhibitor receptor on platelets is missing. The in vivo data are merely circumstantial and alternative explanations are not considered.

Specific:

The Pf4-cre ACKR3^{fl/fl} mouse is not sufficiently analyzed. Pf4 is NOT exclusively expressed in megakaryocytes, but also in multipotent progenitor cells and hematopoietic precursor cells, which could potentially delete ACKR3 in B cells, the only ACKR3⁺ leukocyte population (PubMed 17804806 and PubMed 21300915). Moreover, Pf4 is expressed in heart fibroblasts together with ACKR3 (Tabula Muris). The impact of potential deletion in these cells on cardiac function is completely ignored. Expression of CXCL4 (PF4) was reported in several types of leukocytes (PubMed 25393502 and references therein), which may markedly contribute the inflammatory status of Pf4-cre ACKR3^{null} mice.

The Pf4-cre mouse strain was generated using a 100 kB BAC that also encodes for several inflammatory chemokines (CXCL3, 5, 7 and 15; <https://www.jax.org/strain/008535>). Elevated CXCL5 levels were reported in Pf4-cre ACKR3 null mice (lines 211 and 268). Are these caused by BAC genes? In addition, the authors should determine the CXCL4 levels in platelets of Pf4-cre ACKR3 null mice

The effects of ACKR3 agonist are borderline. VUF11207 only shows a marginal inhibitory effect on

CRP stimulated P-selectin expression at > 1000 fold of the reported kd (PubMed 22424612). Similarly, the poorly described compound C10 (I could only find an abstract published in 2021 by the same authors on in silico data, without unveiling any affinity (https://dggk.org/kongress_programme/jt2021/aP530.html). Again, any effects are only observed at high microM concentrations. The systemic application of the ACKR3 "agonists" are useless, unless investigated more in detail.

Expression of ACKR3 on platelets is not granted as stated by the authors. In the past, the authors published western blots of platelets with various Mw for ACKR3: PubMed 25266363 50kD, PubMed 24668750 35 kD and PubMed 21468792 55kD. The actual size is now 42 kD, while reliable antibodies indicate ~55kD. What is detected? The response to my previous critique is insufficient. Were the samples boiled before loading on SDS PAGE (the question was ignored)? The resolution of the fluorescence microscopy images is poor and membrane localization of ACKR3 is not revealed. What is the meaning of the CRP secretome. The authors describe two genes up and 21 down, and? Incubation of human monocytes with whole mouse blot (line 180) is not clear. This experimental setup may induce a plethora of responses that might not be correlated with platelet ACKR3 expression.

The conclusion from the NanoString experiments (lines 217-219) is farfetched. Given the expression of Pf4 in different cells the dramatically enhanced inflammation cannot be attributed only to lack of ACKR3 expression in platelets.

The finding that CXCL12, which has about 10 fold higher affinity for ACKR3 than CXCL11, has no stimulatory capacity on platelets compared to CXCL11 needs explanation.

Minor:

Line 200: should read Ly6G+

CXCR7 was partially replaced with ACKR3, but not consistently including supplementary material, figures, tables and legends

Supplementary figure legend 9: should read "volcano plot"

In general the paper would benefit from rigorous editing (typos and grammar).

Reviewer #4 (Remarks to the Author):

Impressed by the amount of data generated for this rebuttal. Unbelievable

Nature Communications NCOMMS-20-50883A

Title: ACKR3 regulates platelet activation and ischemia - reperfusion tissue injury

Corresponding Author: Meinrad Gawaz

Response to the Reviewers

Reviewer 1

Comments of Reviewer #1:

This is a revised submission of a potentially interesting manuscript that this reviewer raised many concerns about. I give the authors a lot of credit for making substantial efforts to address some of the critical issues.

Response to Reviewer #1:

We thank the reviewer for his/her positive assessment of our revised manuscript. In the following we answer your questions point by point.

1. They have performed additional experiments to characterize the CXCR7 (now called *Ackr3*) knockout mice in terms of the possible impact on bone marrow-derived immune cells. Their new data showed that *Ackr3* knockout had no obvious effect on the number of bone marrow-derived immune cells or receptor expression on these cells. Thus, an acceptable conclusion from the new data set is that *Ackr3* knockout does not appear to affect the development of bone marrow-derived immune cells. As the authors acknowledged, a remaining issue is the lack of data showing that the functions of bone marrow-derived immune cells are not altered under the *Ackr3* knockout background, especially after myocardial or cerebral ischemia. This is a relatively minor concern.

Response to Reviewer #1:

We thank the reviewer for pointing out this observation. As the reviewer states, we did show no difference in number or receptor presentation in bone marrow-derived immune cells in the Pf4-Cre *Ackr3*^{-/-} background. This FACS analysis could be performed rather fast with our existing multi-panel set up, harvesting bones from animals euthanized for other experiments. Functional analysis, e.g. migration assays, of all various bone marrow-derived immune cells is far more complicated, time-consuming and we would need a vast number of experimental animals with pending approval by the local authorities to perform the experiments. The performed I/R experiments in PF4-Cre mice do show no significant difference in infarct size between wild type and PF4-Cre positive animals after 24 hours. This could indicate, that the PF4-Cre expression alone does not alter the function/migration of bone marrow-derived immune cells (**Figure I A/B**). We hope that these explanations will satisfy Reviewer 1.

Figure I: 30 min ischemia and 24 h reperfusion in Pf4-cre^{+/-} mice. **A** Time line depicting the performed procedure and exemplary TTC/Evans Blue staining 24 h post MI. **B** Statistical analysis of the infarct in percent of area at risk and the area at risk in percent of the left ventricle. Plotted: Mean±SEM; n = 5; Statistics: Student's t-test; n.s. = not significant.

2. A major concern about the study was the missing long-term outcome data for both heart and brain ischemia models. The authors have performed 30-min ischemia and 28 days of reperfusion in the heart to address this concern and showed both histological and functional outcomes. These efforts are appreciated. However, the remaining issue is the missing long-term outcome data from the tMCAO brain ischemia model. The authors explained that the mortality rate for the tMCAO model is too high. In this reviewer's opinion, such difficulty can be overcome by: 1) finding a stroke collaborator; 2) reducing the duration of MCAO. Obtaining long-term (>7 days) stroke outcomes has become a gold standard for basic science stroke research in the field. At 24 hours after tMCAO, the infarct and the neurological functions of the stroke mice are still evolving. Accordingly, the conclusion on stroke outcomes in the current study is not considered to be fully supported. This reviewer is not trying to be unreasonably picky; rather, they state the minimal requirement for reporting animal stroke research data nowadays.

Response to Reviewer #1:

As suggested we performed additional mice experiments looking at the effect of a 7-day reperfusion following I/R injury on brain ischemia. After a transient ischemia followed by reperfusion (tMCAO, 25 min), *Ackr3^{fl/fl}* and *Ackr3^{-/-}* animals were sacrificed and analyzed after 7 days. In correspondence with the 24 h experiments a significantly larger infarct volume was measured in *Ackr3^{-/-}* animals compared to wild type animals (**Figure II B**). These results further support our hypothesis that ACKR3 is crucial to control platelet-associated ischemia following I/R in the immediate 24 h and 7 d brain injury. This new data set has been incorporated into the revised version of our manuscript. We hope that we have now answered all concerns raised by Reviewer 1 and that our revised manuscript is now acceptable for publication.

Figure II: Additional long-term stroke experiments. **A** Schematic of the experimental set-up. **B** Representative images of Cresyl Violet stained brain sections and statistical analysis of the infarct volume 7 days post tMCAO. $n \geq 6$, Student's T-test, * $p < 0.05$.

Reviewer 2

Comments of Reviewer #2:

The authors have addressed my comments very well and have substantially improved and strengthened the manuscript.

Response to Reviewer 2

We thank the reviewer for his/her positive assessment of our revised manuscript.

Reviewer 3

We thank the reviewer for his/her additional comments and criticism. Whenever possible we have done our best to thoroughly answer your questions and concerns. In the following we would like to answer your questions step by step.

Comments of Reviewer #3:

General:

The authors have extensively revised the manuscript and added substantially more data. Unfortunately, the main weaknesses of the message were not addressed. The manuscript remains superficial and descriptive.

Response to Reviewer #3:

Indeed, we have done our best to add a significant amount of additional experimental data as suggested by you and the other three reviewers. However, we do not fully agree with Reviewer 3 that our manuscript is superficial and descriptive. We are convinced that our present manuscript describes several novel aspects that deepens our understanding of **i)** the role of platelets in the pathophysiology of I/R in heart and brain, **ii)** discloses platelet ACKR3 as a critical regulator in platelet function, thrombus formation and tissue inflammation, and **iii)** provide profound evidence that ACKR3 stimulation via agonists attenuates platelet activation, degranulation, Ca²⁺ signaling, and *ex vivo* and *in vivo* thrombosis.

In the past, a lot of data has been published that has addressed the ACKR3-dependent signaling events in nucleated cells. However, the exact underlying mechanisms of ACKR3-dependent signaling is still poorly defined. Disclosing the molecular mechanisms of ACKR3-dependent platelet function requires an extensive and longstanding research which is currently in progress in our group and also pursued by other researchers. This issue is not easily accomplished since ACKR3 interacts with CXCR4 under distinct circumstances which may alter signaling substantially. Thus, we hope that the reviewer will accept our explanation at this stage that we can only offer a few novel aspects on how ACKR3 may control platelet function.

1. Any attempt to elaborate the mechanism how ACKR3 could act as inhibitor receptor on platelets is missing. The *in vivo* data are merely circumstantial and alternative explanations are not considered.

Response to Reviewer #3:

As stated above, we provide novel data that discloses an important role of platelet ACKR3 in platelet function, thrombosis and inflammation. The *Ackr3*^{-/-} platelets are hyperreactive represented by increased degranulation, significantly enhanced *ex vivo* and *in vivo* thrombus formation (**Figure 3 a/d/e**). Calcium release, a main signaling pathway connected to degranulation of platelets, is significantly increased in *Ackr3*^{-/-} platelets. In addition, platelet co-aggregate formation with certain cell types is intensified (**Figure 4 b-c**). Also, apoptosis is increased in *Ackr3*^{-/-} platelets. All this points to a loss of inhibitory signal within the *Ackr3*^{-/-} platelets. The ACKR3 specific agonist binding to ACKR3 does significantly decrease the reactivity of platelets to stimuli such as ADP- or CRP- induced degranulation, aggregation and *ex vivo* thrombus formation (**Figure 8 a-f**). Further, *in vivo* thrombus formation is decreased in mice after application of an ACKR3 agonist (**Figure 8 d**). Therefore, ACKR3 has without doubt protective/inhibitory effects on platelet function.

As stated above, the investigation of the intracellular signaling pathways dependent on ACKR3 is still ongoing in nucleated cells. Three candidates have been described as potential ACKR3 interaction partner, G-proteins, G-protein-coupled receptor kinase 2 (GRK2) and β -arrestin (**Figure III A**)^{1,2,3,4}. β -arrestin has been long discussed and proven to be an interaction partner of ACKR3 in various cell types³. Recently, GRK2 gained attention as an independent interaction partner of ACKR3 and signal molecule in brain tissue apart from its role in β -arrestin recruitment^{1,2,5}. Furthermore, in recent publications ACKR3 has been excluded to interact with G-proteins in various cell types^{6,7}.

Figure III: **A** Schematic presentation of ACKR3 interaction partners. **B** Western blot depicting ACKR3 phosphorylation upon 1 μ g/ml SDF-1 stimulation. **C** Flow cytometry experiments with GRK2 inhibitor (100 μ M β -ARK Inhibitor), plotted: Mean \pm SD, $n \geq 5$, Student's T-test, **** $p > 0.0001$; * $p > 0.05$. **D** Flow chamber experiments with GRK2 inhibitor (100 μ M β -ARK Inhibitor), plotted: Mean \pm SD, $n \geq 12$, Student's T-test, ** $p > 0.01$ (preliminary data).

To address the issue of ACKR3-dependent signaling in platelets, we analyzed CXCL12 dependent phosphorylation of ACKR3 using phospho-specific antibodies against pS350/pT352 of ACKR3². We were able to detect an increase of ACKR3 phosphorylation at position pS350/pT352 after stimulation (**Figure III B**). ACKR3 is phosphorylated by GRK2^{2,5}. Inhibition of GRK2 by a specific inhibitor (β -ARK inhibitor) does decrease platelet degranulation and significantly inhibits *ex vivo* thrombus formation (**Figure III C/D**). This implies that phosphorylation by GRK2, independent or dependent on β -arrestin, seems to be part of the ACKR3 signaling in platelets. However, at this stage we are not able to give enough data to disclose the signaling cascade of ACKR3 in platelets. Further analysis of the ACKR3 interactome is necessary, new tools like chemokine chimera or functional ACKR3 reporter strains can help to reveal the network^{7,8}. The situation becomes even more complex when the potential co-receptor CXCR4 is taken into consideration. CXCR4 and ACKR3 form heterodimers under certain circumstances, which may have a dynamical effect on ACKR3-dependent signaling in platelets. Generation of platelet-specific CXCR4-deficient mice and also platelet-specific double CXCR4/ACKR3 knock out mice is a strategy to follow up this issue. Further, it seems to be necessary to apply a phosphor-proteomic approach to disclose the underlying complex signaling network in various functional states of platelets. This currently ongoing research work requires an intensive and thorough investigation at least for the next two years to come. Thus, we would like to ask the Reviewer 3 to accept our response at present and not to include our preliminary data into the present manuscript.

Specific:

2. The Pf4-cre ACKR3fl/fl mouse is not sufficiently analyzed. Pf4 is NOT exclusively expressed in megakaryocytes, but also in multipotent progenitor cells and hematopoietic precursor cells, which could potentially delete ACKR3 in B cells, the only ACKR3+ leukocyte population (PubMed 17804806 and PubMed 21300915).

Response to Reviewer #3:

We thank the reviewer for these valuable and justified comments. At the moment only two platelet-specific Cre systems exist: the Pf4-Cre and Gp1ba-Cre. Both systems have advantages and disadvantages as has been well summarized in the recent review by Gollomp and Poncz⁹. Like other cell-specific expression systems, the PF4-Cre system is not perfect and may have small leakages into other cell types. However, the Cre system is currently a state-of-the-art technology to create cell type specific knock out strains with a high level of target gene depletion. This allows us to have a more specific look in the environment of platelets, especially, as a global ACKR3 knock out mouse is not viable¹⁰.

Figure IV: Extraction of tabula muris data sets for murine marrow sample (source: Tabula Muris Consortium). **A** Expression in cell population in percent. **B** Expression level in counts per million.

As the reviewer points out, PF4/ACKR3 expression in multipotent progenitor cells and hematopoietic precursor cells has been demonstrated by the Tabula Muris project (<https://tabula-muris.ds.czbiohub.org/>)¹¹. But the extent of the expression has to be taken in account as well as how the data can be interpreted. Between approx. 3 and 12 % cells express PF4 except for megakaryocyte-erythroid progenitor cell which express PF4 in about 50 % of the cells (**Figure IV A/B**). In the same population only 1.5 to 7 % of the cells express ACKR3 in low levels. Often only 20-250 positive cells were identified in the whole FACS-based full length transcript analysis and used as basis to extrapolate expression levels. In the megakaryocyte-erythroid progenitor cell it is reasonable to presume that the PF4 expression is predominantly in the population of progenitor cells destined for the megakaryocyte pathway and also constitute the ACKR3 expressing cell pool. The amount of ACKR3 expressing cells is

very low in general (mostly 1.5–3 %) and without individual measurements to confirm the expression, this might be false positive results in a batch measurement set up. Especially, as false positive results can be explained by the strong interaction of platelets with other cell types, e.g. platelet/immune cell co-aggregates, and platelet phagocytosis by immune cells which are common events. The samples would have to be screened for platelet contamination, e.g., by CD42b detection. Either way the number of effected cells is quite low, direct measurements in the mature cell populations would be necessary to evaluate the actual effect. To measure PF4 and ACKR3 expression in all the various cell types and to evaluate the individual effects would be a major project and not constructive for the manuscript. But we did measure the cell count and the expression levels of cell type-specific receptors on bone marrow cells derived from *Ackr3^{fl/fl}* and *Ackr3^{-/-}* animals (**see Supplement Figure 2**). There was no significant difference in number or receptor presentation detectable. These results indicate a normal bone marrow cell population in *Ackr3^{-/-}* animals compared to wild type.

The above used reasoning can also be applied to B cells. 2.27 % of all B cells express PF4 and 4.55 % express ACKR3, with low expression levels ($< 1.4 \ln(1+CPM)$) in the applied batch set up¹¹. The cited article from Siervo *et al.* (PMID: 17804806) confirmed *Ackr3* mRNA expression in B cells, exact expression levels of mRNA or ACKR3 protein B-cells in wild type animals are not given¹⁰. PF4 expression was not analyzed in this publication. But the author stated that B cell development in fetal liver and bone marrow was normal in global ACKR3 knock out mice embryos. The global ACKR3 knock outs are not viable only in rare occasions animals survive until adulthood and even then, they are severely compromised. In two of these surviving animals a modest reduction in marginal zone B cells (MZ B cell) of the spleen was observed¹⁰. Given the small n and poor general state of health of these animals, these findings are not likely to be transferable to our platelet-specific knock out. Particularly as B cells were included in the multi-color flow cytometry screen we performed with bone marrow samples for the first revision. The B-cell count, MHC-II, CD45R/B220 and CD19 expression levels were not affected by the *Ackr3^{-/-}* knock out. This indicates the platelet-specific ACKR3 knock out did not affect cell count or cell type specific receptor presentation in B cells.

3. Moreover, Pf4 is expressed in heart fibroblasts together with ACKR3 (Tabula Muris). The impact of potential deletion in these cells on cardiac function is completely ignored. Expression of CXCL4 (PF4) was reported in several types of leukocytes (PubMed 25393502 and references therein), which may markedly contribute the inflammatory status of Pf4-cre ACKR3 null mice.

Response to Reviewer #3:

We thank the reviewer for this interesting comment. Indeed, the Tabula Muris does show an overlap of ACKR3 and PF4 expression in heart fibroblast (<https://tabula-muris.ds.czbiohub.org/>)¹¹. But only 6.18 % of the cells do express PF4, whereas ACKR3 expression could be detected in 49.83 % of all heart fibroblast (**Figure V**). Recently, Ishizuka *et al.* (Nature, 2021; PubMed: 34099846) did create a fibroblast-specific ACKR3 knock out mice¹². In these mice the myocardial injury after I/R of the heart was not significantly altered compared to wild type controls. Thus, any effect of the PF4-Cre background on heart fibroblast within our *Ackr3^{-/-}* animal model is highly unlikely to have caused the effects observed in our I/R experiments.

In leukocytes, the Tabula Muris lists inverse results, 76.67 % of the leukocytes highly express PF4 whereas only 5.16 % express ACKR3 at very low levels (**Figure V**)¹¹. Therefore, this might be a cofounder but only on a low scale in a very small population of leukocytes. In our opinion, these cells could only have a small effect on the heart tissues and could not explain the strong reproducible effect we are observing in our experiments.

Figure V: Tabula Muris flow cytometers data on PF4 and ACKR3 expression levels in heart leukocytes and fibroblast (source: Tabula Muris Consortium). **A** Percentage of cells expressing PFA or ACKR3. **B** Expression levels of PF4 and ACKR3 in positive cell populations.

4. The *Pf4-cre* mouse strain was generated using a 100 kB BAC that also encodes for several inflammatory chemokines (CXCL3, 5, 7 and 15; <https://www.jax.org/strain/008535>). Elevated CXCL5 levels were reported in *Pf4-cre* ACKR3 null mice (lines 211 and 268). Are these caused by BAC genes? In addition, the authors should determine the CXCL4 levels in platelets of *Pf4-cre* ACKR3 null mice.

Response to Reviewer #3:

We appreciate the question of the reviewer. Elevation of CXCL5 is may be caused by the BAC insertion. At this moment we are not able to separate the potential BAC effect from our data in terms of inflammation. However, it is highly unlikely that it may influence our results on platelet function and thrombosis, especially the *ex vivo* data.

In addition, control I/R experiments in *Pf4-Cre* mice showed no differences in area-at-risk and infarct size between wild type and *Pf4-Cre* positive animals after 24 hours, even though the BAC presence is identical to the *Ackr3^{-/-}* strain. Moreover, there was no difference in the cell infiltration in the area-at-risk 24 h after ischemia in *Pf4-Cre* positive animals compared to wild type littermates (see below **Figure VI**). Thus, it is highly unlikely that BAC dependent chemokine expression causes the enhanced myocardial ischemia and inflammation observed in the *Ackr3^{-/-}* animals following I/R.

Figure VI: 30 min ischemia and 24 h reperfusion in *Pf4Cre^{-/-}* mice. **A** Time line depicting the performed procedure and exemplary TTC/Evans Blue staining 24 h post MI. **B** Statistical analysis of the infarct in percent of area at risk and the area at risk in percent of the left ventricle. Plotted: Mean±SEM; n = 5; Statistics: Student's t-test; n.s. = not significant. **C** Representative HE staining area at risk of *Pf4Cre^{-/-}* and *Pf4Cre^{+/-}* samples and statistical analysis of the nuclei count. Plotted: Mean±SEM; n = 5; Statistics: Student's t-test; n.s. = not significant.

As suggested, we evaluated protein expression of PF4 in platelets. The PF4 protein levels are not altered between *Ackr3*^{fl/fl} and *Ackr3*^{-/-} platelets as verified by immunoblot experiments (**Figure VII**).

Figure VII: **A** Immunoblot detecting platelet PF4 (black arrow) in lysates from *Ackr3*^{-/-} and *Ackr3*^{fl/fl} samples. As loading control α -tubulin was detected (white arrow head). **B** Densitometric measurements of the PF4 signal in samples from *Ackr3*^{-/-} and *Ackr3*^{fl/fl}. Mean \pm SEM, n \geq 3, Student's t-test.

5. The effects of ACKR3 agonist are borderline. VUF11207 only shows a marginal inhibitory effect on CRP stimulated P-selectin expression at > 1000fold of the reported kd (PubMed 22424612). Similarly, the poorly described compound C10 (I could only find an abstract published in 2021 by the same authors on in silico data, without unveiling any affinity (https://dgk.org/kongress_programme/jt2021/aP530.html)). Again, any effects are only observed at high μ M concentrations. The systemic application of the ACKR3 "agonists" are useless, unless investigated more in detail.

Response to Reviewer #3:

We thank the reviewer for this comment. The authors of the cited article (PubMed 22424612) did indeed determine the pK_i (8.1) and EC_{50} (1.6 nM)¹³. The inhibition constant K_i was determined by an inhibition assay (inhibition kinetics) and not by direct binding kinetics, which would allow to assert the dissociation constant K_D . Further, the cited articles measured an EC_{50} of 1.6 nM using a bioluminescence resonance energy transfer (BRET) signal assay which has been performed in HEK293 cells. An ACKR3 internalization assay, also performed in HEK293 cells, did yield an EC_{50} of 14.1 nM. We gained comparable results for VUF11207 in CHO cells (EC_{50} of 3.986 nM) using a PathHunter β -Arrestin eXpress GPCR Assay system (Eurofins DiscoverX, Fremont, CA, USA). Admittedly, these results are substantially lower than the concentration (100 μ M) used in our submission. However, the EC_{50} is dependent on several factors including exposure time, species, tissue, cell type and genetics. If we compare our measurements with the assays performed above, the ACKR3 internalization assay required 60 min incubation with VUF11207, whereas our experiments are performed after a 30 min preincubation period. The assays were performed in Human Embryonic Kidney cells (HEK293) or Chinese Hamster Ovary (CHO) cells transfected with recombinant human ACKR3, which results in overexpression of the receptor, whereas our experiments use murine or human platelets with an endogenous ACKR3 expression. Therefore, the tissue (kidney/ovary vs hematopoietic system), cell type (epithelial cells vs platelets), genetics (recombinant expression vs endogen expression) and also the species are different between the experiments. Our standard readout in platelets is P-selectin surface expression after stimulation. The results are depicted in bar diagrams and dose-response curves in Figure 8A of the submitted manuscript. After 30 min pre-treatment with 100 μ M VUF11207 and stimulation with 0.5 μ g/ml CRP the P-selectin surface expression was reduced by 86 % (**Figure VIII**

below). The general less effective ADP stimulation was suppressed by 69 %. The dose-response curve did indeed prove that 100 μ M VUF11207 is within the dynamic range of the agonistic effect of VUF11207 in human and murine platelets. This allows us to measure increase and decrease of the agonistic effect of VUF11207 by additional treatments and conditions.

Figure VIII: Extraction of panel a from Figure 8. **Upper left** Statistical analysis of platelet P-selectin (CD62P) flow cytometry signals after treatment with ACKR3 agonists (100 μ M; vehicle control: 1 % DMSO) and activation. **Upper right** Dose-response curves of P-selectin expression after treatment with increasing concentrations of ACKR3 agonists (VUF11207 or C10; control: C46) acquired by flow cytometry. Plotted: mean \pm S.E.M.; $n \geq 5$; statistics: one-way ANOVA.

C10 is a novel ACKR3 agonist identified by our lab using *in silico* computer modeling of the ACKR3 binding site. Agonists were designed to have comparable properties as other ACKR3 agonists but with a lower toxicity compared to commercially available compounds (**Figure IX**). However, our experiments using *Ackr3*^{f/f} and *Ackr3*^{-/-} animals provide strong evidence that the agonist C10 is specific for ACKR3 (please see **Figure 8 e/f**) and has indeed comparable properties as VUF11207 (please see **Figure 8 a-f**). Further, our *in vivo* experiments clearly show that systemic administration of agonist attenuates thrombosis in wild type animals and reduces myocardial ischemia and cell infiltration in *Ackr3*^{f/f} but not in *Ackr3*^{-/-} mice (please see **Figure 8 d,g-i**). Thus, we are convinced that these data are useful to support our conclusion of the manuscript as presented.

Figure IX: Cell viability performed in HEK293 cells after exposure to increasing concentration of ACKR3 agonists VUF11207 and C10 (CellTiter-Glo[®] 2.0 Cell Viability Assay, Promega, Madison, WI, USA).

6. Expression of ACKR3 on platelets is not granted as stated by the authors. In the past, the authors published western blots of platelets with various Mw for ACKR3: PubMed 25266363 50kD, PubMed 24668750 35 kD and PubMed 21468792 55kD. The actual size is now 42 kD, while reliable antibodies indicate ~55kD. What is detected? The response to my previous critique is insufficient. Were the samples boiled before loading on SDS PAGE (the question was ignored)? The resolution of the fluorescence microscopy images is poor and membrane localization of ACKR3 is not revealed.

Response to Reviewer #3:

We thank the reviewer for the valuable comments.

Point 1: Preparation of the SDS Page samples.

Immunoblots were prepared using a standard immunoblot protocol based on Cloning: A Laboratory Manual by J. Sambrook, E.R. Fritsch, and T. Maniatis, Molecular (1989)¹⁴. Before loading, the samples were lysed with 3x RIPA buffer, mixed with sample buffer containing 2 % SDS and boiled for 10 min at 96°C. Standard gel-electrophoresis and blotting were performed.

Point 2: Presence and molecular weight of ACKR3 in platelets.

The molecular weight of ACKR3 can vary broadly between tissues and species, as demonstrated by two examples from commercial sides (**Figure X E/F**). To compare the antibodies used in our lab and the 7TM antibody recommended by the reviewer, western blot analysis with the human platelet lysate treated with and without PNGase F were performed. PNGase F treatment removes N-glycosylations from proteins (PNGase F, P0704S, New England Biolabs, Ipswich, MA, USA). The antibody 7TM0089N (7TMantibodies) detects a band at approximately 55 kDa and a weaker signal at 50 kDa (**Figure X B**). The same two bands are detected by the Abcam antibody ab72100 and the NBP1-31309 antibody from NOVUS biologicals (**Figure X B**). The 50 kDa band becomes more prominent after 60 min PNGase F treatment. ACKR3 is constitutive ubiquitinated, which is key aspect for ACKR3 trafficking and recycling to the cell surface^{15,16}. Thus, the 50 kDa signal should represent ubiquitinated ACKR3 (41.5 + 8.6 kDa). Ab72100 and to a lower extent NBP1-31309 do also detect a band at 42 kDa, which corresponds to the predicted molecular weight of native ACKR3. This band is also primarily detected by our own manufactured ACKR3 antibody (**Figure X C**). The 55 kDa and 50 kDa band are also detected by our antibody, albeit weaker. This antibody binds to the N-terminal region of ACKR3 overlapping with the glycosylation sites. This might explain the preference for native ACKR3, as glycosylation may mask the binding sequence. 7TM0089N, ab72100 and NBP1-31309, do bind to intracellular regions of ACKR3 or the C-terminus of ACKR3. Thereby, they are not affected by the N-glycosylation of ACKR3. We therefore can identify three distinct forms of ACKR3 in platelets: native ACKR3 (42kDa), ACKR3-Ub (50 kDa), and glycosylated ACKR3-Ub (55 kDa) (**Figure X A/B**). Depending on the modification status of ACKR3 the mobility of ACKR3 changes and detection of immunoreactive bands depends on the applied antibody.

Regarding our previous publications, we indeed have published varying size for ACKR3 in western blots. 10 years ago, fewer antibodies and less knowledge about ACKR3 was available and even distributors like Abcam depicted ACKR3 at/around 37 kDa, e.g., ab72100, ab113410 or ab138509 (**Figure X D**). In previous blots shown in the revision and other publications we mostly concentrated on the calculated molecular weight of ACKR3 of 41.5 kDa, sometimes given as at/around 37 kDa. Only few blots depicted the higher molecular weight bands as α -tubulin (55 kDa) has been used as loading control. Therefore, the blot membranes were cut at 50 kDa to speed up the process and the pieces stained simultaneously for ACKR3 and α -tubulin or two differently label secondary antibody (IRDyae 680RD and IRDye 800WC labeling, Licor, Lincoln, NE, U.S.A.) were used at the same time. Thereby, higher molecular weight bands were masked by the strong α -tubulin signal of the control staining or lost due to the separation of the membrane pieces.

Figure X: **A** Schematic drawing of the ACKR3 receptor and its glycosylation sites with the binding sites of the antibodies used in this manuscript. **B** Immunoblot performed with lysates from human platelets treated with or without PNGaseF to remove N-glycosylation. The identical blots were tested with three different antibodies and three ACKR3 signals were identified (black arrow). **C** Immunoblot with our ACKR3 antibody generated against the extracellular tail of ACKR3 comprising the glycosylation sites. Again, the black arrow marked the three ACKR3 signals. **D** Recombinant Anti-GPCR RDC1/CXCR-7 antibody [EPR9321] (ab138509) (Source: <https://www.abcam.com/gpcr-rdc1cxcr-7-antibody-epr9321-ab138509.html>). **E** Abclonal, ACKR3 Polyclonal Antibody, A12712 (source: <https://www.antibodies.com/de/ackr3-antibody-a90458>) **F** MyBioSource.com, anti-ACKR3 antibody: Rabbit GPCR RDC1/CXCR7 Monoclonal Antibody, MBS4752782 (source: <https://www.mybiosource.com/monoclonal-ackr3-human-mouse-rat-antibody/gpcr-rdc1-cxcr7/4752782>).

To further validate our finding that ACKR3 is expressed in platelets, we obtained samples of isolated platelets and bone marrow samples from an ACKR3-GFP reporter mouse strain (GENSAT Project; 030077-UCD). These mice contain a BAC with GFP fused to the first part of ACKR3. The BAC expression is under the endogen ACKR3-promotor region and thereby reflects the endogen ACKR3 in the tissue albeit only as intracellular GFP signal as the transmembrane domain is not integral to the construct. Within bone marrow samples of the reporter mouse, we could identify CD42b and GFP double positive cells, representing megakaryocytes (**supplement Figure 1, Figure XI** below). Green fluorescence (488/GFP) was not detected in CD42b positive wild type cells using the same microscopic set-up. The platelet samples from the ACKR3 reporter strain and wild type control samples were spread (1 μ g/ml CRP) on fibrinogen coated cover slips. Immunofluorescence microscopy revealed an endogen GFP signal in platelets from the ACKR3 reporter strain, whereas no GFP signal could be detected in wild type platelets using the same microscopic settings. These results provide additional proof of ACKR3 presence in human and murine platelets (**supplement Figure 1, Figure XI** below).

Figure XI: **A** Megakaryocyte precursor cell isolated from wild type and ACKR3-GFP animals stained with GFP (left) and CD42b (right) antibodies. **B** Platelets isolated from wild type and ACKR3-GFP animals, activated with 1 μ g/ml CRP and spread on fibrinogen coated slides. An endogen ACKR3-GFP signal is visible in the platelets from ACKR3-GFP animals (scale bar = 10 μ m).

Point 3: Image quality regarding the microscopic imaging of platelets.

Human platelets are anucleate cell particles which are approximately 2–3 μm in diameter, murine platelets even smaller but densely packed with organelles and proteins. Spread platelets can reach a maximum of 10 μm in diameter but are very low in height (**Figure XII A**). We use a light microscope with a 100x objective and differential interference contrast (DIC or Normaski microscopy) to depict unstained platelets. Immuno-fluorescence microscopy on platelets also requires a light microscope with a 100x objective and sometimes is applied in combination with deconvolution techniques in our lab. Because of the small size of platelets, we are operating at the lower end of light microscopy resolution. Protein distribution in native/resting platelets is complicated to image and often appears as not more than a cycle, if the protein is integral to the outer membrane like in the sample below (**Figure XII B**). Distribution in spread platelets is easier to depict (**Figure XII B**), but the low height can cause problems with three-dimensional resolution. The applied microscopy analysis is well comparable with other platelet researchers in this field.

Figure XII: DIC and CD62P IFF staining samples of resting and activated platelets (1 $\mu\text{g}/\text{ml}$ CRP). The scale bar is 10 μm .

7. What is the meaning of the CRP secretome. The authors describe two genes up and 21 down, and?

Response to Reviewer #3:

We thank the reviewer for this legitimate question. The experiment was performed on specific request of reviewer 2. The results indicate that the actual secretome is only marginally altered between wildtype and knock out. Instead, proteins associated with mitochondrial metabolism and microvesicles are significantly reduced in the *Ackr3*^{-/-} animals. Platelets do shed microvesicles. They can include mitochondria and fuse to other cells types, thereby directly transmitting signals, e.g., activation or apoptosis signals, into the recipient cell. Based on the reduced amount of proteins connected to microvesicles and mitochondria in APS from *Ackr3*^{-/-} platelets we could hypothesize that this specific cell-cell interaction pathway is reduced in *Ackr3*^{-/-} animals and thus could lead to the phenotype observed post I/R. But we will need to further analyze the results and perform additional experiments beyond the scope of this revision to be fully able to interpret this finding.

8. Incubation of human monocytes with whole mouse blot (line 180) is not clear. This experimental setup may induce a plethora of responses that might not be correlated with platelet ACKR3 expression.

Response to Reviewer #3:

The general-set up of the experiment has been published previously¹⁷. The perfusion (**Figure 4b, Figure XIII** below) was performed with isolated human monocytes and isolated murine platelet. The number of circulating monocytes needed for the perfusion experiments was too high to be isolated from mice. Therefore, isolated monocytes from human blood had to be used in the experiment. The fact that isolated cells were used has been highlighted in the manuscript. The putative effect of an interspecies experiment was also reduced by using isolated cell populations. The static adhesion (**Figure 4c, Figure XIII** below), however, required fewer monocytes and therefore was performed with murine monocytes. We apologize for not having thoroughly described the experimental conditions. The manuscript has been corrected accordingly. Both adhesion experiments showed a significant higher monocyte/platelet interaction in *Ackr3*^{-/-} platelets compared to wild type ($p < 0.05$).

Figure XIII: Extract of Figure 4. **b** Representative images of flow chamber experiments were performed with isolated human monocytes. The monocytes were perfused over *Ackr3*^{fl/fl} (left panel) and *Ackr3*^{-/-} (right panel) platelets spread (1 μ g/ml CRP) on fibrinogen-coated cover slides. Yellow arrow heads point at rolling monocytes whereas asterisks indicate adhesive monocytes (scale bar: 100 μ m). Statistical analysis of the number of rolling monocytes within 40 sec of perfusion. An enhanced adhesion between *Ackr3*^{-/-} platelets and human monocytes compared to *Ackr3*^{fl/fl} platelets was observed, $n = 6$. **c** Statistical analysis of the static adhesion of *Ackr3*^{-/-} platelets to murine monocytes compared to wild type control platelets after a 2 h incubation period. An enhanced adhesion between *Ackr3*^{-/-} platelets and murine monocytes was observed compared to *Ackr3*^{fl/fl} platelets after thrombin (0.1 U/ml) stimulation, $n \geq 4$. Plotted: mean \pm S.E.M.; statistics: student's t-test; ns = not significant, * $p < 0.05$.

9. The conclusion from the NanoString experiments (lines 217-219) is farfetched. Given the expression of Pf4 in different cells the dramatically enhanced inflammation cannot be attributed only to lack of ACKR3 expression in platelets.

Response to Reviewer #3:

As outlined above, the expression of Pf4 in various cells is not a likely explanation for the observed altered transcriptome expression in inflamed tissue. Our I/R experiments with PF4 Cre animals, flow cytometry analysis of immune cell populations (bone marrow and spleen) and the recent literature (Ishizuka *et al.*, Nature, 2021; PubMed: 34099846) make it unlikely that the PF4 expression outside the megakaryocyte lineage may cause the observed *Ackr3*^{-/-} phenotype after I/R and the *ex vivo* data gained with isolated platelets¹².

10. The finding that CXCL12, which has about 10-fold higher affinity for ACKR3 than CXCL11, has no stimulatory capacity on platelets compared to CXCL11 needs explanation.

Response to Reviewer #3:

Indeed, this is an interesting observation by the reviewer. We used a flow chamber assay to analyze the effect of endogenous ACKR3 ligands on *ex vivo* thrombus formation. This assay allows a multifactorial activation process, combining chemokine treatment of whole blood samples and shear

stress. All three tested chemokines, MIF, CXCL11 and CXCL12, are known to activate ACKR3 in other cell types. CXCL11 binds exclusive to ACKR3 and has been shown to activate platelet ACKR3¹⁸. In other cell types MIF can interact with CXCR4 and ACKR3 but platelets are missing the co-factor CD74, which is important for CXCR4 activation by MIF^{19,20}. Therefore, MIF can bind platelet CXCR4 but cannot trigger platelet function via this pathway. Thus, pre-treatment of samples with CXCL11 or MIF in flow chamber experiments results in an ACKR3-dependent platelet modulation excluding CXCR4 as concomitant receptor. In contrast, CXCL12 can bind to CXCR4 and ACKR3 in platelets, thus generating a complex interplay between the receptors and the underlying signaling pathways^{21,22,2}. Therefore, the above considerations may explain the differential effects of CXCL11, CXCL12 and MIF on platelet function.

11. Minor:

Line 200: should read Ly6G+

CXCR7 was partially replaced with ACKR3, but not consistently including supplementary material, figures, tables and legends

Supplementary figure legend 9: should read "volcano plot"

In general, the paper would benefit from rigorous editing (typos and grammar).

Response to Reviewer #3:

We apologize for these mistakes and have corrected our manuscript accordingly.

Reviewer 4

Reviewer #4 (Remarks to the Author):

Impressed by the amount of data generated for this rebuttal. Unbelievable

Response to Reviewer #4:

We thank the reviewer for his/her favorable review that has helped significantly improve our manuscript.

References:

1. Komolov KE, Benovic JL. G protein-coupled receptor kinases: Past, present and future. *Cell Signal* **41**, 17-24 (2018).
2. Saaber F, et al. ACKR3 Regulation of Neuronal Migration Requires ACKR3 Phosphorylation, but Not beta-Arrestin. *Cell Rep* **26**, 1473-1488 e1479 (2019).
3. Jean-Charles PY, Kaur S, Shenoy SK. G Protein-Coupled Receptor Signaling Through beta-Arrestin-Dependent Mechanisms. *J Cardiovasc Pharmacol* **70**, 142-158 (2017).
4. Zarca A, et al. Differential Involvement of ACKR3 C-Tail in beta-Arrestin Recruitment, Trafficking and Internalization. *Cells* **10**, (2021).
5. Hoffmann F, et al. Rapid uptake and degradation of CXCL12 depend on CXCR7 carboxyl-terminal serine/threonine residues. *J Biol Chem* **287**, 28362-28377 (2012).
6. Le Mercier A, et al. GPR182 is an endothelium-specific atypical chemokine receptor that maintains hematopoietic stem cell homeostasis. *Proc Natl Acad Sci U S A* **118**, (2021).
7. Ehrlich AT, et al. Akr3-Venus knock-in mouse lights up brain vasculature. *Mol Brain* **14**, 151 (2021).
8. Ameti R, et al. Characterization of a chimeric chemokine as a specific ligand for ACKR3. *J Leukoc Biol* **104**, 391-400 (2018).
9. Gollomp K, Poncz M. Gp1ba-Cre or Pf4-Cre: pick your poison. *Blood* **133**, 287-288 (2019).
10. Sierra F, et al. Disrupted cardiac development but normal hematopoiesis in mice deficient in the second CXCL12/SDF-1 receptor, CXCR7. *Proc Natl Acad Sci U S A* **104**, 14759-14764 (2007).
11. Tabula Muris C, et al. Single-cell transcriptomics of 20 mouse organs creates a Tabula Muris. *Nature* **562**, 367-372 (2018).
12. Ishizuka M, et al. Author Correction: CXCR7 ameliorates myocardial infarction as a beta-arrestin-biased receptor. *Sci Rep* **11**, 12340 (2021).
13. Wijtman M, et al. Synthesis, modeling and functional activity of substituted styrene-amides as small-molecule CXCR7 agonists. *Eur J Med Chem* **51**, 184-192 (2012).
14. Sambrook J, Fritsch, E.R., Maniatis, T. *Molecular Cloning: A Laboratory Manual, 2nd ed.* Cold Spring Harbor Laboratory Press, Cold Spring Harbor, New York (1989).
15. Canals M, Scholten DJ, de Munnik S, Han MK, Smit MJ, Leurs R. Ubiquitination of CXCR7 controls receptor trafficking. *PLoS One* **7**, e34192 (2012).
16. Chatterjee M, et al. SDF-1alpha induces differential trafficking of CXCR4-CXCR7 involving cyclophilin A, CXCR7 ubiquitination and promotes platelet survival. *FASEB J* **28**, 2864-2878 (2014).
17. Chatterjee M, et al. Platelet-derived CXCL12 regulates monocyte function, survival, differentiation into macrophages and foam cells through differential involvement of CXCR4-CXCR7. *Cell Death Dis* **6**, e1989 (2015).
18. Chatterjee M, Rath D, Gawaz M. Role of chemokine receptors CXCR4 and CXCR7 for platelet function. *Biochem Soc Trans* **43**, 720-726 (2015).
19. Schwartz V, et al. A functional heteromeric MIF receptor formed by CD74 and CXCR4. *FEBS Lett* **583**, 2749-2757 (2009).
20. Chatterjee M, et al. Macrophage migration inhibitory factor limits activation-induced apoptosis of platelets via CXCR7-dependent Akt signaling. *Circ Res* **115**, 939-949 (2014).
21. Walsh TG, Harper MT, Poole AW. SDF-1alpha is a novel autocrine activator of platelets operating through its receptor CXCR4. *Cell Signal* **27**, 37-46 (2015).
22. Rath D, et al. Expression of stromal cell-derived factor-1 receptors CXCR4 and CXCR7 on circulating platelets of patients with acute coronary syndrome and association with left ventricular functional recovery. *Eur Heart J* **35**, 386-394 (2014).

REVIEWERS' COMMENTS

Reviewer #2 (Remarks to the Author):

The authors again invested major efforts to address the reviewers' comments. The questions raised were addressed appropriately. This reviewer is overall impressed by the extent of the work presented and the novelty of the data.

Reviewer #3 (Remarks to the Author):

The authors have addressed most concerns, however, mostly discussing the critiques without adjusting the manuscript.

The use of exuberant concentrations of VUF11207 and C10 must be pointed out and critically discussed. C10 must be described.

The different Mw of ACKR3 on western blots is not clear. Again a critical discussion is missing. The authors should be aware that most GPCR's when boiled aggregate and do not resolve on SDS-PAGE (see response to reviewer).

Reviewer #4 (Remarks to the Author):

I have no further comments

Nature Communications NCOMMS-20-50883B

Title: ACKR3 regulates platelet activation and ischemia - reperfusion tissue injury

Corresponding Author: Prof. Meinrad Gawaz

Response to the reviewers

We thank the reviewers for their positive assessment of our manuscript. In the following we answer their questions step by step.

Comments of Reviewer #1:

The authors again invested major efforts to address the reviewers' comments. The questions raised were addressed appropriately. This reviewer is overall impressed by the extent of the work presented and the novelty of the data.

Response to Reviewer #1:

We thank the reviewer for his/her favorable and constructive comments which helped us significantly improve our manuscript.

Comments of Reviewer #3:

The authors have addressed most concerns, however, mostly discussing the critiques without adjusting the manuscript. The use of exuberant concentrations of VUF11207 and C10 must be pointed out and critically discussed. C10 must be described. The different Mw of ACKR3 on western blots is not clear. Again a critical discussion is missing. The authors should be aware that most GPCR's when boiled aggregate and do not resolve on SDS-PAGE (see response to reviewer).

Response to Reviewer #3:

We thank the reviewer again for his/her favorable and constructive comments. As suggested we have now added a paragraph to our revised manuscript regarding the concentrations of the ACKR3-agonists VUF11207 and C10. In addition, we have added the chemical description of the C10 agonist to the method section.

Further, we have added a paragraph to the discussion section concerning the molecular mobility of ACKR3 using various anti-ACKR3 monoclonal antibodies for immunoblotting.

Comments of Reviewer #4:

I have no further comments

Response to Reviewer #4:

We thank the reviewer for his/her favorable and constructive comments which helped us significantly improve our manuscript.